# Federated Learning with Client Subsampling, Data Heterogeneity, and Unbounded Smoothness: A New Algorithm and Lower Bounds

**Michael Crawshaw**[*]
Department of Computer Science
George Mason University
Fairfax, VA 22030, USA
mcrawsha@gmu.edu

**Yajie Bao**[*]
School of Mathematical Sciences
Shanghai Jiao Tong University
Shanghai, China
baoyajie2019stat@sjtu.edu.cn

**Mingrui Liu**[†]
Department of Computer Science
George Mason University
Fairfax, VA 22030, USA
mingruil@gmu.edu

## Abstract

We study the problem of Federated Learning (FL) under client subsampling and data heterogeneity with an objective function that has potentially unbounded smoothness. This problem is motivated by empirical evidence that the class of relaxed smooth functions, where the Lipschitz constant of the gradient scales linearly with the gradient norm, closely resembles the loss functions of certain neural networks such as recurrent neural networks (RNNs) with possibly exploding gradient. We introduce EPISODE++, the first algorithm to solve this problem. It maintains historical statistics for each client to construct control variates and decide clipping behavior for sampled clients in the current round. We prove that EPISODE++ achieves linear speedup in the number of participating clients, reduced communication rounds, and resilience to data heterogeneity. Our upper bound proof relies on novel techniques of recursively bounding the client updates under unbounded smoothness and client subsampling, together with a refined high probability analysis. In addition, we prove a lower bound showing that the convergence rate of a special case of clipped minibatch SGD (without randomness in the stochastic gradient and with randomness in client subsampling) suffers from an explicit dependence on the maximum gradient norm of the objective in a sublevel set, which may be large. This effectively demonstrates that applying gradient clipping to minibatch SGD in our setting does not eliminate the problem of exploding gradients. Our lower bound is based on new constructions of hard instances tailored to client subsampling and a novel analysis of the trajectory of the algorithm in the presence of clipping. Lastly, we provide an experimental evaluation of EPISODE++ when training RNNs on federated text classification tasks, demonstrating that EPISODE++ outperforms strong baselines in FL. The code is available at https://github.com/MingruiLiu-ML-Lab/episode_plusplus.

---

[*]Equal Contribution.
[†]Corresponding Author.

37th Conference on Neural Information Processing Systems (NeurIPS 2023).

Table 1: Best complexity to find an $\epsilon$-stationary point for various methods and settings. The setting column describes the features of the setting in which the algorithm can solve the problem: (Re) denotes relaxed smoothness, (H) heterogeneous data, and (S) client subsampling. $\sigma$: stochastic gradient noise, $\kappa$: client heterogeneity, $\Delta$: objective gap at the initial solution, $I$: local steps, $R$: communication rounds, $T = RI$: iteration complexity, $N$: number of clients, $S$: number of subsampled clients. $^\dagger$ denotes a high probability guarantee. $\tilde{O}(\cdot)$ and $\tilde{\Omega}(\cdot)$ omit logarithmic terms.

| Method | Communication Complexity ($R$) | Best Iteration Complexity | Largest $I$ to guarantee linear speedup | Setting |
|---|---|---|---|---|
| Local SGD [48] | $O\left(\frac{\Delta L\sigma^2}{NI\epsilon^4} + \frac{\Delta L\kappa^2 NI}{\sigma^2\epsilon^2} + \frac{\Delta LN}{\epsilon^2}\right)$ | $O\left(\frac{L\sigma^2}{N\epsilon^4}\right)$ | $O\left(\frac{\sigma^2}{\kappa N\epsilon}\right)$ | (H) |
| SCAFFOLD [24] | $O\left(\frac{\Delta L\sigma^2}{SI\epsilon^4} + \frac{\Delta L}{\epsilon^2}\right)$ | $O\left(\frac{\Delta L\sigma^2}{S\epsilon^4}\right)$ | $O\left(\frac{\sigma^2}{N\epsilon^2}\right)$ | (H), (S) |
| CELGC [32] | $O\left(\frac{\Delta L_0\sigma^2}{NI\epsilon^4}\right)$ | $O\left(\frac{\Delta L_0\sigma^2}{N\epsilon^4}\right)$ | $O\left(\frac{\sigma}{N\epsilon}\right)$ | (Re) |
| EPISODE [9] | $O\left(\frac{\Delta L_0\sigma^2}{NI\epsilon^4} + \frac{\Delta(L_0+L_1(\kappa+\sigma))}{\epsilon^2}\left(1+\frac{\sigma}{\epsilon}\right)\right)$ | $O\left(\frac{\Delta L_0\sigma^2}{N\epsilon^4}\right)$ | $O\left(\frac{L_0\sigma^2}{\left(L_0+L_1(\kappa+\sigma)\left(1+\frac{\sigma}{\epsilon}\right)\right)N\epsilon^2}\right)$ | (Re), (H) |
| EPISODE++ (Theorem 1)$^\dagger$ | $\tilde{O}\left(\frac{\Delta L_0\sigma^2}{SI\epsilon^4} + \frac{\Delta(L_0+L_1(\kappa+\rho\sigma))}{I\epsilon^3}\frac{L_0}{L_1\rho}\right)$ | $\tilde{O}\left(\frac{\Delta L_0\sigma^2}{S\epsilon^4}\right)$ | $\tilde{O}\left(\frac{L_0\sigma^2}{\left(L_0+L_1(\kappa+\rho\sigma)\right)\left(\sigma+\frac{L_0}{L_1\rho}\right)S\epsilon}\right)$ | (Re), (H), (S) |
| Clipped Minibatch SGD (Theorem 2) | $\tilde{\Omega}\left(\frac{\Delta L_1 M}{\epsilon^2}\right)$ | $\tilde{\Omega}\left(\frac{\Delta L_1 MI}{\epsilon^2}\right)$ | - | (Re), (H), (S) |

# 1  Introduction

Federated Learning (FL) [33, 23] is a distributed learning paradigm in which many clients collaboratively train a machine learning model while communicating over a network, which preserves privacy and leverages parallelism across many clients. Minimizing communication cost, accounting for data heterogeneity across clients, and allowing for partial client participation are core principles of FL. Interest in FL has grown in recent years, especially with user-facing applications such as next word prediction on smartphones [19].

Optimization is a central part of FL algorithms, and most work on non-convex optimization assumes a smooth objective in both the single machine setting [15, 16, 1] and the FL setting [41, 48, 24, 26]. However, recent work [52, 8] has shown empirical evidence that certain neural networks (LSTMs [21], and Transformers [42]) do not satisfy this assumption, but do satisfy a weaker condition known as *relaxed smoothness* [52]. Under relaxed smoothness, techniques such as gradient clipping [37] are essential for avoiding exploding gradients. To avoid the negative effects of exploding gradients, GD without gradient clipping requires a step size inversely proportional to the maximum gradient norm of the objective in a sublevel set (denoted as $M$), resulting in very slow convergence. The usefulness of gradient clipping under relaxed smoothness matches observations of training these neural networks in practice, for example on natural language tasks, which are common in FL [23].

However, little work yet exists for FL in the relaxed smoothness setting. Liu et al. [32] introduced a communication-efficient gradient clipping algorithm for FL under relaxed smoothness, with the additional assumption of homogeneous client data and a distributional assumption on the noise of stochastic gradients. The EPISODE algorithm [9] was subsequently introduced to handle heterogeneous client data in this setting, but requires full client participation, that is, that every client participates in every communication round. This significantly decreases the practical applicability of EPISODE, since full client participation is rarely achievable with large-scale FL in practice [23].

In this work, we introduce EPISODE++, the first algorithm for FL under relaxed smoothness, client heterogeneity, and client subsampling. EPISODE++ maintains statistics of the history of gradients for each client, and uses these statistics to (1) correct each local update step to approximate an update on the global loss and (2) determine at which steps the clipping operation should be performed. We prove that EPISODE++ achieves linear speedup in the number of participating clients, has reduced communication cost, and enjoys convergence rate independent of client heterogeneity.

A previous line of work in the non-convex smooth stochastic setting [45, 47, 27] compares FL algorithms against a classical baseline: minibatch SGD [40]. Despite the impressive results of newer

algorithms in practice, only a few works have proven that some FL algorithms have a theoretical advantage over minibatch SGD [24, 27]. It is therefore natural to ask what is the analogue of minibatch SGD in the relaxed smooth setting, and whether it is possible to improve upon this analogue. In this work we consider clipped minibatch SGD, which is obtained by applying gradient clipping to each update of minibatch SGD. We demonstrate a surprising negative result for clipped minibatch SGD: under client subsampling, relaxed smoothness, and client heterogeneity, the convergence rate of clipped minibatch SGD still depends on $M$. This implies that *gradient clipping does not help minibatch SGD* in this setting.

Our contributions can be summarized as follows:

- We introduce EPISODE++, the first algorithm for FL under relaxed smoothness, client heterogeneity, and client subsampling. We prove that EPISODE++ achieves linear speedup in the number of participating clients, reduced communication cost, and has convergence rate independent of client heterogeneity. Table 1 shows a detailed comparison of the complexity of various algorithms. To achieve this result, we introduce novel techniques of recursively bounding the client updates in the presence of unbounded smoothness and client data heterogeneity, together with a refined high probability analysis.

- We demonstrate a lower bound for clipped minibatch SGD in which the convergence rate depends on $M$ (the maximum gradient norm of the objective function in a sublevel set). This shows that, in our setting, clipped minibatch SGD is susceptible to exploding gradients, and avoiding them requires a very small learning rate, which slows down convergence. Our lower bound is based on new constructions of hard instances tailored to client subsampling and a novel analysis of the trajectory of the algorithm in the presence of clipping.

- We empirically evaluate EPISODE++ against strong baselines when training RNNs on federated text classification tasks. The results show that EPISODE++ consistently outperforms baselines across various client participation ratios and is resilient to heterogeneous client data, which is consistent with our theory.

## 2 Related Work

**Federated Learning** FL was proposed by [33], where the authors designed the FedAvg algorithm, which is also referred to as local SGD in the literature [41, 31, 48]. The local SGD algorithm has been analyzed in various settings, including convex smooth setting [41, 11, 28, 24, 45, 47, 26, 49, 17, 25], convex composite setting [50, 2, 35], and nonconvex smooth setting [22, 43, 31, 18, 48, 28, 24, 38, 53, 26]. There is a line of work which specially considered client partial participation in FL [6, 13, 24, 29, 44, 7] under convex or nonconvex smooth settings. Recently, Liu et al. [32] and Crawshaw et al. [9] considered FL with nonconvex and relaxed smooth functions for homogeneous and heterogeneous data respectively. However, they assume full client participation and neither of them are applicable to the case of client subsampling.

**Relaxed Smoothness** Relaxed smoothness was proposed by [52] as a relaxation of the standard smoothness condition, which is used to model the exploding gradient problem in training deep neural networks such as recurrent neural networks [36, 37] and long-short term memory networks [52], language models [14, 34] and transformers [8]. Zhang et al. [52] proved that gradient clipping converges faster than any fixed step size gradient descent for relaxed smooth functions. The complexity bound in [52] was further improved by [51]. Recently, there is a line of work which considered different algorithms and various analyses under relaxed smoothness [8, 39, 12]. However, all of them focused on single machine setting and may not be applicable to FL setting.

**Lower bounds in Federated Learning** There are several lower bound results for FL algorithms. Woodworth et al. [45, 47] compared minibatch SGD and local SGD in the regime of federated stochastic convex optimization setting for homogeneous and heterogeneous data and established lower bounds for local SGD. Woodworth et al. [46] proved a min-max complexity of distributed stochastic convex optimization for any intermittent communication algorithm. Glasgow et al. [17] established improved lower bounds of local SGD in convex optimization setting for both homogeneous and heterogeneous data. However, all of these lower bounds are not applicable to our setting where the problem instance is relaxed smooth with heterogeneous data and client subsampling.

# 3 Problem Setup

We consider federated learning with heterogeneous and stochastic objectives, where the goal is to minimize the average loss function across $N$ clients. For $i \in [N]$, let $f_i(\boldsymbol{x}) = \mathbb{E}_{\xi \sim \mathcal{D}_i}[F_i(\boldsymbol{x}; \xi)]$ be the objective of the $i$-th client, where $\mathcal{D}_i$ is the underlying data distribution of the $i$-th client. Then the global objective is

$$\min_{\boldsymbol{x} \in \mathbb{R}^d} \left\{ f(\boldsymbol{x}) := \frac{1}{N} \sum_{i=1}^{N} f_i(\boldsymbol{x}) \right\}. \tag{1}$$

Since each $f_i$ is not necessarily convex, we consider the problem of finding an $\epsilon$-stationary point, that is, a point $\boldsymbol{x} \in \mathbb{R}^d$ such that $\|\nabla f(\boldsymbol{x})\| \leq \epsilon$.

Most works on non-convex optimization [15, 16, 1] consider the case where each $f_i$ is $L$-smooth, so that $\|\nabla^2 f_i(\boldsymbol{x})\| \leq L$ for every $\boldsymbol{x} \in \mathbb{R}^d$. However, several works [52, 8] have shown empirical evidence that objective functions corresponding to some types of neural networks (such as LSTMs [21] and Transformers [42]) do not satisfy this condition, but do satisfy a strictly weaker condition known as $(L_0, L_1)$-smoothness, or relaxed smoothness. A second-order differentiable function $g : \mathbb{R}^d \to \mathbb{R}$ is called $(L_0, L_1)$-smooth if $\|\nabla^2 g(\boldsymbol{x})\| \leq L_0 + L_1 \|\nabla g(\boldsymbol{x})\|$ for all $\boldsymbol{x} \in \mathbb{R}^d$. Notice that any $L$-smooth function is $(L, 0)$-smooth.

In this work we consider the problem described in (1) under the following assumptions.

**Assumption 1.** *(i) Denoting by $\boldsymbol{x}_0$ the initial iterate, there exists some $\Delta > 0$ such that $f(\boldsymbol{x}_0) - \min_{\boldsymbol{x} \in \mathbb{R}^d} f(\boldsymbol{x}) \leq \Delta$. (ii) Each $f_i$ and $f$ is $(L_0, L_1)$-smooth. (iii) There exist $\kappa \geq 0$, $\rho \geq 1$ such that $\|\nabla f_i(\boldsymbol{x})\| \leq \kappa + \rho \|\nabla f(\boldsymbol{x})\|$ for all $\boldsymbol{x} \in \mathbb{R}^d$. (iv) There exists $\sigma \geq 0$ such that $\mathbb{E}_{\xi \sim \mathcal{D}_i}[\nabla F_i(\boldsymbol{x}; \xi)] = \nabla f_i(x)$ and $\|\nabla F_i(\boldsymbol{x}; \xi) - \nabla f_i(\boldsymbol{x})\| \leq \sigma$ almost surely for $\xi \sim \mathcal{D}_i$.*

Assumption $(i)$ is standard in non-convex optimization [15, 16]. Assumption $(ii)$ is typically used in the FL literature [32, 9]. Assumption $(iii)$ is used in heterogeneous federated learning [24] and describes the heterogeneity between client objectives: if $\kappa = 0$ and $\rho = 1$, then all client objectives $f_i$ are equal. Assumption $(iv)$ is common in the relaxed smoothness setting [52, 51, 32, 8, 9].

In addition, we consider the case of partial client participation in federated learning, also known as client subsampling. With partial participation, only $S$ out of $N$ clients will participate in each communication round, which exacerbates the issue of client heterogeneity.

# 4 Algorithm and Convergence Analysis

## 4.1 Main Challenges and Algorithm Design

We first illustrate why existing algorithms such as SCAFFOLD [24] and EPISODE [9] are not able to handle heterogeneous data, relaxed smoothness and client subsampling simultaneously. The analysis of SCAFFOLD crucially requires the function to be $L$-smooth to recursively bound (1) the lag error from client subsampling and (2) client drift from local updates, but this argument is not applicable for relaxed smooth functions whose gradient information changes quickly. EPISODE has convergence guarantees for relaxed smooth functions and heterogeneous data, but only with full client participation. A naive variant of EPISODE in the client subsampling case does not work: the indicator of gradient clipping is based only on information from clients participating in the current round and ignores information from unsampled clients, which introduces non-negligible bias from client heterogeneity.

To address these challenges, we design a new algorithm named EPISODE++, which is presented in Algorithm 1. Similar to EPISODE [9], our algorithm utilizes *episodic gradient clipping*, which determines whether a clipping operation will be performed depending on the size of the average control variate $\boldsymbol{G}_r$. This means that during each round, either (1) all clients perform a normalized update for all steps, or (2) all clients will perform an unnormalized update for all steps. However, different from EPISODE, our algorithm corrects local updates with control variates $\boldsymbol{G}_r^i$ computed as the averaged stochastic gradient over the previous round in which client $i$ participated, as in SCAFFOLD. As we will show in our proof, the episodic gradient clipping together with the update correction strategy allows the algorithm to progress in a stable manner: and it will sufficiently decrease the objective value and also avoid the negative effect of possibly exploding gradients.

---

**Algorithm 1** EPISODE++

1: Initialize $\bar{x}_0$, $G_0^i \leftarrow \nabla F_i(\bar{x}_0, \tilde{\xi}_i)$, $G_0 \leftarrow \frac{1}{N}\sum_{i=1}^N G_0^i$
2: **for** $r = 0, 1, \ldots, R-1$ **do**
3:    Sample $\mathcal{S}_r \subset [N]$ uniformly at random such that $|\mathcal{S}_r| = S$
4:    **for** $i \in \mathcal{S}_r$ **do**
5:       $x_{r,0}^i \leftarrow \bar{x}_r$
6:       **for** $k = 0, \ldots, I-1$ **do**
7:         Sample $\nabla F_i(x_{r,k}^i; \xi_{r,k}^i)$, where $\xi_{r,k}^i \sim \mathcal{D}_i$
8:         $g_{r,k}^i \leftarrow \nabla F_i(x_{r,k}^i; \xi_{r,k}^i) - G_r^i + G_r$
9:         $x_{r,k+1}^i \leftarrow x_{r,k}^i - \eta g_{r,k}^i \mathbb{1}_{\|G_r\| \leq \gamma/\eta} - \gamma \frac{g_{r,k}^i}{\|g_{r,k}^i\|}\mathbb{1}_{\|G_r\| \geq \gamma/\eta}$
10:       **end for**
11:       $G_{r+1}^i \leftarrow \frac{1}{I}\sum_{k=0}^{I-1} \nabla F_i(x_{r,k}^i; \xi_{r,k}^i)$
12:       $\Delta G_r^i \leftarrow G_{r+1}^i - G_r^i$
13:    **end for**
14:    Update $\bar{x}_{r+1} \leftarrow \frac{1}{S}\sum_{i \in \mathcal{S}_r} x_{r,I}^i$
15:    Update $G_{r+1} \leftarrow G_r + \frac{1}{N}\sum_{i \in \mathcal{S}_r} \Delta G_r^i$
16:    Denote $G_{r+1}^i \leftarrow G_r^i$ for all $i \notin \mathcal{S}_r$ [3]
17: **end for**

---

## 4.2 Convergence Result

The following result proves that EPISODE++ converges to an $\epsilon$-stationary point with high probability.

**Theorem 1.** *Let $\epsilon \leq \frac{AL_0}{16BL_1\rho}$ and $\delta \in (0,1)$. Denote $K = \left\lceil \frac{\log(RN/\delta)}{\log(N/(N-S))} \right\rceil$, $\Gamma_1 := AL_0 + BL_1\kappa + 4BL_1\rho\left(2\sigma + \frac{\gamma}{\eta}\right)$ and $\Gamma_2 := 64\left(\kappa + 5\rho\left(2\sigma + \frac{\gamma}{\eta}\right)\right)^2\left(\frac{1}{S} - \frac{1}{N}\right)$. If*

$$\eta \leq \min\left\{\frac{1}{90(K+1)\Gamma_1 I}, \frac{\epsilon}{32\Gamma_1 I\left(74\sigma + \frac{AL_0}{BL_1\rho}\right)}, \frac{S\epsilon^2}{216AL_0\sigma^2\log\frac{1}{\delta}}, \frac{\Delta}{\log\frac{1}{\delta}}\min\left\{\frac{1}{12\sigma^2}, \frac{1}{\Gamma_2 I}\right\}\right\}, \tag{2}$$

*and $\gamma = \left(72\sigma + \frac{AL_0}{BL_1\rho}\right)\eta$, then Algorithm 1 satisfies $\frac{1}{R}\sum_{r=0}^{R-1}\|\nabla f(\bar{x}_r)\| \leq 35\epsilon$ with probability at least $1 - 15\delta$, as long as $R \geq \frac{8\Delta}{\epsilon^2\eta I}$.*

The above result holds for a wide range of $\eta$ and $I$, and for any noise level $\sigma$. The corollary below summarizes the best possible iteration complexity and communication complexity implied by Theorem 1. The full proofs of Theorem 1 and Corollary 1 can be found in Appendix A.5 and A.6.

**Corollary 1.** *Suppose $\sigma > 0$. If, under the setting of Theorem 1, we additionally choose $\epsilon \leq \min\left\{\frac{AL_0}{16BL_1\rho}, \frac{16\left(74\sigma + \frac{AL_0}{BL_1\rho}\right)}{45(K+1)}, \sqrt{\frac{18\Delta AL_0}{S}}, \frac{32\Delta\Gamma_1\left(74\sigma + \frac{AL_0}{BL_1\rho}\right)}{\Gamma_2 \log\frac{1}{\delta}}\right\}$, $\eta$ as large as possible under (2), and $I \leq \frac{27AL_0\sigma^2\log\frac{1}{\delta}}{4\epsilon\Gamma_1 S\left(74\sigma + \frac{AL_0}{BL_1\rho}\right)}$, then Algorithm 1 has iteration complexity $RI = O\left(\frac{\Delta L_0\sigma^2\log\frac{1}{\delta}}{S\epsilon^4}\right)$. If additionally $I = \Theta\left(\frac{L_0\sigma^2\log\frac{1}{\delta}}{\epsilon S(L_0 + L_1(\kappa+\rho\sigma))\left(\sigma + \frac{L_0}{L_1\rho}\right)}\right)$, then Algorithm 1 has communication complexity $R = O\left(\frac{\Delta(L_0 + L_1(\kappa+\rho\sigma))\left(\sigma + \frac{L_0}{L_1\rho}\right)}{\epsilon^3}\right)$.*

---

[3]Although it appears that this operation must be performed by unsampled clients, this is just an artifact of the notation: unsampled clients do not have to execute any operations.

## 4.3 Proof Sketch

In this section we provide a sketch for the proof of Theorem 1. We wish to establish the descent inequality for the global objective function in each round by applying Lemma A.3 in [51]:

$$f(\bar{\boldsymbol{x}}_{r+1}) \leq f(\bar{\boldsymbol{x}}_r) + \langle \nabla f(\bar{\boldsymbol{x}}_r), \bar{\boldsymbol{x}}_{r+1} - \bar{\boldsymbol{x}}_r \rangle + \frac{AL_0 + BL_1 \|\nabla f(\bar{\boldsymbol{x}}_r)\|}{2} \|\bar{\boldsymbol{x}}_{r+1} - \bar{\boldsymbol{x}}_r\|^2, \quad (3)$$

where $A = 1 + e^C - \frac{e^C - 1}{C}$, $B = \frac{e^C - 1}{C}$ and $C > 0$ is an absolute constant. However, using this descent inequality requires $\|\bar{\boldsymbol{x}}_{r+1} - \bar{\boldsymbol{x}}_r\| \leq C/L_1$. Achieving a universal bound on the distance over two consecutive rounds is nontrivial in the presence of client subsampling, data heterogeneity and relaxed smoothness, which requires a new analysis.

Compared with SCAFFOLD [24] and EPISODE [9], the main difficulty of analyzing EPISODE++ lies in controlling the distance between local weights $\boldsymbol{x}_{r,k}^i$ and the synchronization point $\bar{\boldsymbol{x}}_r$ under relaxed smoothness and client subsampling. In EPISODE, the magnitude of $g_{r,k}^i$ can be bounded in terms of $\|\nabla f_i(\boldsymbol{x}_{r,k}^i) - \nabla f_i(\bar{\boldsymbol{x}}_r)\|$ and $\|\boldsymbol{G}_r\|$, since $\boldsymbol{G}_r^i$ is evaluated at the synchronization point $\bar{\boldsymbol{x}}_r$. Then the distance can be bounded recursively when clipping does not happen (i.e., $\|\boldsymbol{G}_r\| \leq \gamma/\eta$). However, EPISODE++ utilizes historical gradients to construct the indicator $\boldsymbol{G}_r$, which means the increment also depends on the lag error $\|\boldsymbol{G}_r^i - \nabla f_i(\bar{\boldsymbol{x}}_r)\|$. Due to relaxed smoothness, we cannot bound the lag error as in SCAFFOLD [24] because we do not know the distance between $\bar{\boldsymbol{x}}_r$ and where $\boldsymbol{G}_r^i$ is evaluated. In the extreme case that client $i$ has never been sampled before round $r$, we have $\boldsymbol{G}_r^i = \nabla F_i(\bar{\boldsymbol{x}}_0; \tilde{\xi}_i)$, so the lag error can be unbounded when $r$ is large.

To address the issue mentioned above, we analyze the convergence of EPISODE++ in a high-probability framework. Let $\boldsymbol{y}_{r,k}^i$ denote the local model of client $i$ at step $k$ during the most recent round in which client $i$ participated (before round $r$), and let $q_r^i$ denote the index of this round. Since client subsampling independent across rounds, $\mathbb{P}(q_r^i - r \geq K) = (1 - S/N)^K$. Therefore, we introduce the event $\mathcal{E} := \left\{ \max_{0 \leq r \leq R-1, 1 \leq i \leq N} r - q_r^i \leq K \right\}$, where $K$ is a logarithmic factor such that $\mathbb{P}(\mathcal{E}) \geq 1 - \delta$. Under $\mathcal{E}$, we obtain the following lemma to bound the distance of local updates.

**Lemma 1.** *Suppose that* (9) *holds. Then for any $r \geq 0$ we have*

$$\mathbb{1}_{\mathcal{E}} \cdot \max_{k \in [I]} \|\boldsymbol{x}_{r,k}^i - \bar{\boldsymbol{x}}_r\| \leq 2I(2\sigma\eta + \gamma), \quad \text{for any } i \in \mathcal{S}_r, \quad (4)$$

$$\mathbb{1}_{\mathcal{E}} \cdot \max_{k \in [I]} \|\boldsymbol{y}_{r,k}^i - \bar{\boldsymbol{x}}_r\| \leq 2(K+1)I(2\sigma\eta + \gamma), \quad \text{for any } i \in [N]. \quad (5)$$

The proof of Lemma 1 is deferred to Appendix A.3, which relies on a *jump start* analysis. If $\|\boldsymbol{G}_r\| > \gamma/\eta$, then (4) trivially holds due to the clipping operation. When $\|\boldsymbol{G}_r\| \leq \gamma/\eta$, a recursive argument shows that the discrepancy $\|\boldsymbol{x}_{r,k}^i - \bar{\boldsymbol{x}}_r\|$ depends on the magnitude of the increment at the starting point, that is the lag error $\|\mathbb{E}[\boldsymbol{G}_r^i] - \nabla f_i(\bar{\boldsymbol{x}}_r)\|$ and $\|\boldsymbol{G}_r\|$ after removing noise. According to the construction of $\boldsymbol{G}_r^i$, we know $\mathbb{E}[\boldsymbol{G}_r^i] = \frac{1}{I} \sum_{k=1}^I \nabla f_i(\boldsymbol{y}_{r,k}^i) = \frac{1}{I} \sum_{k=1}^I \nabla f_i(\boldsymbol{x}_{q_r^i,k}^i)$. Therefore, under the event $\mathcal{E}$, the lag error at the $r$-th round can be bounded by the discrepancies of previous $K + 1$ rounds at most, which is the bound (5). In summary, the discrepancy at any round can be controlled recursively if the discrepancy at the initial round is small. This insight motivates the initialization $\boldsymbol{G}_0^i = \nabla F_i(\bar{\boldsymbol{x}}_0, \tilde{\xi}_i)$ in Algorithm 1, which enjoys zero initial lag error.

Finally, Lemma 1 shows that the condition of (3) can be satisfied by choosing $\eta, \gamma, I$, which establishes descent of the global objective from $\bar{\boldsymbol{x}}_r$ to $\bar{\boldsymbol{x}}_{r+1}$. Summing from $r = 0, \ldots, R - 1$ and applying concentration bounds over martingale difference sequences to yield high probability bounds for error terms coming from stochastic gradient noise and client subsampling yields

$$f(\bar{\boldsymbol{x}}_R) - f(\bar{\boldsymbol{x}}_0) \leq \sum_{r=0}^{R-1} [\mathbb{1}_{\bar{\mathcal{A}}_r} U(\bar{\boldsymbol{x}}_r) + \mathbb{1}_{\mathcal{A}_r} V(\bar{\boldsymbol{x}}_r)] + (12\sigma^2 + \Gamma_2 I) \eta \log \frac{1}{\delta}, \quad (6)$$

where $\mathcal{A}_r = \{\|\boldsymbol{G}_r\| \leq \gamma/\eta\}$ denotes the clipping indicator, and $U(x), V(x)$ are defined in the proof of Theorem 1. According to the choice of $\eta, \gamma$, the dominant term of $U(\bar{\boldsymbol{x}}_r)$ is $-\gamma I \|\nabla f(\bar{\boldsymbol{x}}_r)\|$ and that of $V(\bar{\boldsymbol{x}}_r)$ is $-\eta I \|\nabla f(\bar{\boldsymbol{x}}_r)\|^2$. Plugging into (6) and rearranging proves Theorem 1.

# 5 Lower Bound for Clipped Minibatch SGD

Clipped minibatch SGD is a natural extension of minibatch SGD [40, 45] to the relaxed smooth setting (see pseudocode in Algorithm 2, Appendix B). Clipped minibatch SGD is nearly identical to minibatch SGD: with the addition of gradient clipping to each round's update. In the similar spirit of [45], we are interested in this algorithm because it has the same computation and communication structure as EPISODE++ and it is important to understand whether EPISODE++ has any advantage over clipped minibatch SGD. In fact, we will show that clipped minibatch SGD is significantly hindered by the combination of relaxed smoothness, client heterogeneity, and client subsampling.

**Assumption 2.** *There exists $M > 0$ such that $\|\nabla f(\boldsymbol{x})\| \le M$ for all $\boldsymbol{x}$ with $f(\boldsymbol{x}) \le f(\boldsymbol{x}_0)$.*

A line of work on relaxed smoothness in the single-machine setting [52, 8] has shown that the number of iterations required to find an $\epsilon$-stationary point by gradient descent (GD) under relaxed smoothness is $\tilde{\Omega}\left(\frac{\Delta L_1 M}{\epsilon^2}\right)$, while that of GD with gradient clipping is $O\left(\frac{\Delta L_0}{\epsilon^2}\right)$. In this way, gradient clipping can remove the dependence on $M$, which can be large, and significantly speed up optimization.

Theorem 2 shows a surprising result: under some conditions on the participation ratio, number of clients, and heterogeneity parameter $\rho$, clipped minibatch SGD requires $\tilde{\Omega}\left(\frac{\Delta L_1 M}{\epsilon^2}\right)$ rounds, showing that applying gradient clipping to minibatch SGD *does not eliminate dependence on $M$*.

**Theorem 2.** *Fix $\epsilon > 0$, $0 < \delta < 1$, $L_0 > 0$, $L_1 > 0$, $\kappa > 0$, $\rho > \frac{2 + \log(2-\delta)}{\log(2-\delta)}$, $M > \max(\frac{L_0}{L_1}, \epsilon)$, $N \ge \frac{(\rho+1)(1+\log(2-\delta))}{(\rho-1)\log(2-\delta)-2}$. Define $Q = \left\lfloor \frac{\kappa+(\rho-1)M}{\kappa+(\rho+1)M} N \right\rfloor$. Let $\mathcal{F}(L_0, L_1, M, \kappa, \rho, N)$ denote the set of problem instances $\{f_i\}_{i=1}^N$ satisfying Assumptions 1(ii)-(iv) and 2 with $\sigma = 0$. For any fixed choice of parameters $\gamma, \eta$ based on the knowledge of above constants, there exists $\{f_i\}_{i=1}^N \in \mathcal{F}(L_0, L_1, M, \kappa, \rho, N)$ such that clipped minibatch SGD initialized at $\boldsymbol{x}_0$ with $1 \le S \le \frac{\log(2-\delta)(Q+1)}{N-Q+\log(2-\delta)}$ will satisfy $\mathbb{P}\left(\|\nabla f(\boldsymbol{x}_r)\| < \epsilon \text{ for some } 0 \le r \le R-1\right) > 1 - \delta$ only if*

$$R \ge \frac{L_1 M \left(f(\boldsymbol{x}_0) - f^* - \frac{15\epsilon^2}{16L_0}\right)}{2\epsilon^2 \left(1 + \log \frac{L_1 M}{L_0}\right)}.$$

The proof is included in Appendix B. This result shows that clipped minibatch SGD in our setting suffers the same problem as GD in the single-machine setting: divergence can only be avoided with a very small step size, leading to slow convergence. In contrast, the convergence rate of EPISODE++ in the same setting is independent of $M$.

The proof of Theorem 2 analyzes clipped minibatch SGD for three different problem instances. The first contains linear local objectives with high heterogeneity: if the clipping threshold is sufficiently small ($\frac{\gamma}{\eta} \le M$), then clipped minibatch SGD will never converge with probability $\delta$. The second instance contains homogeneous, exponential local objectives: the learning rate must be sufficiently small $\eta < O\left(\frac{1}{L_1 M}\right)$ to avoid divergence due to the exponentially increasing gradient magnitude. However, with a large clipping threshold and small learning rate, the convergence of clipped minibatch SGD will depend on $M$ for the third problem instance, which has homogeneous linear objectives. Note that our lower bound is different from previous lower bounds in [52, 51] which are in the settings of single machine [52] or almost sure bounded noise [51], since our lower bound is considering noise from client subsampling and client data heterogeneity which is not almost surely bounded.

# 6 Experiments

To validate our theory, we evaluate EPISODE++ and baselines in the training of RNNs for two text classification tasks. We compare EPISODE++ to CELGC [32], clipped minibatch SGD, and NaiveParallelClip [32], which is a naive parallel implementation of SGD with gradient clipping that requires communication at every iteration. As an ablation study, we also evaluate two algorithms closely related to EPISODE++. The first is a naive extension of EPISODE [9] for client subsampling, where each participating client's control variate $\boldsymbol{G}_r^i$ is resampled at the beginning of each round, and $\boldsymbol{G}_r = \frac{1}{S} \sum_{i \in \mathcal{S}_r} \boldsymbol{G}_r^i$. The second is an algorithm which we refer to as SCAFFOLDClip [9], which applies gradient clipping to each local step of SCAFFOLD [24]. We evaluate these six

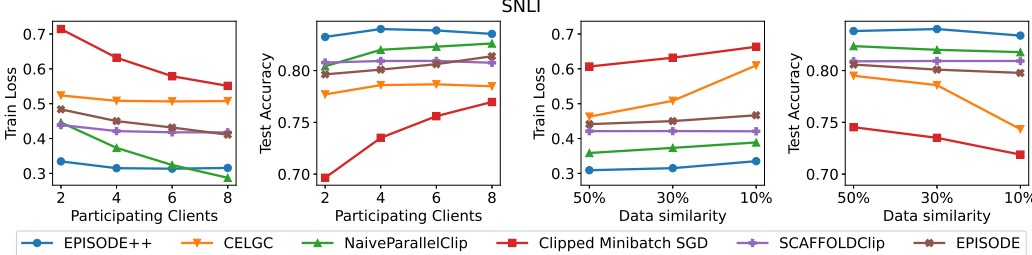

(a) Training loss and testing accuracy for SNLI dataset.

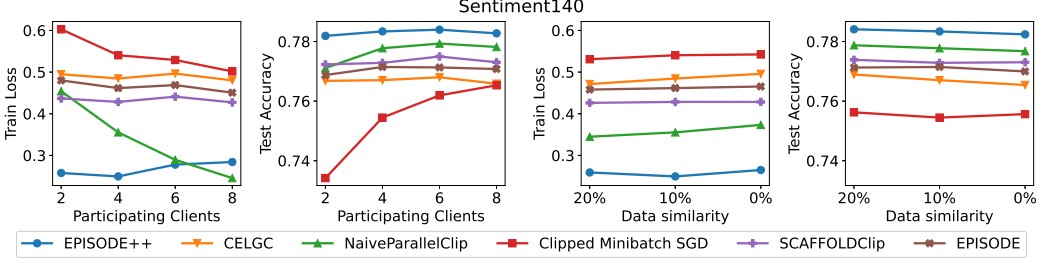

(b) Training loss and testing accuracy for Sentiment140 dataset.

Figure 1: Final training loss and testing accuracy for all algorithms, as participation ratio and data similarity varies. (a) and (b) show results for SNLI and Sentiment140, respectively.

algorithms on natural language inference with the SNLI dataset [4] and sentiment classification with the Sentiment140 dataset [5]. More experimental results can be found in Appendix C.

## 6.1 Setup

All experiments use uniform client sampling, a batch size of 64 (on each client) and the multi-class hinge loss. See Appendix C.1 for full details on hyperparameters. All experiments were implemented in PyTorch and ran on eight NVIDIA V100 GPUs.

**SNLI** SNLI [4] is a 3-way text classification task, in which the logical relationship of a pair of sentences must be classified as either entailment, neutral, or contradictory. The dataset contains 570k pairs of sentences. Because SNLI is a centralized dataset, we follow the heterogeneity protocol in [24] to divide the dataset into clients according to a similarity parameter $s$ between $0\%$ and $100\%$. According to this protocol, $s\%$ of each client's local dataset is allocated from a randomly shuffled set of examples, while the remaining $(100 - s)\%$ is allocated from a set of examples which is sorted by label. In this way, the similarity of the label distributions of client datasets grows with $s$. The network consists of a one-layer bidirectional RNN encoder followed by a three-layer fully connected classifier. We train for $R = 5375$ communication rounds with $I = 4$ for all algorithms except NaiveParallelClip, which uses $R = 21500$ and $I = 1$, so that every algorithm runs the same number of training steps.

**Sentiment140** Sentiment140 [5] is a sentiment prediction task designed for FL. The dataset is comprised of tweets, each labeled as either positive or negative. We follow the data processing steps of [28] to discard users with small datasets and to split into training and testing sets. In order to control the data heterogeneity between clients, we follow a similar protocol as described for SNLI to form client datasets by combining the datasets of original users in the Sentiment140 dataset. The process is nearly identical to that of SNLI, but here we allocate *users* to each local dataset instead of *examples*. $s\%$ of each client's local dataset is allocated from a randomly shuffled set of *users*, while the remaining $(100 - s)\%$ is allocated from a set of users sorted by the proportion of positive samples. We train for $R = 2000$ communication rounds with $I = 4$ for all algorithms except NaiveParallelClip, which uses $R = 8000$ and $I = 1$. We use the same network architecture as SNLI.

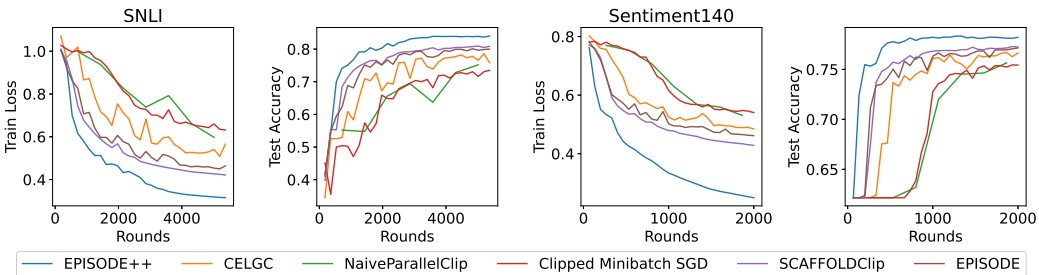

Figure 2: Learning curves for SNLI and Sentiment140 under the setting $S = 4, s = 30\%$ (SNLI) and $S = 4, s = 10\%$ (Sentiment140). For NaiveParallelClip, we show the first 5375 (SNLI) and 2000 (Sentiment140) rounds to compare all algorithms with a fixed number of communication rounds.

| Method | SNLI | | Sent140 | |
| | Train Loss | Test Accuracy | Train Loss | Test Accuracy |
|---|---|---|---|---|
| EPISODE++ | $\mathbf{0.317 \pm 0.001}$ | $\mathbf{83.89 \pm 0.07}$ | $\mathbf{0.258 \pm 0.004}$ | $\mathbf{77.97 \pm 0.16}$ |
| CELGC | $0.529 \pm 0.005$ | $77.45 \pm 0.37$ | $0.493 \pm 0.005$ | $76.36 \pm 0.18$ |
| NaiveParallelClip | $0.377 \pm 0.001$ | $81.74 \pm 0.09$ | $0.365 \pm 0.004$ | $77.73 \pm 0.07$ |
| Clipped MinibatchSGD | $0.642 \pm 0.008$ | $72.74 \pm 0.20$ | $0.549 \pm 0.004$ | $75.10 \pm 0.27$ |
| SCAFFOLDClip | $0.424 \pm 0.002$ | $81.11 \pm 0.06$ | $0.431 \pm 0.001$ | $77.38 \pm 0.16$ |
| EPISODE | $0.455 \pm 0.005$ | $80.24 \pm 0.13$ | $0.466 \pm 0.003$ | $76.89 \pm 0.17$ |

Table 2: Average results for three trials under $S = 4, s = 30\%$ (SNLI) or $S = 4, s = 10\%$ (Sent140). The error is the distance from the average to the max/min across three runs.

We study the effect of client subsampling and data heterogeneity by varying each of these values in a controlled way. We first fix the heterogeneity $s = 30\%$ and vary $S \in \{2, 4, 6, 8\}$, then we fix the number of participating clients $S = 4$ and vary $s \in \{10\%, 30\%, 50\%\}$. It should be noted that in the experiments with varying $S$, we always train for a fixed number of iterations $RI$. This means that separate training runs with different $S$ will have the same per-client computation cost (number of gradient computations), but the total computation cost of a training run scales with $S$. For this setting, we use $N = 8$ clients.

To simulate large-scale federated learning, we also include results with a larger number of clients $N = 128$. For this setting, we use $S = 16$ participating clients in each round, and data heterogeneity of $s = 30\%$ (SNLI) and $s = 10\%$ (Sentiment140).

## 6.2 Results

Figure 1 contains the final training loss and testing accuracy for the variety of settings of client participation and data heterogeneity. For a single setting of participation and heterogeneity, Figure 2 and Table 2 show learning curves and average results over three trials, respectively. More learning curves are given in Appendix C.2. Learning curves for large-scale experiments are shown in Figure 3. EPISODE++ achieves the minimum training loss and maximum testing accuracy of all algorithms in nearly every setting. Only with full participation $S = 8$ does NaiveParallelClip achieve a lower training loss than EPISODE++, but EPISODE++ maintains a higher testing accuracy. Also, NaiveParallelClip requires a much larger communication cost than EPISODE++ to perform the same number of training iterations: when the number of communication rounds is fixed, EPISODE++ significantly outperforms NaiveParallelClip.

**Effect of Subsampling** The first two plots of Figures 1(a) and 1(b) show the performance of each algorithm as the number of participating clients $S$ varies over $\{2, 4, 6, 8\}$ with fixed data similarity $s = 30\%$. Clipped minibatch SGD, NaiveParallelClip, and EPISODE all exhibit degraded performance as $S$ decreases, whereas EPISODE++, SCAFFOLDClip, and CELGC maintain relatively constant performance as $S$ decreases. Despite the constant performance of CELGC and SCAFFOLDClip under client sampling, both algorithms are significantly outperformed by EPISODE++.

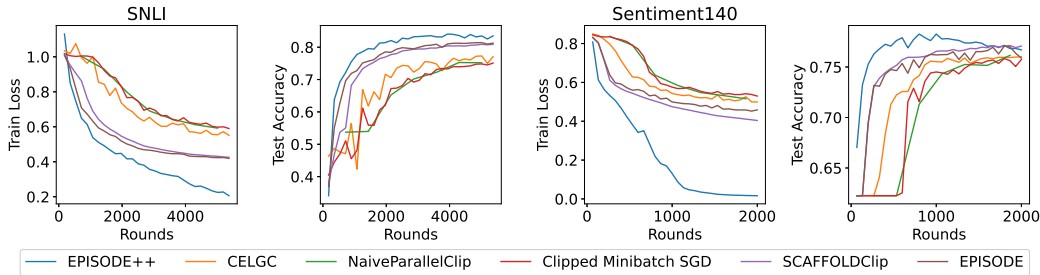

Figure 3: Learning curves for large-scale training with $N = 128, S = 16$, and $s = 30\%$ (SNLI) or $s = 10\%$ (Sentiment140). We compare all algorithms with a fixed number of communication rounds.

**Effect of Heterogeneity**    The last two plots of each row in Figure 1 show each algorithm's performance as the data similarity $s$ varies over $\{50\%, 30\%, 10\%\}$ for SNLI and $\{20\%, 10\%, 0\%\}$ for Sentiment140, with fixed $S = 4$. For both datasets, decreasing data similarity negatively impacts the performance of clipped minibatch SGD, CELGC, and NaiveParallelClip. EPISODE++, SCAF-FOLDClip, and EPISODE are able to maintain performance as data similarity decreases, though EPISODE++ maintains a significantly better performance than SCAFFOLDClip and EPISODE.

**Large-Scale Experiments**    With a larger number of clients ($N = 128, S = 16$), the relative performance of each algorithm is similar to the $N = 8$ setting, as shown in Figure 3. Here, the proportion of participating clients $S/N = 1/8$ is smaller than that of the $N = 8$ experiments, where $S/N \geq 1/4$. This suggests that the effect of partial client participation may be stronger in the large-scale experiments. EPISODE++ still outperforms all other algorithms in the large-scale setting, while maintaining about the same test accuracy as the $N = 8$ setting, demonstrating the effectiveness of EPISODE++ for large-scale federated learning.

**Comparison with Ablations**    By comparing EPISODE++ against the closely related EPISODE and SCAFFOLDClip, we can see that the use of information from previous rounds and episodic gradient clipping are both critical for the superior performance of EPISODE++. EPISODE only utilizes information from the currently participating clients, ignoring information from clients that participated in previous rounds. As a result, the performance of EPISODE degrades as $S$ decreases. On the other hand, SCAFFOLDClip determines whether to perform clipping individually for each local step, as opposed to the episodic gradient clipping of EPISODE++ and EPISODE. Although SCAFFOLDClip maintains performance under changes in the participation ratio and data similarity, the level it maintains is significantly lower than that of EPISODE++.

**Communication Cost**    NaiveParallelClip suffers a large communication cost for the same number of training iterations compared with other algorithms, due to the cost of synchronizing clients at every iteration. As shown in Figure 2 and Figure 3, EPISODE++ outperforms all other algorithms by a wide margin when the number of communication rounds is fixed. Also, EPISODE requires twice the number of communication operations per training round, which doubles the time required for communication per round compared to all other algorithms.

## 7  Conclusion

We have presented EPISODE++, the first algorithm for FL with heterogeneous data and client subsampling under relaxed smoothness. We proved that EPISODE++ finds an $\epsilon$-stationary point with high probability, and its convergence rate satisfies linear speedup and resilience to heterogeneity while enjoying reduced communication. We also presented a lower bound showing that the convergence rate of a special case of clipped minibatch SGD in our setting suffers a dependence on $M$ (the maximum gradient norm of the objective in a sublevel set), implying that applying gradient clipping to minibatch SGD does not alleviate the problem of exploding gradients. Our experimental results for RNN training on text classification tasks demonstrate the superior performance of EPISODE++ compared to baselines. One limitation of our current work is that our lower bound assumes $\sigma = 0$, and we plan to get a better lower bound in the future for $\sigma > 0$.

## Acknowledgments and Disclosure of Funding

We would like to thank the anonymous reviewers for their helpful comments. Michael Crawshaw is supported by the Institute for Digital Innovation fellowship from George Mason University. Michael Crawshaw and Mingrui Liu are both supported by a grant from George Mason University. Computations were run on ARGO, a research computing cluster provided by the Office of Research Computing at George Mason University (URL: https://orc.gmu.edu). The work of Yajie Bao was done when he was virtually visiting Mingrui Liu's research group in the Department of Computer Science at George Mason University.

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

# A Deferred Proofs of the Upper Bound

## A.1 Notation and Preliminaries

Let $q_0^i = 0$, $\boldsymbol{y}_{0,k}^i = \bar{\boldsymbol{x}}_0$, and define

$$q_r^i := \begin{cases} r - 1 & i \in \mathcal{S}_{r-1} \\ q_{r-1}^i & \text{otherwise} \end{cases} \tag{7}$$

and

$$\boldsymbol{y}_{r,k}^i := \begin{cases} \boldsymbol{x}_{r-1,k}^i & i \in \mathcal{S}_{r-1} \\ \boldsymbol{y}_{r-1,k}^i & \text{otherwise} \end{cases} \tag{8}$$

The intermediate variable $q_r^i$ is the most recent round before (not including) round $r$ in which client $i$ participated. Similarly, $\boldsymbol{y}_{r,k}^i$ is the local model at step $k$ during the most recent round before (not including) round $r$ in which client $i$ participated. Then we know that $\boldsymbol{y}_{r,k}^i = \boldsymbol{x}_{q_r^i,k}^i$. Therefore $\boldsymbol{G}_r^i = \frac{1}{I} \sum_{k=0}^{I-1} \nabla F_i(\boldsymbol{y}_{r,k}^i; \xi_{q_r^i,k}^i)$. Also, we will initialize $\boldsymbol{G}_0^i = \nabla F_i(\bar{\boldsymbol{x}}_0^i; \tilde{\xi}^i)$ for all clients.

Denote by $\mathcal{A}_r = \{\|\boldsymbol{G}_r\| \le \frac{\gamma}{\eta}\}$. Our analysis is simplified by taking the following conditions throughout the proof:

$$(K+1)\eta I \left( AL_0 + BL_1\kappa + 4BL_1\rho \left( 2\sigma + \frac{\gamma}{\eta} \right) \right) \le \frac{1}{60}$$

$$2(K+1)I (2\sigma\eta + \gamma) \le \frac{C}{L_1}, \tag{9}$$

where $C > 0$ is a constant, $A$ and $B$ are defined in terms of $C$ (see Lemma 2), and $K$ is defined in Lemma 8. Usage of these conditions will be explicitly stated when used.

Through out this section, we denote the filtration

$$\mathcal{F}_r = \sigma \left( \{\tilde{\xi}_i\}_{i \in [N]}, \{\xi_{q,k}^i : q \le r, k \in [I]\}_{i \in [N]} \right).$$

We use $\mathbb{E}_r[\cdot] = \mathbb{E}[\cdot \mid \mathcal{F}_r]$ and $\mathbb{P}_r(\cdot)$ to denote the conditional expectation and probability given the filtration $\mathcal{F}_r$. Notice that, given $\mathcal{F}_r$, the global weight $\bar{\boldsymbol{x}}_r$ is fixed but the subsampling set $\mathcal{S}_r$ is still random and is independent of $\mathcal{F}_r$.

## A.2 Auxiliary Lemmas

**Lemma 2** (Corollary A.4 in [51]). *Let $f : \mathbb{R}^d \to \mathbb{R}$ be $(L_0, L_1)$-smooth and $C > 0$. For any $\boldsymbol{x}, \boldsymbol{y} \in \mathbb{R}^d$ with $\|\boldsymbol{x} - \boldsymbol{y}\| \le \frac{C}{L_1}$,*

$$\|\nabla f(\boldsymbol{x}) - \nabla f(\boldsymbol{y})\| \le (AL_0 + BL_1\|\nabla f(\boldsymbol{x})\|)\|\boldsymbol{x} - \boldsymbol{y}\|,$$

*where $A = 1 + e^C - \frac{e^C - 1}{C}$ and $B = \frac{e^C - 1}{C}$.*

**Lemma 3** (Lemma A.3 in [51]). *Let $f : \mathbb{R}^d \to \mathbb{R}$ be $(L_0, L_1)$-smooth and $C > 0$. For any $\boldsymbol{x}, \boldsymbol{y} \in \mathbb{R}^d$ with $\|\boldsymbol{x} - \boldsymbol{y}\| \le \frac{C}{L_1}$,*

$$f(\boldsymbol{y}) \le f(\boldsymbol{x}) + \langle \nabla f(\boldsymbol{x}), \boldsymbol{y} - \boldsymbol{x} \rangle + \frac{AL_0 + BL_1\|\nabla f(\boldsymbol{x})\|}{2}\|\boldsymbol{x} - \boldsymbol{y}\|^2,$$

*where $A = 1 + e^C - \frac{e^C - 1}{C}$ and $B = \frac{e^C - 1}{C}$.*

**Lemma 4** (Lemma B.1 in [51]). *For any $\mu \ge 0$ and $\boldsymbol{x}, \boldsymbol{y} \in \mathbb{R}^d$,*

$$-\left\langle \boldsymbol{x}, \frac{\boldsymbol{y}}{\|\boldsymbol{y}\|} \right\rangle \le -\mu\|\boldsymbol{x}\| - (1 - \mu)\|\boldsymbol{y}\| + (1 + \mu)\|\boldsymbol{y} - \boldsymbol{x}\|.$$

**Lemma 5** (Lemma 1 in [30]). *Assume that $Z_1, \ldots, Z_T$ is a martingale difference sequence with respect to the filtration $\mathcal{F}_t$ and $\mathbb{E}_t[\exp(Z_t^2/\sigma_t^2)] \le \exp(1)$ for all $t$, where $\sigma_1, \ldots, \sigma_T$ is a sequence of random variables such that $\sigma_t \in \mathcal{F}_t$. Then for any fixed $\lambda > 0$ and $\delta \in (0, 1)$, with probability at least $1 - \delta$,*

$$\sum_{t=1}^{T} Z_t \le \frac{3}{4}\lambda \sum_{t=1}^{T} \sigma_t^2 + \frac{1}{\lambda} \log \frac{1}{\delta}.$$

**Lemma 6** (Improved Serfling's inequality, Proposition 2.3 in [3]). *Let $\mathcal{X} = (x_1, \ldots, x_N)$ be a finite population of $N$ real numbers and $X_1, \ldots, X_n$ denote a random sample without replacement from $\mathcal{X}$. Let $\mu_\mathcal{X} = \frac{1}{N} \sum_{i=1}^N x_i$, $\sigma_\mathcal{X}^2 = \frac{1}{N} \sum_{i=1}^N (x_i - \mu)^2$, and $\hat{\mu}_n = \frac{1}{n} \sum_{i=1}^n X_i$. If $\max_{i \in [N]} |x_i| \leq b$, then for any $\lambda > 0$,*

$$\mathbb{E}[\exp(\lambda(\hat{\mu}_n - \mu_\mathcal{X}))] \leq \exp\left(\frac{b^2\lambda^2}{2} \frac{n+1}{n^2} \left(1 - \frac{n}{N}\right)\right).$$

**Lemma 7** (Lemma 12 in [10]). *Suppose $X_1, \ldots, X_T$ is a martingale difference sequence in a Hilbert space such that $\|X_t\| \leq b$ almost surely for some constant b. Further, assume $\mathbb{E}_{t-1}[\|X_t\|^2] \leq \sigma_t^2$ with probability 1 for some constants $\sigma_t$. Then with probability at least $1 - 3\delta$, for all $k$:*

$$\left\|\sum_{t=1}^k X_t\right\| \leq 3b \max\left(1, \log\frac{1}{\delta}\right) + 3\sqrt{\sum_{t=1}^k \sigma_t^2 \max\left(1, \log\frac{1}{\delta}\right)}$$

### A.3 Proof of Lemma 1

Define

$$\Xi_r := \frac{1}{NI} \sum_{i=1}^N \sum_{k=0}^{I-1} \|\bar{\boldsymbol{x}}_r - \boldsymbol{y}_{r,k}^i\|.$$

$\Xi_r$ is the average "lag" (over clients) of the correction $\boldsymbol{G}_r^i$ behind $\nabla f(\bar{\boldsymbol{x}}_r)$ due to client sampling, since each $\boldsymbol{G}_r^i$ is set according to stochastic gradients at $\boldsymbol{y}_{r,k}^i$.

For any $\delta \in (0, 1)$, define

$$K = \begin{cases} \left\lceil \frac{\log\left(\frac{RN}{\delta}\right)}{\log\left(\frac{N}{N-S}\right)} \right\rceil & S < N \\ 1 & S = N \end{cases}$$

and

$$\mathcal{E} = \left\{ \max_{0 \leq r \leq R-1, 1 \leq i \leq N} r - q_r^i \leq K \right\}. \tag{10}$$

The event $\mathcal{E}$ occurs when every round in which a client is sampled occurs after no more than $K$ rounds following the previous round in which that client was sampled. This means that under $\mathcal{E}$, the correction $\boldsymbol{G}_r^i$ was computed within $K$ rounds of round $r$. The next lemma shows that $\mathcal{E}$ occurs with probability $1 - \delta$.

**Lemma 8.** *For any $\delta$ with $0 < \delta < 1$, $\mathbb{P}(\mathcal{E}) \geq 1 - \delta$.*

*Proof.* If $S = N$, then $r - q_r^i = 1$, and we are done. Otherwise, since each round's participating clients are sampled uniformly without replacement and independently at each round,

$$\begin{aligned}
\mathbb{P}(r - q_r^i \leq k) &= \sum_{i=1}^k \mathbb{P}(r - q_r^i = i) \\
&= \begin{cases} \sum_{i=1}^k \frac{S}{N}\left(1 - \frac{S}{N}\right)^{i-1} & k < r \\ 1 & \text{otherwise} \end{cases} \\
&\geq \frac{S}{N} \sum_{i=1}^k \left(1 - \frac{S}{N}\right)^{i-1} \\
&= \frac{S}{N} \frac{1 - \left(1 - \frac{S}{N}\right)^k}{1 - \left(1 - \frac{S}{N}\right)} \\
&= 1 - \left(1 - \frac{S}{N}\right)^k.
\end{aligned}$$

Also by the choice of $K$: $K \geq \frac{\log\left(\frac{RN}{\delta}\right)}{\log\left(\frac{N}{N-S}\right)}$ , we can show that $\left(1 - \frac{S}{N}\right)^K \leq \frac{\delta}{RN}$. So $\mathbb{P}(r - q_r^i \leq K) \geq 1 - \frac{\delta}{RN}$. Therefore

$$\mathbb{P}(\mathcal{E}) = \mathbb{P}\left(r - q_r^i \leq K, \forall 0 \leq r \leq R - 1, 1 \leq i \leq N\right) \geq 1 - \sum_{r=0}^{R-1} \sum_{i=1}^{N} \mathbb{P}(r - q_r^i > K) \leq 1 - \delta.$$

$\square$

**Lemma 9.** *Let $r \geq 0$. If $\|\bar{x}_r - y_{r,k}^i\| \leq \frac{C}{L_1}$ for all $i \in [N]$ and $k \in \{0, \ldots, I-1\}$, and $BL_1\rho\Xi_r < 1$, then*

$$\|\nabla f(\bar{x}_r)\| \leq \frac{\sigma + \|G_r\| + (AL_0 + BL_1\kappa)\Xi_r}{1 - BL_1\rho\Xi_r}.$$

*Proof.* Due to the assumption $\|\bar{x}_r - y_{r,k}^i\| \leq \frac{C}{L_1}$, we have

$$\|\nabla f(\bar{x}_r)\| \leq \|\nabla f(\bar{x}_r) - G_r\| + \|G_r\|$$

$$\leq \frac{1}{N} \sum_{i=1}^{N} \|\nabla f_i(\bar{x}_r) - G_r^i\| + \|G_r\|$$

$$\leq \frac{1}{NI} \sum_{i=1}^{N} \sum_{k=0}^{I-1} \|\nabla f_i(\bar{x}_r) - \nabla F_i(y_{r,k}^i; \xi_{q_r,k}^i)\| + \|G_r\|$$

$$\leq \frac{1}{NI} \sum_{i=1}^{N} \sum_{k=0}^{I-1} \|\nabla f_i(\bar{x}_r) - \nabla f_i(y_{r,k}^i)\| + \frac{1}{NI} \sum_{i=1}^{N} \sum_{k=0}^{I-1} \|\nabla f_i(y_{r,k}^i) - \nabla F_i(y_{r,k}^i; \xi_{q_r,k}^i)\| + \|G_r\|$$

$$\overset{(i)}{\leq} \frac{1}{NI} \sum_{i=1}^{N} \sum_{k=0}^{I-1} (AL_0 + BL_1\|\nabla f_i(\bar{x}_r)\|)\|\bar{x}_r - y_{r,k}^i\| + \sigma + \|G_r\|$$

$$\overset{(ii)}{\leq} (AL_0 + BL_1(\kappa + \rho\|\nabla f(\bar{x}_r)\|)) \frac{1}{NI} \sum_{i=1}^{N} \sum_{k=0}^{I-1} \|\bar{x}_r - y_{r,k}^i\| + \sigma + \|G_r\|$$

$$\leq (AL_0 + BL_1(\kappa + \rho\|\nabla f(\bar{x}_r)\|))\Xi_r + \sigma + \|G_r\|,$$

where we used Lemma 2 in $(i)$ and Assumption 1(iv) in $(ii)$. Rearranging to isolate $\|\nabla f(\bar{x}_r)\|$ gives the result. $\square$

**Lemma 10.** *Suppose that Equation 9 holds, then for any $i \in [N]$ and $k \in [I]$,*

$$\|x_{0,k}^i - \bar{x}_0\| \leq 2I(2\sigma\eta + \gamma). \tag{11}$$

*Proof.* Notice that, due to the clipping operation,

$$\mathbb{1}_{\bar{A}_0} \|x_{0,k}^i - \bar{x}_0\| = k\gamma \leq 2k(2\sigma\eta + \gamma). \tag{12}$$

holds for any $k \leq I$. Next we verify Equation 11 under the event $\mathcal{A}_0$. To see this, we first consider the case $k = 1$,

$$\mathbb{1}_{\mathcal{A}_0} \|x_{0,1}^i - \bar{x}_0\| = \mathbb{1}_{\mathcal{A}_0} \cdot \eta \|\nabla F_i(\bar{x}_0; \xi_{0,0}^i) - G_0^i + G_0\|$$

$$\leq \mathbb{1}_{\mathcal{A}_0} (\eta\|\nabla F_i(\bar{x}_0; \xi_{0,0}^i) - \nabla f_i(\bar{x}_0)\| + \eta\|G_0^i - \nabla f_i(\bar{x}_0)\| + \eta\|G_0\|)$$

$$\leq 2\sigma\eta + \gamma,$$

where we used the initialization $G_0^i = \nabla F_i(\bar{x}_0; \tilde{\xi}^i)$ and $\|G_0\| \leq \frac{\gamma}{\eta}$ under the event $\mathcal{A}_0$. This indicates that Equation 11 holds with $k = 1$.

Now suppose that Equation 11 holds for some $k$ with $1 \le k < I$. Then we have

$$
\begin{aligned}
\mathbb{1}_{\mathcal{A}_0}\|\boldsymbol{x}_{0,k+1}^i - \bar{\boldsymbol{x}}_0\| &= \mathbb{1}_{\mathcal{A}_0}\|\boldsymbol{x}_{0,k}^i - \eta(\nabla F_i(\boldsymbol{x}_{0,k}^i; \xi_{0,k}^i) - \boldsymbol{G}_0^i + \boldsymbol{G}_0) - \bar{\boldsymbol{x}}_0\| \\
&\le \mathbb{1}_{\mathcal{A}_0}\left(\|\boldsymbol{x}_{0,k}^i - \bar{\boldsymbol{x}}_0\| + \eta\|\nabla F_i(\boldsymbol{x}_{0,k}^i; \xi_{0,k}^i) - \boldsymbol{G}_0^i + \boldsymbol{G}_0\|\right) \\
&\overset{(i)}{\le} \mathbb{1}_{\mathcal{A}_0}\Big(\|\boldsymbol{x}_{0,k}^i - \bar{\boldsymbol{x}}_0\| + \eta\|\nabla F_i(\boldsymbol{x}_{0,k}^i; \xi_{0,k}^i) - \nabla f_i(\boldsymbol{x}_{0,k}^i)\| + \eta\|\boldsymbol{G}_0^i - \nabla f_i(\bar{\boldsymbol{x}}_0)\| \\
&\qquad + \eta\|\nabla f_i(\boldsymbol{x}_{0,k}^i) - \nabla f_i(\bar{\boldsymbol{x}}_0)\| + \eta\|\boldsymbol{G}_0\|\Big) \\
&\overset{(ii)}{\le} \mathbb{1}_{\mathcal{A}_0}\left(\|\boldsymbol{x}_{0,k}^i - \bar{\boldsymbol{x}}_0\| + 2\sigma\eta + \gamma + \eta\|\nabla f_i(\boldsymbol{x}_{0,k}^i) - \nabla f_i(\bar{\boldsymbol{x}}_0)\|\right) \\
&\overset{(iii)}{\le} \mathbb{1}_{\mathcal{A}_0}\left(\|\boldsymbol{x}_{0,k}^i - \bar{\boldsymbol{x}}_0\| + 2\sigma\eta + \gamma + \eta(AL_0 + BL_1\|\nabla f_i(\bar{\boldsymbol{x}}_0)\|)\|\boldsymbol{x}_{0,k}^i - \bar{\boldsymbol{x}}_0\|\right) \\
&\le \mathbb{1}_{\mathcal{A}_0}\left(\|\boldsymbol{x}_{0,k}^i - \bar{\boldsymbol{x}}_0\| + 2\sigma\eta + \gamma + \eta(AL_0 + BL_1(\kappa + \rho\|\nabla f(\bar{\boldsymbol{x}}_0)\|))\|\boldsymbol{x}_{0,k}^i - \bar{\boldsymbol{x}}_0\|\right).
\end{aligned}
$$

where $(i)$ holds due to the initialization $\boldsymbol{G}_0 = \nabla F_i(\bar{\boldsymbol{x}}_0; \tilde{\xi}^i)$; $(ii)$ follows from $\|\boldsymbol{G}_0\| \le \frac{\gamma}{\eta}$; and $(iii)$ follows from Lemma 2. Note that the conditions of Lemma 2 are satisfied here by the inductive hypothesis together with Equation 9.

Notice that $\Xi_0 = 0$, so by Lemma 9, we have

$$
\mathbb{1}_{\mathcal{A}_0}\|\nabla f(\bar{\boldsymbol{x}}_0)\| \le \|\boldsymbol{G}_0 - \nabla f(\bar{\boldsymbol{x}}_0)\| + \mathbb{1}_{\mathcal{A}_0}\|\boldsymbol{G}_0\| \le \frac{1}{N}\sum_{i=1}^{N}\left\|\nabla F_i(\bar{\boldsymbol{x}}_0; \tilde{\xi}^i) - \nabla f_i(\bar{\boldsymbol{x}}_0)\right\| + \frac{\gamma}{\eta} \le \sigma + \frac{\gamma}{\eta}.
$$

Therefore

$$
\begin{aligned}
\mathbb{1}_{\mathcal{A}_0}\|\boldsymbol{x}_{0,k+1}^i - \bar{\boldsymbol{x}}_0\| &\le \|\boldsymbol{x}_{0,k}^i - \bar{\boldsymbol{x}}_0\| + 2\sigma\eta + \gamma + \eta\left(AL_0 + BL_1\kappa + BL_1\rho\left(\sigma + \frac{\gamma}{\eta}\right)\right)\|\boldsymbol{x}_{0,k}^i - \bar{\boldsymbol{x}}_0\| \\
&\overset{(i)}{\le} \|\boldsymbol{x}_{0,k}^i - \bar{\boldsymbol{x}}_0\| + 2\sigma\eta + \gamma + \frac{1}{4I}\|\boldsymbol{x}_{0,k}^i - \bar{\boldsymbol{x}}_0\| \\
&\overset{(ii)}{\le} 2\sigma\eta + \gamma + \left(1 + \frac{1}{4I}\right)2k(2\sigma\eta + \gamma) \\
&= 2\left(k + \frac{k}{4I} + \frac{1}{2}\right)(2\sigma\eta + \gamma) \\
&\le 2(k+1)(2\sigma\eta + \gamma),
\end{aligned}
$$

where $(i)$ holds due to Equation 9 and $(ii)$ holds because of the inductive hypothesis that Equation 11 holds for $k$. This completes the induction over $k$ under the event $\mathcal{A}_0$. This fact together with Equation 12 proves that Equation 11 holds for all $k = 1, \ldots, I$. $\qquad\square$

**Lemma 11.** *Suppose that Equation 9 holds. If* $\max_{k\in[I]}\|\boldsymbol{x}_{q,k}^i - \bar{\boldsymbol{x}}_q\| \le 2I(2\sigma\eta + \gamma)$ *holds for any* $q \le r - 1$, *then we have*

$$
\mathbb{1}_{\mathcal{E}}\max_{k\in[I]}\|\boldsymbol{x}_{r,k}^i - \bar{\boldsymbol{x}}_r\| \le 2I(2\sigma\eta + \gamma), \quad \text{for any } i \in \mathcal{S}_r, \tag{13}
$$

$$
\mathbb{1}_{\mathcal{E}}\max_{k\in[I]}\|\boldsymbol{y}_{r,k}^i - \bar{\boldsymbol{x}}_r\| \le 2(K+1)I(2\sigma\eta + \gamma), \quad \text{for any } i \in [N],. \tag{14}
$$

*Proof.* Under $\bar{\mathcal{A}}_r$, Equation 13 trivially holds due to clipping. Next we verify Equation 13 under $\mathcal{A}_r$. First consider the case $k = 1$.

$$
\mathbb{1}_{\mathcal{A}_r} \| \boldsymbol{x}_{r,1}^i - \bar{\boldsymbol{x}}_r \|
$$

$$
= \mathbb{1}_{\mathcal{A}_r} \cdot \eta \| \nabla F_i(\bar{\boldsymbol{x}}_r; \xi_{r,0}^i) - \boldsymbol{G}_r^i + \boldsymbol{G}_r \|
$$

$$
\leq \eta \left\| \nabla F_i(\bar{\boldsymbol{x}}_r; \xi_{r,0}^i) - \frac{1}{I} \sum_{k=0}^{I-1} \nabla F_i(\boldsymbol{y}_{r,k}^i; \xi_{q_r^i,k}^i) \right\| + \eta \mathbb{1}_{\mathcal{A}_r} \| \boldsymbol{G}_r \|
$$

$$
\leq \eta \| \nabla F_i(\bar{\boldsymbol{x}}_r; \xi_{r,0}^i) - \nabla f_i(\bar{\boldsymbol{x}}_r) \| + \frac{\eta}{I} \sum_{k=0}^{I-1} \| \nabla F_i(\boldsymbol{y}_{r,k}^i) - \nabla F_i(\boldsymbol{y}_{r,k}^i; \xi_{q_r^i,k}^i) \| +
$$

$$
\frac{\eta}{I} \sum_{k=0}^{I-1} \| \nabla f_i(\bar{\boldsymbol{x}}_r) - \nabla f_i(\boldsymbol{y}_{r,k}^i) \| + \eta \mathbb{1}_{\mathcal{A}_r} \| \boldsymbol{G}_r \|
$$

$$
\leq 2\sigma\eta + \gamma + \frac{\eta}{I} \sum_{k=0}^{I-1} \| \nabla f_i(\bar{\boldsymbol{x}}_r) - \nabla f_i(\boldsymbol{y}_{r,k}^i) \|. \tag{15}
$$

Recall that $\boldsymbol{y}_{r,k}^i = \boldsymbol{x}_{q_r^i,k}^i$ from the definitions Equation 7 and Equation 8, so

$$
\| \bar{\boldsymbol{x}}_r - \boldsymbol{y}_{r,k}^i \| \leq \| \bar{\boldsymbol{x}}_r - \bar{\boldsymbol{x}}_{q_r^i} \| + \| \bar{\boldsymbol{x}}_{q_r^i} - \boldsymbol{x}_{q_r^i,k} \| \leq \sum_{q=q_r^i}^{r-1} \| \bar{\boldsymbol{x}}_{q+1} - \bar{\boldsymbol{x}}_q \| + \| \bar{\boldsymbol{x}}_{q_r^i} - \boldsymbol{x}_{q_r^i,k}^i \|
$$

$$
\overset{(i)}{\leq} (r - q_r^i) 2I \, (2\sigma\eta + \gamma) + 2k(2\sigma\eta + \gamma)
$$

$$
\overset{(ii)}{\leq} 2(K+1)I \, (2\sigma\eta + \gamma)
$$

$$
\overset{(iii)}{\leq} \frac{C}{L_1}, \tag{16}
$$

where $(i)$ holds due to inductive hypothesis for rounds $q \in \{0, \dots, r-1\}$; $(ii)$ holds due to the event $\mathcal{E}$ such that $r - q_r^i \leq K$; and $(iii)$ follows from Equation 9. The above bound on $\| \bar{\boldsymbol{x}}_r - \boldsymbol{y}_{r,k}^i \|$ verifies Equation 14 and shows that the condition of Lemma 2 is satisfied. Then applying it to Equation 15 yields

$$
\| \boldsymbol{x}_{r,1}^i - \bar{\boldsymbol{x}}_r \| \mathbb{1}_{\mathcal{A}_r} \leq 2\sigma\eta + \gamma + \eta \mathbb{1}_{\mathcal{A}_r} (AL_0 + BL_1 \| \nabla f_i(\bar{\boldsymbol{x}}_r) \|) \frac{1}{I} \sum_{k=0}^{I-1} \| \bar{\boldsymbol{x}}_r - \boldsymbol{y}_{r,k}^i \|
$$

$$
\leq 2\sigma\eta + \gamma + 2(K+1)\eta \mathbb{1}_{\mathcal{A}_r} I(AL_0 + BL_1\kappa + BL_1\rho \| \nabla f(\bar{\boldsymbol{x}}_r) \|)(2\sigma\eta + \gamma)
$$

$$
\leq (1 + 2(K+1)\eta I(AL_0 + BL_1\kappa + BL_1\rho \mathbb{1}_{\mathcal{A}_r} \| \nabla f(\bar{\boldsymbol{x}}_r) \|)) \, (2\sigma\eta + \gamma), \tag{17}
$$

where we used Equation 16 and Assumption 1 (iii) and (iv). Invoking Equation 9 and Equation 16 gives

$$
BL_1\rho \Xi_r = BL_1\rho \frac{1}{NI} \sum_{i=1}^{N} \sum_{k=0}^{I-1} \| \bar{\boldsymbol{x}}_r - \boldsymbol{y}_{r,k}^i \| \leq 2(K+1)BL_1\rho I(2\sigma\eta + \gamma) \leq \frac{1}{2}.
$$

Therefore the conditions of Lemma 9 are satisfied, and we can bound $\| \nabla f(\bar{\boldsymbol{x}}_r) \|$ as follows:

$$
\| \nabla f(\bar{\boldsymbol{x}}_r) \| \mathbb{1}_{\mathcal{A}_r} \leq \frac{\sigma + \frac{\gamma}{\eta} + (AL_0 + BL_1\kappa)\Xi_r}{1 - BL_1\rho \Xi_r}
$$

$$
\leq 2 \left( \sigma + \frac{\gamma}{\eta} + (AL_0 + BL_1\kappa)\Xi_r \right)
$$

$$
\leq 2 \left( \sigma + \frac{\gamma}{\eta} + 2(K+1)I(AL_0 + BL_1\kappa)(2\sigma\eta + \gamma) \right)
$$

$$
\leq 2 \left( 2\sigma + \frac{\gamma}{\eta} \right) (1 + 2(K+1)\eta I(AL_0 + BL_1\kappa))
$$

$$
\leq 4 \left( 2\sigma + \frac{\gamma}{\eta} \right), \tag{18}
$$

where we used Equation 9. Finally, plugging back into Equation 17 yields

$$\|\boldsymbol{x}_{r,1}^i - \bar{\boldsymbol{x}}_r\|\mathbb{1}_{\mathcal{A}_r} \leq \left(1 + 2(K+1)\eta I\left(AL_0 + BL_1\kappa + 4BL_1\rho\left(2\sigma + \frac{\gamma}{\eta}\right)\right)\right)(2\sigma\eta + \gamma)$$
$$\leq 2(2\sigma\eta + \gamma),$$

where we used Equation 9 again. This proves Equation 13 for the base case $k = 1$.

Now suppose that Equation 13 holds for some $k$ with $1 \leq k < I$. Then

$$\begin{aligned}
\|\boldsymbol{x}_{r,k+1}^i - \bar{\boldsymbol{x}}_r\|\mathbb{1}_{\mathcal{A}_r} &= \mathbb{1}_{\mathcal{A}_r}\|\boldsymbol{x}_{r,k}^i - \eta(\nabla F_i(\boldsymbol{x}_{r,k}^i; \xi_{r,k}^i) - \boldsymbol{G}_r^i + \boldsymbol{G}_r) - \bar{\boldsymbol{x}}_r\| \\
&\leq \mathbb{1}_{\mathcal{A}_r}(\|\boldsymbol{x}_{r,k}^i - \bar{\boldsymbol{x}}_r\| + \eta\|\nabla F_i(\boldsymbol{x}_{r,k}^i; \xi_{r,k}^i) - \boldsymbol{G}_r^i + \boldsymbol{G}_r\|) \\
&\leq \mathbb{1}_{\mathcal{A}_r}(\|\boldsymbol{x}_{r,k}^i - \bar{\boldsymbol{x}}_r\| + \eta\|\nabla F_i(\boldsymbol{x}_{r,k}^i; \xi_{r,k}^i) - \nabla f_i(\boldsymbol{x}_{r,k}^i)\|) \\
&\quad + \mathbb{1}_{\mathcal{A}_r}\frac{\eta}{I}\sum_{j=0}^{I-1}\|\nabla F_j(\boldsymbol{y}_{r,j}^i; \xi_{q_r^i;j}^i) - \nabla f_i(\boldsymbol{y}_{r,j}^i)\| \\
&\quad + \mathbb{1}_{\mathcal{A}_r}\frac{\eta}{I}\sum_{j=0}^{I-1}\|\nabla f_i(\boldsymbol{x}_{r,k}^i) - \nabla f_i(\boldsymbol{y}_{r,j}^i)\| + \eta\mathbb{1}_{\mathcal{A}_r}\|\boldsymbol{G}_r\| \\
&\leq \mathbb{1}_{\mathcal{A}_r}\|\boldsymbol{x}_{r,k}^i - \bar{\boldsymbol{x}}_r\| + 2\sigma\eta + \gamma + \eta\mathbb{1}_{\mathcal{A}_r}\|\nabla f_i(\boldsymbol{x}_{r,k}^i) - \nabla f_i(\bar{\boldsymbol{x}}_r)\| \\
&\quad + \mathbb{1}_{\mathcal{A}_r}\frac{\eta}{I}\sum_{j=0}^{I-1}\|\nabla f_i(\bar{\boldsymbol{x}}_r) - \nabla f_i(\boldsymbol{y}_{r,j}^i)\|,
\end{aligned} \tag{19}$$

where we used $\|\boldsymbol{G}_r\| \leq \frac{\gamma}{\eta}$ under the event $\mathcal{A}_r$ and Assumption 1 (iii). We can individually bound each of the two terms $\mathbb{1}_{\mathcal{A}_r}\|\nabla f_i(\boldsymbol{x}_{r,k}^i) - \nabla f_i(\bar{\boldsymbol{x}}_r)\|$ and $\mathbb{1}_{\mathcal{A}_r}\|\nabla f_i(\bar{\boldsymbol{x}}_r) - \nabla f_i(\boldsymbol{y}_{r,j}^i)\|$. By the inductive hypothesis over $k$, together with Equation 9: $\|\boldsymbol{x}_{r,k}^i - \bar{\boldsymbol{x}}_r\| \leq 2I(2\sigma\eta + \gamma) \leq \frac{C}{L_1}$. So we can apply Lemma 2 and obtain:

$$\begin{aligned}
\|\nabla f_i(\boldsymbol{x}_{r,k}^i) - \nabla f_i(\bar{\boldsymbol{x}}_r)\|\mathbb{1}_{\mathcal{A}_r} &\leq \mathbb{1}_{\mathcal{A}_r}(AL_0 + BL_1\|\nabla f_i(\bar{\boldsymbol{x}}_r)\|)\|\boldsymbol{x}_{r,k}^i - \bar{\boldsymbol{x}}_r\| \\
&\leq \mathbb{1}_{\mathcal{A}_r}(AL_0 + BL_1\kappa + BL_1\rho\|\nabla f(\bar{\boldsymbol{x}}_r)\|)\|\boldsymbol{x}_{r,k}^i - \bar{\boldsymbol{x}}_r\| \\
&\leq \mathbb{1}_{\mathcal{A}_r}\left(AL_0 + BL_1\kappa + 4BL_1\rho\left(2\sigma + \frac{\gamma}{\eta}\right)\right)\|\boldsymbol{x}_{r,k}^i - \bar{\boldsymbol{x}}_r\| \\
&\leq \frac{1}{4\eta I}\|\boldsymbol{x}_{r,k}^i - \bar{\boldsymbol{x}}_r\|,
\end{aligned} \tag{20}$$

where we used Assumption 1 (iii) and (iv), Equation 18, and Equation 9. Similarly, by the inductive hypothesis over $r$,

$$\|\bar{\boldsymbol{x}}_r - \boldsymbol{y}_{r,j}^i\| \leq \|\bar{\boldsymbol{x}}_r - \bar{\boldsymbol{x}}_{q_r^i}\| + \|\bar{\boldsymbol{x}}_{q_r^i} - \boldsymbol{x}_{q_r^i,j}^i\| \leq 2(K+1)I(2\sigma\eta + \gamma) \leq \frac{C}{L_1}.$$

So we can apply Lemma 2:

$$\begin{aligned}
\|\nabla f_i(\bar{\boldsymbol{x}}_r) - \nabla f_i(\boldsymbol{y}_{r,j}^i)\| &\leq (AL_0 + BL_1\|\nabla f_i(\bar{\boldsymbol{x}}_r)\|)\|\bar{\boldsymbol{x}}_r - \boldsymbol{y}_{r,j}^i\| \\
&\leq \left(AL_0 + BL_1\kappa + 4BL_1\rho\left(2\sigma + \frac{\gamma}{\eta}\right)\right)\|\bar{\boldsymbol{x}}_r - \boldsymbol{y}_{r,j}^i\| \\
&\leq \frac{1}{4(K+1)\eta I}\|\bar{\boldsymbol{x}}_r - \boldsymbol{y}_{r,j}^i\|
\end{aligned} \tag{21}$$

where the last inequality follows from the assumption Equation 9.

Finally, plugging Equation 20 and Equation 21 into Equation 19 gives

$$\mathbb{1}_{\mathcal{A}_r}\|\boldsymbol{x}_{r,k+1}^i - \bar{\boldsymbol{x}}_r\| \le 2\sigma\eta + \gamma + \left(1 + \frac{1}{4I}\right)\|\boldsymbol{x}_{r,k}^i - \bar{\boldsymbol{x}}_r\| + \frac{1}{4(K+1)I^2}\sum_{j=0}^{I-1}\|\bar{\boldsymbol{x}}_r - \boldsymbol{y}_{r,j}^i\|$$

$$\le 2\sigma\eta + \gamma + 2k\left(1 + \frac{1}{4I}\right)(2\sigma\eta + \gamma) + \frac{1}{4(K+1)I^2}\sum_{j=0}^{I-1}2(K+1)I(2\sigma\eta + \gamma)$$

$$\le 2\left(\frac{1}{2} + k + \frac{k}{4I} + \frac{1}{4}\right)(2\sigma\eta + \gamma)$$

$$\le 2(k+1)(2\sigma\eta + \gamma),$$

where we used the inductive hypothesis over $k$ and the inductive hypothesis over $r$. This fact together with $\mathbb{1}_{\bar{\mathcal{A}}_r}\|\boldsymbol{x}_{r,k+1}^i - \bar{\boldsymbol{x}}_r\| \le 2(k+1)(2\sigma\eta + \gamma)$ (which trivially holds due to the clipping operator), we prove that Equation 13 holds for $k+1$, completing the induction over $k$. Therefore Equation 13 holds for all $k \in [I]$. □

**Lemma 1 restated.** Suppose that Equation 9 holds. Then for any $r \ge 0$ we have

$$\mathbb{1}_{\mathcal{E}} \max_{k\in[I]}\|\boldsymbol{x}_{r,k}^i - \bar{\boldsymbol{x}}_r\| \le 2I(2\sigma\eta + \gamma), \quad \text{for any } i \in \mathcal{S}_r, \tag{22}$$

$$\mathbb{1}_{\mathcal{E}} \max_{k\in[I]}\|\boldsymbol{y}_{r,k}^i - \bar{\boldsymbol{x}}_r\| \le 2(K+1)I(2\sigma\eta + \gamma), \quad \text{for any } i \in [N]. \tag{23}$$

*Proof.* We will use induction to prove Equation 22 by a induction over $r$. And Equation 23 holds as a consequence of the inductive assumption due to Lemma 11. Invoking Lemma 10, we can verify the base case of the induction over $r$. Now we assume $\max_{k\in[I],i\in[N]}\|\boldsymbol{x}_{q,k}^i - \bar{\boldsymbol{x}}_q\| \le 2(K+1)I(2\sigma\eta + \gamma)$ holds for any $q \le r-1$. Then according to Lemma 11, we can prove Equation 22 and Equation 23 immediately. It also completes the induction of Equation 22 over $r$. □

**Corollary 2.** *Suppose that Equation 9 holds. Then for any $r \ge 0$ and $i \in [N]$,*

$$\mathbb{1}_{\mathcal{E}}\Xi_r \le 2(K+1)\eta I\left(\sigma + \frac{\gamma}{\eta}\right) \tag{24}$$

$$\mathbb{1}_{\mathcal{E}\cap\mathcal{A}_r}\|\nabla f(\bar{\boldsymbol{x}}_r)\| \le 4\left(2\sigma + \frac{\gamma}{\eta}\right) \tag{25}$$

$$\mathbb{1}_{\mathcal{E}\cap\mathcal{A}_r}\|\nabla f_i(\bar{\boldsymbol{x}}_r) - \boldsymbol{G}_r^i\| \le 2\sigma + \frac{\gamma}{30\eta} \tag{26}$$

$$\mathbb{1}_{\mathcal{E}\cap\mathcal{A}_r}\|\nabla f(\bar{\boldsymbol{x}}_r) - \boldsymbol{G}_r\| \le 2\sigma + \frac{\gamma}{30\eta}. \tag{27}$$

*Proof.* All proofs of this corollary are under the event $\mathcal{E}$. Equation 24 follows directly from Equation 14 and the definition $\Xi_r = \frac{1}{NI}\sum_{i=1}^{N}\sum_{k=1}^{I}\|\boldsymbol{y}_{r,k}^i - \bar{\boldsymbol{x}}_r\|$. Equation 25 is exactly Equation 18.

Due to Lemma 1, we can apply Lemma 2 such that

$$\mathbb{1}_{\mathcal{E}}\|\nabla f_i(\bar{\boldsymbol{x}}_r) - \boldsymbol{G}_r^i\| \le \mathbb{1}_{\mathcal{E}}\left(\frac{1}{I}\sum_{k=0}^{I-1}\|\nabla f_i(\bar{\boldsymbol{x}}_r) - \nabla f_i(\boldsymbol{y}_{r,k}^i)\| + \frac{1}{I}\sum_{k=0}^{I-1}\|\nabla f_i(\boldsymbol{y}_{r,k}^i) - \nabla F_i(\boldsymbol{y}_{r,k}^i; \xi_{q_r^i,k}^i)\|\right)$$

$$\le \mathbb{1}_{\mathcal{E}}\left(\frac{1}{I}\sum_{k=0}^{I-1}(AL_0 + BL_1\|\nabla f_i(\bar{\boldsymbol{x}}_r)\|)\|\bar{\boldsymbol{x}}_r - \boldsymbol{y}_{r,k}^i\| + \sigma\right)$$

$$\le (AL_0 + BL_1\kappa + BL_1\rho\mathbb{1}_{\mathcal{E}}\|\nabla f(\bar{\boldsymbol{x}}_r)\|)\Xi_r + \sigma.$$

So, using Equation 24 and Equation 25,

$$\mathbb{1}_{\mathcal{E}\cap\mathcal{A}_r}\|\nabla f_i(\bar{\boldsymbol{x}}_r) - \boldsymbol{G}_r^i\| \le 2(K+1)\eta I\left(2\sigma + \frac{\gamma}{\eta}\right)\left(AL_0 + BL_1\kappa + 4BL_1\rho\left(2\sigma + \frac{\gamma}{\eta}\right)\right) + \sigma$$

$$\le \frac{1}{30}\left(2\sigma + \frac{\gamma}{\eta}\right) + \sigma$$

$$\le 2\sigma + \frac{\gamma}{30\eta},$$

where we used Equation 9. Equation 27 follows immediately from Equation 26 by

$$\mathbb{1}_{\mathcal{E}\cap\mathcal{A}_r}\|\nabla f(\bar{\boldsymbol{x}}_r) - \boldsymbol{G}_r\| \le \frac{1}{N}\sum_{i=1}^{N}\mathbb{1}_{\mathcal{E}\cap\mathcal{A}_r}\|\nabla f_i(\bar{\boldsymbol{x}}_r) - \boldsymbol{G}_r^i\| \le 2\sigma + \frac{\gamma}{30\eta}.$$

$\square$

## A.4   Proof of descent inequality

**Lemma 12.** *Suppose Equation 9 holds. Then with probability $1 - 15\delta$, Algorithm 1 satisfies*

$$f(\bar{\boldsymbol{x}}_R) - f(\bar{\boldsymbol{x}}_0)$$
$$\le \sum_{r=0}^{R-1}\mathbb{1}_{\mathcal{A}_r}\left[-\frac{1}{8}\eta I\|\nabla f(\bar{\boldsymbol{x}}_r)\|^2 + \left(2\Gamma_1\eta^2 I^2\left(2\sigma + \frac{\gamma}{\eta}\right) + \frac{216BL_1\sigma^2\eta^2 I\log\frac{1}{\delta}}{S}\right)\|\nabla f(\bar{\boldsymbol{x}}_r)\|\right.$$
$$\left. + 96\Gamma_1^3\eta^4 I^4\left(2\sigma + \frac{\gamma}{\eta}\right)^2 + \frac{108AL_0\sigma^2\eta^2 I\log\frac{1}{\delta}}{S}\right]$$
$$+ \sum_{r=0}^{R-1}\mathbb{1}_{\bar{\mathcal{A}}_r}\left[-\gamma I\left(\frac{3}{4} - 2BL_1\rho\gamma I - \frac{1}{2}BL_1\gamma I\right)\|\nabla f(\bar{\boldsymbol{x}}_r)\|\right.$$
$$\left. - \gamma I\left(\frac{1}{8}\frac{\gamma}{\eta} - 6\sigma - 2AL_0\gamma I - \frac{5}{2}BL_1\kappa\gamma I\right)\right]$$
$$+ \eta\left(12\sigma^2 + 64I\left(\kappa + 5\rho\left(2\sigma + \frac{\gamma}{\eta}\right)\right)^2\left(\frac{1}{S} - \frac{1}{N}\right)\right)\log\frac{1}{\delta},$$

*where $\Gamma_1 = AL_0 + BL_1\kappa + 4BL_1\rho\left(2\sigma + \frac{\gamma}{\eta}\right)$.*

*Proof.* By Lemma 8, $\mathbb{P}(\mathcal{E}) \ge 1 - \delta$. The remainder of the proof will suppose the event $\mathcal{E}$ happens, and the results will hold with probability $1 - \delta$. By Lemma 1 and Equation 9, $\|\bar{\boldsymbol{x}}_{r+1} - \bar{\boldsymbol{x}}_r\| \le \frac{C}{L_1}$. Denote $\mathcal{A}_r = \{\|\boldsymbol{G}_r\| \le \gamma/\eta\}$. Therefore we may apply Lemma 3 to obtain

$$f(\bar{\boldsymbol{x}}_{r+1}) - f(\bar{\boldsymbol{x}}_r)$$
$$\le \langle\nabla f(\bar{\boldsymbol{x}}_r), \bar{\boldsymbol{x}}_{r+1} - \bar{\boldsymbol{x}}_r\rangle + \frac{AL_0 + BL_1\|\nabla f(\bar{\boldsymbol{x}}_r)\|}{2}\|\bar{\boldsymbol{x}}_{r+1} - \bar{\boldsymbol{x}}_r\|^2$$
$$\le -\mathbb{1}_{\mathcal{A}_r}\eta\left\langle\nabla f(\bar{\boldsymbol{x}}_r), \frac{1}{S}\sum_{i\in\mathcal{S}_r}\sum_{k=0}^{I-1}\boldsymbol{g}_{r,k}^i\right\rangle - \mathbb{1}_{\bar{\mathcal{A}}_r}\gamma\left\langle\nabla f(\bar{\boldsymbol{x}}_r), \frac{1}{S}\sum_{i\in\mathcal{S}_r}\sum_{k=0}^{I-1}\frac{\boldsymbol{g}_{r,k}^i}{\|\boldsymbol{g}_{r,k}^i\|}\right\rangle +$$
$$\mathbb{1}_{\mathcal{A}_r}\frac{AL_0 + BL_1\|\nabla f(\bar{\boldsymbol{x}}_r)\|}{2}\|\bar{\boldsymbol{x}}_{r+1} - \bar{\boldsymbol{x}}_r\|^2 + \mathbb{1}_{\bar{\mathcal{A}}_r}\frac{AL_0 + BL_1\|\nabla f(\bar{\boldsymbol{x}}_r)\|}{2}\|\bar{\boldsymbol{x}}_{r+1} - \bar{\boldsymbol{x}}_r\|^2.$$

Summing over $r = 0, \dots, R - 1$ yields

$$f(\bar{\boldsymbol{x}}_R) - f(\bar{\boldsymbol{x}}_0) \leq \underbrace{-\eta \sum_{r=0}^{R-1} \mathbb{1}_{\mathcal{A}_r} \left\langle \nabla f(\bar{\boldsymbol{x}}_r), \frac{1}{S} \sum_{i \in \mathcal{S}_r} \sum_{k=0}^{I-1} \boldsymbol{g}_{r,k}^i \right\rangle}_{A_1}$$

$$\underbrace{-\gamma \sum_{r=0}^{R-1} \mathbb{1}_{\bar{\mathcal{A}}_r} \left\langle \nabla f(\bar{\boldsymbol{x}}_r), \frac{1}{S} \sum_{i \in \mathcal{S}_r} \sum_{k=0}^{I-1} \frac{\boldsymbol{g}_{r,k}^i}{\|\boldsymbol{g}_{r,k}^i\|} \right\rangle}_{A_2}$$

$$\underbrace{+ \sum_{r=0}^{R-1} \mathbb{1}_{\mathcal{A}_r} \frac{AL_0 + BL_1 \|\nabla f(\bar{\boldsymbol{x}}_r)\|}{2} \|\bar{\boldsymbol{x}}_{r+1} - \bar{\boldsymbol{x}}_r\|^2}_{A_3}$$

$$\underbrace{+ \sum_{r=0}^{R-1} \mathbb{1}_{\bar{\mathcal{A}}_r} \frac{AL_0 + BL_1 \|\nabla f(\bar{\boldsymbol{x}}_r)\|}{2} \|\bar{\boldsymbol{x}}_{r+1} - \bar{\boldsymbol{x}}_r\|^2}_{A_4} . \tag{28}$$

We proceed by bounding each of the four terms in Equation 28.

**Bounding $A_1$.** Denote

$$\begin{aligned}
\boldsymbol{\epsilon}_{r,k}^i &= \nabla F_i(\boldsymbol{x}_{r,k}^i; \xi_{r,k}^i) - \nabla f_i(\boldsymbol{x}_{r,k}^i), \\
\boldsymbol{\epsilon}_r^{(1)} &= \frac{1}{S} \sum_{i \in \mathcal{S}_r} \boldsymbol{G}_r^i - \boldsymbol{G}_r, \\
\boldsymbol{\epsilon}_r^{(2)} &= \frac{1}{S} \sum_{i \in \mathcal{S}_r} \nabla f_i(\bar{\boldsymbol{x}}_r) - \nabla f(\bar{\boldsymbol{x}}_r).
\end{aligned}$$

Then we can write

$$\begin{aligned}
\frac{1}{S} \sum_{i \in \mathcal{S}_r} \boldsymbol{g}_{r,k}^i &= \frac{1}{S} \sum_{i \in \mathcal{S}_r} \nabla F_i(\boldsymbol{x}_{r,k}^i; \xi_{r,k}^i) - \frac{1}{S} \sum_{i \in \mathcal{S}_r} \boldsymbol{G}_r^i + \boldsymbol{G}_r \\
&= \frac{1}{S} \sum_{i \in \mathcal{S}_r} \nabla f_i(\boldsymbol{x}_{r,k}^i) + \frac{1}{S} \sum_{i \in \mathcal{S}_r} \boldsymbol{\epsilon}_{r,k}^i - \boldsymbol{\epsilon}_r^{(1)} \\
&= \frac{1}{S} \sum_{i \in \mathcal{S}_r} \nabla f_i(\bar{\boldsymbol{x}}_r) + \frac{1}{S} \sum_{i \in \mathcal{S}_r} (\nabla f_i(\boldsymbol{x}_{r,k}^i) - \nabla f_i(\bar{\boldsymbol{x}}_r)) + \frac{1}{S} \sum_{i \in \mathcal{S}_r} \boldsymbol{\epsilon}_{r,k}^i - \boldsymbol{\epsilon}_r^{(1)} \\
&= \nabla f(\bar{\boldsymbol{x}}_r) - \boldsymbol{\epsilon}_r^{(1)} + \boldsymbol{\epsilon}_r^{(2)} + \frac{1}{S} \sum_{i \in \mathcal{S}_r} (\nabla f_i(\boldsymbol{x}_{r,k}^i) - \nabla f_i(\bar{\boldsymbol{x}}_r)) + \frac{1}{S} \sum_{i \in \mathcal{S}_r} \boldsymbol{\epsilon}_{r,k}^i.
\end{aligned}$$

It follows that

$$
\begin{aligned}
A_1 &= -\eta \sum_{r=0}^{R-1} \mathbb{1}_{\mathcal{A}_r} \sum_{k=0}^{I-1} \left\langle \nabla f(\bar{\boldsymbol{x}}_r), \frac{1}{S} \sum_{i \in \mathcal{S}_r} \boldsymbol{g}_{r,k}^i \right\rangle \\
&= -\eta I \sum_{r=0}^{R-1} \mathbb{1}_{\mathcal{A}_r} \|\nabla f(\bar{\boldsymbol{x}}_r)\|^2 - \eta \sum_{r=0}^{R-1} \mathbb{1}_{\mathcal{A}_r} \left\langle \nabla f(\bar{\boldsymbol{x}}_r), \frac{1}{S} \sum_{i \in \mathcal{S}_r} \sum_{k=0}^{I-1} \nabla f_i(\boldsymbol{x}_{r,k}^i) - \nabla f_i(\bar{\boldsymbol{x}}_r) \right\rangle \\
&\quad - \eta \sum_{r=0}^{R-1} \mathbb{1}_{\mathcal{A}_r} \sum_{k=0}^{I-1} \left\langle \nabla f(\bar{\boldsymbol{x}}_r), \frac{1}{S} \sum_{i \in \mathcal{S}_r} \boldsymbol{\epsilon}_{r,k}^i \right\rangle + \eta I \sum_{r=0}^{R-1} \mathbb{1}_{\mathcal{A}_r} \left\langle \nabla f(\bar{\boldsymbol{x}}_r), \boldsymbol{\epsilon}_r^{(1)} - \boldsymbol{\epsilon}_r^{(2)} \right\rangle \\
&\leq -\eta I \sum_{r=0}^{R-1} \mathbb{1}_{\mathcal{A}_r} \|\nabla f(\bar{\boldsymbol{x}}_r)\|^2 - \eta \sum_{r=0}^{R-1} \mathbb{1}_{\mathcal{A}_r} \left\langle \nabla f(\bar{\boldsymbol{x}}_r), \frac{1}{S} \sum_{i \in \mathcal{S}_r} \sum_{k=0}^{I-1} \nabla f_i(\boldsymbol{x}_{r,k}^i) - \nabla f_i(\bar{\boldsymbol{x}}_r) \right\rangle \\
&\quad + \eta \left| \sum_{r=0}^{R-1} \mathbb{1}_{\mathcal{A}_r} \sum_{k=0}^{I-1} \left\langle \nabla f(\bar{\boldsymbol{x}}_r), \frac{1}{S} \sum_{i \in \mathcal{S}_r} \boldsymbol{\epsilon}_{r,k}^i \right\rangle \right| + \eta I \left| \sum_{r=0}^{R-1} \mathbb{1}_{\mathcal{A}_r} \left\langle \nabla f(\bar{\boldsymbol{x}}_r), \boldsymbol{\epsilon}_r^{(1)} \right\rangle \right| \\
&\quad + \eta I \left| \sum_{r=0}^{R-1} \mathbb{1}_{\mathcal{A}_r} \left\langle \nabla f(\bar{\boldsymbol{x}}_r), \boldsymbol{\epsilon}_r^{(2)} \right\rangle \right|.
\end{aligned}
\tag{29}
$$

Since $\|\boldsymbol{x}_{r,k}^i - \bar{\boldsymbol{x}}_r\| \leq C/L_1$ due to Equation 13, we can apply Lemma 2 such that

$$
\begin{aligned}
&- \eta \sum_{r=0}^{R-1} \mathbb{1}_{\mathcal{A}_r} \left\langle \nabla f(\bar{\boldsymbol{x}}_r), \frac{1}{S} \sum_{i \in \mathcal{S}_r} \sum_{k=0}^{I-1} \nabla f_i(\boldsymbol{x}_{r,k}^i) - \nabla f_i(\bar{\boldsymbol{x}}_r) \right\rangle \\
&\leq \eta \sum_{r=0}^{R-1} \mathbb{1}_{\mathcal{A}_r} \|\nabla f(\bar{\boldsymbol{x}}_r)\| \left\| \frac{1}{S} \sum_{i \in \mathcal{S}_r} \sum_{k=0}^{I-1} \nabla f_i(\boldsymbol{x}_{r,k}^i) - \nabla f_i(\bar{\boldsymbol{x}}_r) \right\| \\
&\leq \eta \sum_{r=0}^{R-1} \mathbb{1}_{\mathcal{A}_r} \|\nabla f(\bar{\boldsymbol{x}}_r)\| \cdot \frac{1}{S} \sum_{i \in \mathcal{S}_r} \sum_{k=0}^{I-1} \|\nabla f_i(\boldsymbol{x}_{r,k}^i) - \nabla f_i(\bar{\boldsymbol{x}}_r)\| \\
&\leq \eta \sum_{r=0}^{R-1} \mathbb{1}_{\mathcal{A}_r} \|\nabla f(\bar{\boldsymbol{x}}_r)\| \cdot \frac{1}{S} \sum_{i \in \mathcal{S}_r} (AL_0 + BL_1 \|\nabla f_i(\bar{\boldsymbol{x}}_r)\|) \sum_{k=0}^{I-1} \|\boldsymbol{x}_{r,k}^i - \bar{\boldsymbol{x}}_r\| \\
&\overset{(i)}{\leq} 2\eta^2 I^2 \left( 2\sigma + \frac{\gamma}{\eta} \right) \sum_{r=0}^{R-1} \mathbb{1}_{\mathcal{A}_r} \|\nabla f(\bar{\boldsymbol{x}}_r)\| (AL_0 + BL_1\kappa + BL_1\rho \|\nabla f(\bar{\boldsymbol{x}}_r)\|) \\
&\overset{(ii)}{\leq} 2\eta^2 I^2 (AL_0 + BL_1\kappa) \left( 2\sigma + \frac{\gamma}{\eta} \right) \sum_{r=0}^{R-1} \mathbb{1}_{\mathcal{A}_r} \|\nabla f(\bar{\boldsymbol{x}}_r)\| \\
&\quad + 2\eta^2 I^2 BL_1\rho \left( 2\sigma + \frac{\gamma}{\eta} \right) \sum_{r=0}^{R-1} \mathbb{1}_{\mathcal{A}_r} \|\nabla f(\bar{\boldsymbol{x}}_r)\|^2 \\
&\overset{(iii)}{\leq} 2\eta^2 I^2 (AL_0 + BL_1\kappa) \left( 2\sigma + \frac{\gamma}{\eta} \right) \sum_{r=0}^{R-1} \mathbb{1}_{\mathcal{A}_r} \|\nabla f(\bar{\boldsymbol{x}}_r)\| + \frac{1}{8}\eta I \sum_{r=0}^{R-1} \mathbb{1}_{\mathcal{A}_r} \|\nabla f(\bar{\boldsymbol{x}}_r)\|^2,
\end{aligned}
$$

where $(i)$ holds due to Equation 13; $(ii)$ follows from Assumption 1(iv); and $(iii)$ follows from Equation 9. Plugging into Equation 29,

$$A_1 \leq -\frac{7}{8}\eta I \sum_{r=0}^{R-1} \mathbb{1}_{\mathcal{A}_r} \|\nabla f(\bar{\boldsymbol{x}}_r)\|^2 + 2\eta^2 I^2 (AL_0 + BL_1\kappa)\left(2\sigma + \frac{\gamma}{\eta}\right)\sum_{r=0}^{R-1}\mathbb{1}_{\mathcal{A}_r}\|\nabla f(\bar{\boldsymbol{x}}_r)\|$$

$$+ \eta \underbrace{\left|\sum_{r=0}^{R-1} \mathbb{1}_{\mathcal{A}_r} \sum_{k=0}^{I-1}\left\langle \nabla f(\bar{\boldsymbol{x}}_r), \frac{1}{S}\sum_{i\in\mathcal{S}_r}\boldsymbol{\epsilon}_{r,k}^i\right\rangle\right|}_{B_1}$$

$$+ \eta I \underbrace{\left|\sum_{r=0}^{R-1}\mathbb{1}_{\mathcal{A}_r}\left\langle \nabla f(\bar{\boldsymbol{x}}_r), \boldsymbol{\epsilon}_r^{(1)}\right\rangle\right|}_{B_2}$$

$$+ \eta I \underbrace{\left|\sum_{r=0}^{R-1}\mathbb{1}_{\mathcal{A}_r}\left\langle \nabla f(\bar{\boldsymbol{x}}_r), \boldsymbol{\epsilon}_r^{(2)}\right\rangle\right|}_{B_3}. \tag{30}$$

Denote each of the three last terms (those involving $\boldsymbol{\epsilon}_{r,k}^i$, $\boldsymbol{\epsilon}_r^{(1)}$, and $\boldsymbol{\epsilon}_r^{(2)}$, respectively) as $B_1$, $B_2$, $B_3$. $B_1$ can be bounded using the martingale concentration bound in Lemma 5.

Denoting $\mathcal{G}_{r,k} = \sigma(\mathcal{F}_r, \mathcal{S}_r, \{\xi_{r,\ell}^i : i \in \mathcal{S}_r, \ell \leq k\})$, we have

$$\mathbb{E}\left[\mathbb{1}_{\mathcal{A}_r}\left\langle \nabla f(\bar{\boldsymbol{x}}_r), \frac{1}{S}\sum_{i\in\mathcal{S}_r}\boldsymbol{\epsilon}_{r,k}^i\right\rangle \mid \mathcal{G}_{r,k-1}\right] = 0$$

so that $\left\{\mathbb{1}_{\mathcal{A}_r}\left\langle \nabla f(\bar{\boldsymbol{x}}_r), \frac{1}{S}\sum_{i\in\mathcal{S}_r}\boldsymbol{\epsilon}_{r,k}^i\right\rangle\right\}_{r,k}$ is a martingale difference sequence with respect to $\mathcal{G}_{r,k}$. Since $\mathbb{1}_{\mathcal{A}_r} \in \mathcal{F}_r \in \mathcal{G}_{r,k}$ for any $0 \leq k \leq I$, we have

$$\mathbb{E}\left[\exp\left(\frac{\mathbb{1}_{\mathcal{A}_r}\left\langle \nabla f(\bar{\boldsymbol{x}}_r), \frac{1}{S}\sum_{i\in\mathcal{S}_r}\boldsymbol{\epsilon}_{r,k}^i\right\rangle^2}{\mathbb{1}_{\mathcal{A}_r}\|\nabla f(\bar{\boldsymbol{x}}_r)\|^2\sigma^2}\right) \mid \mathcal{G}_{r,k-1}\right]$$

$$\leq \mathbb{E}\left[\exp\left(\frac{\frac{1}{S}\sum_{i\in\mathcal{S}_r}\|\boldsymbol{\epsilon}_{r,k}^i\|^2}{\sigma^2}\right) \mid \mathcal{G}_{r,k-1}\right]$$

$$\leq \exp(1),$$

where we used Cauchy-Schwarz, Jensen's inequality, and $\|\boldsymbol{\epsilon}_{r,k}^i\| \leq \sigma$. Therefore we can apply Lemma 5 to obtain

$$B_1 \leq \frac{3}{4}\eta\sigma^2 I\lambda_1 \sum_{r=0}^{R-1}\mathbb{1}_{\mathcal{A}_r}\|\nabla f(\bar{\boldsymbol{x}}_r)\|^2 + \frac{\eta}{\lambda_1}\log\frac{1}{\delta}, \tag{31}$$

with probability $1 - \delta$ for any $\lambda_1 > 0$.

The concentration bounds for $B_2$ and $B_3$ will rely on the concentration bound for sampling without replacement in Lemma 6. We start with $B_2$. Notice that

$$\mathbb{1}_{\mathcal{A}_r}\langle \nabla f(\bar{\boldsymbol{x}}_r), \boldsymbol{\epsilon}_r^{(1)}\rangle = \frac{1}{S}\sum_{i\in\mathcal{S}_r}\mathbb{1}_{\mathcal{A}_r}\langle \nabla f(\bar{\boldsymbol{x}}_r), \boldsymbol{G}_r^i\rangle - \frac{1}{N}\sum_{i=1}^{N}\mathbb{1}_{\mathcal{A}_r}\langle \nabla f(\bar{\boldsymbol{x}}_r), \boldsymbol{G}_r^i\rangle, \tag{32}$$

so we must upper bound $\mathbb{1}_{\mathcal{A}_r} |\langle \nabla f(\bar{\boldsymbol{x}}_r), \boldsymbol{G}_r^i \rangle|$ to apply Lemma 6. Under $\mathcal{A}_r$,

$$
\begin{aligned}
\|\boldsymbol{G}_r^i\| &\leq \|\nabla f_i(\bar{\boldsymbol{x}}_r)\| + \|\nabla f_i(\bar{\boldsymbol{x}}_r) - \boldsymbol{G}_r^i\| \\
&\leq \kappa + \rho \|\nabla f(\bar{\boldsymbol{x}}_r)\| + \|\nabla f_i(\bar{\boldsymbol{x}}_r) - \boldsymbol{G}_r^i\| \\
&\overset{(i)}{\leq} \kappa + 4\rho \left( 2\sigma + \frac{\gamma}{\eta} \right) + \left( 2\sigma + \frac{\gamma}{\eta} \right) \\
&\leq \kappa + (4\rho + 1) \left( 2\sigma + \frac{\gamma}{\eta} \right) \\
&\overset{(ii)}{\leq} \kappa + 5\rho \left( 2\sigma + \frac{\gamma}{\eta} \right).
\end{aligned}
\tag{33}
$$

where $(i)$ holds due to Equation 25 and Equation 26; and $(ii)$ holds since $\rho \geq 1$. Therefore

$$
\mathbb{1}_{\mathcal{A}_r} |\langle \nabla f(\bar{\boldsymbol{x}}_r), \boldsymbol{G}_r^i \rangle| \leq \left( \kappa + 5\rho \left( 2\sigma + \frac{\gamma}{\eta} \right) \right) \mathbb{1}_{\mathcal{A}_r} \|\nabla f(\bar{\boldsymbol{x}}_r)\|,
$$

and the condition of Lemma 6 is satisfied. Notice that $\mathbb{1}_{\mathcal{A}_r} \in \mathcal{F}_r$ because $\boldsymbol{G}_r$ is constructed by the history gradients before round $r$. Therefore, applying Lemma 6 to Equation 32 gives[4]

$$
\begin{aligned}
&\mathbb{E}_r[\exp(\lambda_2 \mathbb{1}_{\mathcal{A}_r} \langle \nabla f(\bar{\boldsymbol{x}}_r), \boldsymbol{\epsilon}_r^{(1)} \rangle)] \\
&\leq \exp \left\{ \frac{1}{2} \lambda_2^2 \left( \kappa + 5\rho \left( 2\sigma + \frac{\gamma}{\eta} \right) \right)^2 \frac{S+1}{S^2} \left( 1 - \frac{S}{N} \right) \mathbb{1}_{\mathcal{A}_r} \|\nabla f(\bar{\boldsymbol{x}}_r)\|^2 \right\} \\
&\leq \exp \left\{ \lambda_2^2 \left( \kappa + 5\rho \left( 2\sigma + \frac{\gamma}{\eta} \right) \right)^2 \left( \frac{1}{S} - \frac{1}{N} \right) \mathbb{1}_{\mathcal{A}_r} \|\nabla f(\bar{\boldsymbol{x}}_r)\|^2 \right\} \\
&=: \exp \left\{ \lambda_2^2 M_r \right\}
\end{aligned}
\tag{34}
$$

Let

$$
\begin{aligned}
Y_r &= \exp \left\{ \sum_{q=0}^{r} \lambda_2 \left( \mathbb{1}_{\mathcal{A}_q} \langle \nabla f(\bar{\boldsymbol{x}}_q), \boldsymbol{\epsilon}_q^{(1)} \rangle - \lambda_2 M_r \right) \right\} \\
&= Y_{r-1} \exp \left\{ \lambda_2 \mathbb{1}_{\mathcal{A}_r} \langle \nabla f(\bar{\boldsymbol{x}}_r), \boldsymbol{\epsilon}_r^{(1)} \rangle - \lambda_2^2 M_r \right\}.
\end{aligned}
$$

Then we know $\mathbb{E}_r[Y_r] \leq Y_{r-1}$ by Equation 34, which implies

$$
\mathbb{E}[Y_{R-1}] = \mathbb{E}[\mathbb{E}_{R-1}[Y_{R-1}]] \leq \mathbb{E}[Y_{R-2}] \leq \ldots \leq \mathbb{E}[Y_0] \leq 1.
$$

Therefore for any $\lambda_2 > 0$, we have

$$
\begin{aligned}
&\mathbb{P} \left( \sum_{r=0}^{R-1} \mathbb{1}_{\mathcal{A}_r} \langle \nabla f(\bar{\boldsymbol{x}}_r), \boldsymbol{\epsilon}_r^{(1)} \rangle \geq \lambda_2 \sum_{r=0}^{R-1} M_r + \frac{1}{\lambda_2} \log \frac{1}{\delta} \right) \\
&= \mathbb{P} \left( \lambda_2 \sum_{r=0}^{R-1} \mathbb{1}_{\mathcal{A}_r} \langle \nabla f(\bar{\boldsymbol{x}}_r), \boldsymbol{\epsilon}_r^{(1)} \rangle \geq \lambda_2^2 \sum_{r=0}^{R-1} M_r + \log \frac{1}{\delta} \right) \\
&= \mathbb{P} \left( \exp \left\{ \sum_{r=0}^{R-1} \lambda_2 \left( \mathbb{1}_{\mathcal{A}_r} \langle \nabla f(\bar{\boldsymbol{x}}_r), \boldsymbol{\epsilon}_r^{(1)} \rangle - \lambda_2 M_r \right) \right\} \geq \frac{1}{\delta} \right) \\
&= \mathbb{P} \left( Y_{R-1} \geq \frac{1}{\delta} \right) \\
&\leq \delta,
\end{aligned}
$$

where the last line uses Markov's inequality. Repeating this argument for $-\mathbb{1}_{\mathcal{A}_r} \langle \nabla f(\bar{\boldsymbol{x}}_r), \boldsymbol{\epsilon}_r^{(1)} \rangle$, we can obtain

$$
\mathbb{P} \left( \left| \sum_{r=0}^{R-1} \mathbb{1}_{\mathcal{A}_r} \langle \nabla f(\bar{\boldsymbol{x}}_r), \boldsymbol{\epsilon}_r^{(1)} \rangle \right| \geq \lambda_2 \sum_{r=0}^{R-1} M_r + \frac{1}{\lambda_2} \log \frac{1}{\delta} \right) \geq 1 - 2\delta,
$$

---

[4]Here $\mathbb{E}_r[\cdot]$ takes expectation over the randomness of subsampling, that is $\mathcal{S}_r$.

which means that

$$B_2 \leq \eta I \lambda_2 \left( \kappa + 5\rho \left( 2\sigma + \frac{\gamma}{\eta} \right) \right)^2 \left( \frac{1}{S} - \frac{1}{N} \right) \sum_{r=0}^{R-1} \mathbb{1}_{\mathcal{A}_r} \|\nabla f(\bar{\boldsymbol{x}}_r)\|^2 + \frac{\eta I}{\lambda_2} \log \frac{1}{\delta}, \quad (35)$$

holds with probability $1 - 2\delta$.

The same argument can be applied to $B_3$. Notice that

$$\mathbb{1}_{\mathcal{A}_r} \langle \nabla f(\bar{\boldsymbol{x}}_r), \boldsymbol{\epsilon}_r^{(2)} \rangle = \frac{1}{S} \sum_{i \in \mathcal{S}_r} \mathbb{1}_{\mathcal{A}_r} \langle \nabla f(\bar{\boldsymbol{x}}_r), \nabla f_i(\bar{\boldsymbol{x}}_r) \rangle - \frac{1}{N} \sum_{i=1}^{N} \mathbb{1}_{\mathcal{A}_r} \langle \nabla f(\bar{\boldsymbol{x}}_r), \nabla f_i(\bar{\boldsymbol{x}}_r) \rangle,$$

so we must upper bound $\mathbb{1}_{\mathcal{A}_r} |\langle \nabla f(\bar{\boldsymbol{x}}_r), \nabla f_i(\bar{\boldsymbol{x}}_r) \rangle|$ to apply Lemma 6. Using Equation 25 in Corollary 2, we have

$$\mathbb{1}_{\mathcal{A}_r} \|\nabla f_i(\bar{\boldsymbol{x}}_r)\| \leq \kappa + \rho \mathbb{1}_{\mathcal{A}_r} \|\nabla f(\bar{\boldsymbol{x}}_r)\| \leq \kappa + 4\rho \left( 2\sigma + \frac{\gamma}{\eta} \right) \leq \kappa + 5\rho \left( 2\sigma + \frac{\gamma}{\eta} \right).$$

This matches the corresponding upper bound in Equation 33 from the bound of $B_2$. We may therefore apply an identical argument as in the case of $B_2$ and obtain

$$B_3 \leq \eta I \lambda_2 \left( \kappa + 5\rho \left( 2\sigma + \frac{\gamma}{\eta} \right) \right)^2 \left( \frac{1}{S} - \frac{1}{N} \right) \sum_{r=0}^{R-1} \mathbb{1}_{\mathcal{A}_r} \|\nabla f(\bar{\boldsymbol{x}}_r)\|^2 + \frac{\eta I}{\lambda_2} \log \frac{2}{\delta}, \quad (36)$$

with probability $1 - 2\delta$.

Combining Equation 31, Equation 35, and Equation 36 into Equation 30, yields that, with probability $1 - 5\delta$,

$$A_1 \leq -\eta I \left( \frac{7}{8} - \frac{3}{4}\sigma^2 \lambda_1 - 2\lambda_2 \left( \kappa + 5\rho \left( 2\sigma + \frac{\gamma}{\eta} \right) \right)^2 \left( \frac{1}{S} - \frac{1}{N} \right) \right) \sum_{r=0}^{R-1} \mathbb{1}_{\mathcal{A}_r} \|\nabla f(\bar{\boldsymbol{x}}_r)\|^2 +$$

$$2\eta^2 I^2 (AL_0 + BL_1 \kappa) \left( 2\sigma + \frac{\gamma}{\eta} \right) \sum_{r=0}^{R-1} \mathbb{1}_{\mathcal{A}_r} \|\nabla f(\bar{\boldsymbol{x}}_r)\| + \eta \left( \frac{1}{\lambda_1} + \frac{2I}{\lambda_2} \right) \log \frac{1}{\delta}$$

$$\leq -\frac{3}{4}\eta I \sum_{r=0}^{R-1} \mathbb{1}_{\mathcal{A}_r} \|\nabla f(\bar{\boldsymbol{x}}_r)\|^2 + 2\eta^2 I^2 (AL_0 + BL_1 \kappa) \left( 2\sigma + \frac{\gamma}{\eta} \right) \sum_{r=0}^{R-1} \mathbb{1}_{\mathcal{A}_r} \|\nabla f(\bar{\boldsymbol{x}}_r)\| +$$

$$\eta \left( 12\sigma^2 + 64I \left( \kappa + 5\rho \left( 2\sigma + \frac{\gamma}{\eta} \right) \right)^2 \left( \frac{1}{S} - \frac{1}{N} \right) \right) \log \frac{1}{\delta}, \quad (37)$$

where we chose

$$\lambda_1 = \frac{1}{12\sigma^2},$$

$$\lambda_2 = \frac{1}{32 \left( \kappa + 5\rho \left( 2\sigma + \frac{\gamma}{\eta} \right) \right)^2 \left( \frac{1}{S} - \frac{1}{N} \right)}.$$

**Bounding $A_2$.** From Lemma 4,

$$-\left\langle \nabla f(\bar{\boldsymbol{x}}_r), \frac{\boldsymbol{g}_{r,k}^i}{\|\boldsymbol{g}_{r,k}^i\|} \right\rangle \leq -\mu \|\nabla f(\bar{\boldsymbol{x}}_r)\| - (1-\mu)\|\boldsymbol{g}_{r,k}^i\| + (1+\mu)\|\boldsymbol{g}_{r,k}^i - \nabla f(\bar{\boldsymbol{x}}_r)\|$$

for any $\mu \geq 0$. Also,

$$\mathbb{1}_{\bar{\mathcal{A}}_r} \|\boldsymbol{g}_{r,k}^i\| = \mathbb{1}_{\bar{\mathcal{A}}_r} \left\| \nabla F_i(\boldsymbol{x}_{r,k}^i; \xi_{r,k}^i) - \boldsymbol{G}_r^i + \boldsymbol{G}_r \right\|$$

$$\geq \mathbb{1}_{\bar{\mathcal{A}}_r} \left( \|\boldsymbol{G}_r\| - \left\| \nabla F_i(\boldsymbol{x}_{r,k}^i; \xi_{r,k}^i) - \boldsymbol{G}_r^i \right\| \right)$$

$$\geq \mathbb{1}_{\bar{\mathcal{A}}_r} \left( \frac{\gamma}{\eta} - \left\| \nabla F_i(\boldsymbol{x}_{r,k}^i; \xi_{r,k}^i) - \nabla f_i(\boldsymbol{x}_{r,k}^i) \right\| - \left\| \nabla f_i(\boldsymbol{x}_{r,k}^i) - \nabla f_i(\bar{\boldsymbol{x}}_r) \right\| - \left\| \nabla f_i(\bar{\boldsymbol{x}}_r) - \boldsymbol{G}_r^i \right\| \right)$$

$$\geq \mathbb{1}_{\bar{\mathcal{A}}_r} \left( \frac{\gamma}{\eta} - \sigma - (AL_0 + BL_1 \|\nabla f_i(\bar{\boldsymbol{x}}_r)\|) \left\| \boldsymbol{x}_{r,k}^i - \bar{\boldsymbol{x}}_r \right\| - \left\| \nabla f_i(\bar{\boldsymbol{x}}_r) - \boldsymbol{G}_r^i \right\| \right)$$

$$\geq \mathbb{1}_{\bar{\mathcal{A}}_r} \left( \frac{\gamma}{\eta} - \sigma - \gamma I (AL_0 + BL_1 \kappa + BL_1 \rho \|\nabla f(\bar{\boldsymbol{x}}_r)\|) - \left\| \nabla f_i(\bar{\boldsymbol{x}}_r) - \boldsymbol{G}_r^i \right\| \right),$$

and

$$\mathbb{1}_{\bar{\mathcal{A}}_r}\|\boldsymbol{g}_{r,k}^i - \nabla f(\bar{\boldsymbol{x}}_r)\| = \mathbb{1}_{\bar{\mathcal{A}}_r}\left\|\nabla F_i(\boldsymbol{x}_{r,k}^i; \xi_{r,k}^i) - \boldsymbol{G}_r^i + \boldsymbol{G}_r - \nabla f(\bar{\boldsymbol{x}}_r)\right\|$$

$$\leq \mathbb{1}_{\bar{\mathcal{A}}_r}\left(\left\|\nabla F_i(\boldsymbol{x}_{r,k}^i; \xi_{r,k}^i) - \boldsymbol{G}_r^i\right\| + \left\|\boldsymbol{G}_r - \nabla f(\bar{\boldsymbol{x}}_r)\right\|\right)$$

$$\leq \mathbb{1}_{\bar{\mathcal{A}}_r}\Bigg(\left\|\nabla F_i(\boldsymbol{x}_{r,k}^i; \xi_{r,k}^i) - \nabla f_i(\boldsymbol{x}_{r,k}^i)\right\| + \left\|\nabla f_i(\boldsymbol{x}_{r,k}^i) - \nabla f_i(\bar{\boldsymbol{x}}_r)\right\|$$

$$+ \left\|\nabla f_i(\bar{\boldsymbol{x}}_r) - \boldsymbol{G}_r^i\right\| + \frac{1}{N}\sum_{j=1}^{N}\left\|\boldsymbol{G}_r^j - \nabla F_j(\bar{\boldsymbol{x}}_r)\right\|\Bigg)$$

$$\leq \mathbb{1}_{\bar{\mathcal{A}}_r}\Bigg(\sigma + (AL_0 + BL_1\|\nabla f_i(\bar{\boldsymbol{x}}_r)\|)\left\|\boldsymbol{x}_{r,k}^i - \bar{\boldsymbol{x}}_r\right\|$$

$$+ \left\|\nabla f_i(\bar{\boldsymbol{x}}_r) - \boldsymbol{G}_r^i\right\| + \frac{1}{N}\sum_{j=1}^{N}\left\|\boldsymbol{G}_r^j - \nabla F_j(\bar{\boldsymbol{x}}_r)\right\|\Bigg)$$

$$\leq \mathbb{1}_{\bar{\mathcal{A}}_r}\Bigg(\sigma + \gamma I(AL_0 + BL_1\kappa + BL_1\rho\|\nabla f(\bar{\boldsymbol{x}}_r)\|)$$

$$+ \left\|\nabla f_i(\bar{\boldsymbol{x}}_r) - \boldsymbol{G}_r^i\right\| + \frac{1}{N}\sum_{j=1}^{N}\left\|\boldsymbol{G}_r^j - \nabla F_j(\bar{\boldsymbol{x}}_r)\right\|\Bigg).$$

Therefore

$$-\mathbb{1}_{\bar{\mathcal{A}}_r}\left\langle \nabla f(\bar{\boldsymbol{x}}_r), \frac{\boldsymbol{g}_{r,k}^i}{\|\boldsymbol{g}_{r,k}^i\|}\right\rangle$$

$$\leq \mathbb{1}_{\bar{\mathcal{A}}_r}\Bigg(-\mu\|\nabla f(\bar{\boldsymbol{x}}_r)\| - (1-\mu)\frac{\gamma}{\eta} + 2\sigma + 2\gamma I(AL_0 + BL_1\kappa + BL_1\rho\|\nabla f(\bar{\boldsymbol{x}}_r)\|)$$

$$+ 2\|\nabla f_i(\bar{\boldsymbol{x}}_r) - \boldsymbol{G}_r^i\| + \frac{1+\mu}{N}\sum_{j=1}^{N}\|\boldsymbol{G}_r^j - \nabla F_j(\bar{\boldsymbol{x}}_r)\|\Bigg),$$

and

$$-\mathbb{1}_{\bar{\mathcal{A}}_r}\frac{1}{S}\sum_{i \in \mathcal{S}_r}\left\langle \nabla f(\bar{\boldsymbol{x}}_r), \frac{\boldsymbol{g}_{r,k}^i}{\|\boldsymbol{g}_{r,k}^i\|}\right\rangle$$

$$\leq \mathbb{1}_{\bar{\mathcal{A}}_r}\left(-\mu\|\nabla f(\bar{\boldsymbol{x}}_r)\| - (1-\mu)\frac{\gamma}{\eta} + 2\sigma + 2\gamma I(AL_0 + BL_1\kappa + BL_1\rho\|\nabla f(\bar{\boldsymbol{x}}_r)\|) + (3+\mu)\left(2\sigma + \frac{\gamma}{30\eta}\right)\right)$$

$$\leq \mathbb{1}_{\bar{\mathcal{A}}_r}\left(-(\mu - 2BL_1\rho\gamma I)\|\nabla f(\bar{\boldsymbol{x}}_r)\| + \left(-\frac{9}{10} + \frac{31}{30}\mu\right)\frac{\gamma}{\eta} + (5+\mu)\sigma + 2\gamma I(AL_0 + BL_1\kappa)\right),$$

where we used Equation 26. Finally

$$A_2 = -\gamma\sum_{r=0}^{R-1}\mathbb{1}_{\bar{\mathcal{A}}_r}\left\langle \nabla f(\bar{\boldsymbol{x}}_r), \frac{1}{S}\sum_{i \in \mathcal{S}_r}\sum_{k=0}^{I-1}\frac{\boldsymbol{g}_{r,k}^i}{\|\boldsymbol{g}_{r,k}^i\|}\right\rangle$$

$$\leq \sum_{r=0}^{R-1}\mathbb{1}_{\bar{\mathcal{A}}_r}\left[-\gamma I(\mu - 2BL_1\rho\gamma I)\|\nabla f(\bar{\boldsymbol{x}}_r)\| + \left(-\frac{9}{10} + \frac{31}{30}\mu\right)\frac{\gamma^2 I}{\eta} + (5+\mu)\sigma\gamma I + 2\gamma^2 I^2(AL_0 + BL_1\kappa)\right]$$

$$\leq \sum_{r=0}^{R-1}\mathbb{1}_{\bar{\mathcal{A}}_r}\left[-\gamma I\left(\frac{3}{4} - 2BL_1\rho\gamma I\right)\|\nabla f(\bar{\boldsymbol{x}}_r)\| - \gamma I\left(\frac{1}{8}\frac{\gamma}{\eta} - 6\sigma - 2\gamma I(AL_0 + BL_1\kappa)\right)\right],$$

where we used the choice $\mu = \frac{3}{4}$.

**Bounding $A_3$.** To bound $A_3$, we begin by bounding $\mathbb{1}_{\mathcal{A}_r}\|\bar{\boldsymbol{x}}_{r+1} - \bar{\boldsymbol{x}}_r\|^2$.

$$
\mathbb{1}_{\mathcal{A}_r}\|\bar{\boldsymbol{x}}_{r+1} - \bar{\boldsymbol{x}}_r\|^2
$$

$$
= \mathbb{1}_{\mathcal{A}_r}\eta^2 \left\| \frac{1}{S}\sum_{i\in\mathcal{S}_r}\sum_{k=0}^{I-1} \boldsymbol{g}_{r,k}^i \right\|^2
$$

$$
= \mathbb{1}_{\mathcal{A}_r}\eta^2 \left\| \frac{1}{S}\sum_{i\in\mathcal{S}_r}\sum_{k=0}^{I-1} \nabla F_i(\boldsymbol{x}_{r,k}^i; \xi_{r,k}^i) - \boldsymbol{G}_r^i + \boldsymbol{G}_r \right\|^2
$$

$$
= \mathbb{1}_{\mathcal{A}_r}\eta^2 \left\| \frac{1}{S}\sum_{i\in\mathcal{S}_r}\sum_{k=0}^{I-1} (\nabla f_i(\boldsymbol{x}_{r,k}^i) - \nabla f_i(\bar{\boldsymbol{x}}_r)) + (\nabla f_i(\bar{\boldsymbol{x}}_r) - \boldsymbol{G}_r^i) + (\boldsymbol{G}_r - \nabla f(\bar{\boldsymbol{x}}_r)) + \nabla f(\bar{\boldsymbol{x}}_r) \right\|^2
$$

$$
\leq 4\eta^2 I^2 \mathbb{1}_{\mathcal{A}_r}\|\nabla f(\bar{\boldsymbol{x}}_r)\|^2 + 4\eta^2 \mathbb{1}_{\mathcal{A}_r} \left\| \frac{1}{S}\sum_{i\in\mathcal{S}_r}\sum_{k=0}^{I-1} \nabla F_i(\boldsymbol{x}_{r,k}^i; \xi_{r,k}^i) - \nabla f_i(\bar{\boldsymbol{x}}_r) \right\|^2
$$

$$
+ 4\eta^2 I^2 \mathbb{1}_{\mathcal{A}_r} \left\| \frac{1}{S}\sum_{i\in\mathcal{S}_r} \nabla f_i(\bar{\boldsymbol{x}}_r) - \boldsymbol{G}_r^i \right\|^2 + 4\eta^2 I^2 \mathbb{1}_{\mathcal{A}_r}\|\boldsymbol{G}_r - \nabla f(\bar{\boldsymbol{x}}_r)\|^2
$$

$$
\leq 4\eta^2 I^2 \mathbb{1}_{\mathcal{A}_r}\|\nabla f(\bar{\boldsymbol{x}}_r)\|^2 + 8\eta^2 \mathbb{1}_{\mathcal{A}_r} \left\| \frac{1}{S}\sum_{i\in\mathcal{S}_r}\sum_{k=0}^{I-1} \nabla f_i(\boldsymbol{x}_{r,k}^i) - \nabla f_i(\bar{\boldsymbol{x}}_r) \right\|^2
$$

$$
+ 8\eta^2 I^2 \mathbb{1}_{\mathcal{A}_r} \left\| \frac{1}{SI}\sum_{i\in\mathcal{S}_r}\sum_{k=0}^{I-1} \nabla f_i(\bar{\boldsymbol{x}}_r) - \nabla f_i(\boldsymbol{y}_{r,k}^i) \right\|^2
$$

$$
+ 8\eta^2 I^2 \mathbb{1}_{\mathcal{A}_r} \left\| \frac{1}{NI}\sum_{i=1}^{N}\sum_{k=0}^{I-1} \nabla f_i(\boldsymbol{y}_{r,k}^i) - \nabla f_i(\bar{\boldsymbol{x}}_r) \right\|^2
$$

$$
+ 8\eta^2 \mathbb{1}_{\mathcal{A}_r} \left\| \frac{1}{S}\sum_{i\in\mathcal{S}_r}\sum_{k=0}^{I-1} \boldsymbol{\epsilon}_{r,k}^i \right\|^2 + 8\eta^2 \mathbb{1}_{\mathcal{A}_r} \left\| \frac{1}{S}\sum_{i\in\mathcal{S}_r}\sum_{k=0}^{I-1} \tilde{\boldsymbol{\epsilon}}_{r,k}^i \right\|^2 + 8\eta^2 \mathbb{1}_{\mathcal{A}_r} \left\| \frac{1}{N}\sum_{i=1}^{N}\sum_{k=0}^{I-1} \tilde{\boldsymbol{\epsilon}}_{r,k}^i \right\|^2,
$$

$$(38)$$

where we denoted $\tilde{\boldsymbol{\epsilon}}_{r,k}^i = \boldsymbol{\epsilon}_{q_r^i,k}^i = \nabla F_i(\boldsymbol{y}_{r,k}^i; \xi_{q_r^i,k}^i) - \nabla f_i(\boldsymbol{y}_{r,k}^i)$. The first three terms on the RHS of Equation 38 can be bounded with similar arguments as those used previously:

$$
\mathbb{1}_{\mathcal{A}_r} \left\| \frac{1}{S}\sum_{i\in\mathcal{S}_r}\sum_{k=0}^{I-1} \nabla f_i(\boldsymbol{x}_{r,k}^i) - \nabla f_i(\bar{\boldsymbol{x}}_r) \right\|^2
$$

$$
\leq I\mathbb{1}_{\mathcal{A}_r}\frac{1}{S}\sum_{i\in\mathcal{S}_r}\sum_{k=0}^{I-1} \left\| \nabla f_i(\boldsymbol{x}_{r,k}^i) - \nabla f_i(\bar{\boldsymbol{x}}_r) \right\|^2
$$

$$
\leq I\mathbb{1}_{\mathcal{A}_r}\frac{1}{S}\sum_{i\in\mathcal{S}_r} (AL_0 + BL_1\|\nabla f_i(\bar{\boldsymbol{x}}_r)\|)^2 \sum_{k=0}^{I-1} \left\| \boldsymbol{x}_{r,k}^i - \bar{\boldsymbol{x}}_r \right\|^2
$$

$$
\overset{(i)}{\leq} 4\mathbb{1}_{\mathcal{A}_r}\eta^2 I^4 (AL_0 + BL_1\kappa + BL_1\rho\|\nabla f(\bar{\boldsymbol{x}}_r)\|)^2 \left( 2\sigma + \frac{\gamma}{\eta} \right)^2
$$

$$
\leq 8\eta^2 I^4 B^2 L_1^2 \rho^2 \left( 2\sigma + \frac{\gamma}{\eta} \right)^2 \mathbb{1}_{\mathcal{A}_r}\|\nabla f(\bar{\boldsymbol{x}}_r)\|^2 + 8\mathbb{1}_{\mathcal{A}_r}\eta^2 I^4 (AL_0 + BL_1\kappa)^2 \left( 2\sigma + \frac{\gamma}{\eta} \right)^2
$$

$$
\overset{(ii)}{\leq} \frac{1}{32}I^2 \mathbb{1}_{\mathcal{A}_r}\|\nabla f(\bar{\boldsymbol{x}}_r)\|^2 + \mathbb{1}_{\mathcal{A}_r}8\Gamma_1^2\eta^2 I^4 \left( 2\sigma + \frac{\gamma}{\eta} \right)^2,
$$

where $(i)$ holds due to Equation 13 and Assumption 1(iv); and $(ii)$ follows from the condition on $\eta$ in Equation 9. Due to Lemma 1, we can use the same argument to get

$$\mathbb{1}_{\mathcal{A}_r} \left\| \frac{1}{SI} \sum_{i \in \mathcal{S}_r} \sum_{k=0}^{I-1} \nabla f_i(\bar{\boldsymbol{x}}_r) - \nabla f_i(\boldsymbol{y}_{r,k}^i) \right\|^2 \leq \frac{1}{32} \mathbb{1}_{\mathcal{A}_r} \|\nabla f(\bar{\boldsymbol{x}}_r)\|^2 + \mathbb{1}_{\mathcal{A}_r} 8\Gamma_1^2 \eta^2 I^2 \left( 2\sigma + \frac{\gamma}{\eta} \right)^2,$$

and

$$\mathbb{1}_{\mathcal{A}_r} \left\| \frac{1}{NI} \sum_{i=1}^{N} \sum_{k=0}^{I-1} \nabla f_i(\bar{\boldsymbol{x}}_r) - \nabla f_i(\boldsymbol{y}_{r,k}^i) \right\|^2 \leq \frac{1}{32} \mathbb{1}_{\mathcal{A}_r} \|\nabla f(\bar{\boldsymbol{x}}_r)\|^2 + \mathbb{1}_{\mathcal{A}_r} 8\Gamma_1^2 \eta^2 I^2 \left( 2\sigma + \frac{\gamma}{\eta} \right)^2.$$

Plugging back into Equation 38,

$$\mathbb{1}_{\mathcal{A}_r} \|\bar{\boldsymbol{x}}_{r+1} - \bar{\boldsymbol{x}}_r\|^2 \leq 5\eta^2 I^2 \mathbb{1}_{\mathcal{A}_r} \|\nabla f(\bar{\boldsymbol{x}}_r)\|^2 + \mathbb{1}_{\mathcal{A}_r} 192\Gamma_1^2 \eta^4 I^4 \left( 2\sigma + \frac{\gamma}{\eta} \right)^2$$

$$+ 8\eta^2 \mathbb{1}_{\mathcal{A}_r} \left\| \frac{1}{S} \sum_{i \in \mathcal{S}_r} \sum_{k=0}^{I-1} \boldsymbol{\epsilon}_{r,k}^i \right\|^2 + 8\eta^2 \mathbb{1}_{\mathcal{A}_r} \left\| \frac{1}{S} \sum_{i \in \mathcal{S}_r} \sum_{k=0}^{I-1} \tilde{\boldsymbol{\epsilon}}_{r,k}^i \right\|^2$$

$$+ 8\eta^2 \mathbb{1}_{\mathcal{A}_r} \left\| \frac{1}{N} \sum_{i=1}^{N} \sum_{k=0}^{I-1} \tilde{\boldsymbol{\epsilon}}_{r,k}^i \right\|^2. \tag{39}$$

We can bound the remaining terms using the concentration inequality from Lemma 7. We write the index of $\mathcal{S}_r$ as $\{i_1, ..., i_S\}$ For $k = 0, ..., I-1$ and for $j \in [S]$, let $X_{kS+j} = \varepsilon_{r,k}^{i_j}$ and $\mathcal{H}_{kS+j} = \sigma\left( \mathcal{F}_r, \mathcal{S}_r, \{\xi_{r,\ell}^i\}_{0 \leq \ell \leq k-1, i \in \mathcal{S}_r}, \{\xi_{r,k}^{i_m}\}_{m \leq j} \right)$ for $k \geq 1$, where $\{\xi_{r,\ell}^i\}_{0 \leq \ell \leq -1, i \in \mathcal{S}_r} = \varnothing$. For any $t = kS + j$, it holds that

$$\mathbb{E}\left[ X_t \mid \mathcal{H}_{t-1} \right] = \mathbb{E}\left[ \varepsilon_{r,k}^{i_j} \mid \mathcal{H}_{kS+j-1} \right] = \mathbb{E}\left[ \varepsilon_{r,k}^{i_j} \mid \mathcal{F}_r, \{\xi_{r,\ell}^{i_j}\}_{\ell \leq k-1} \right] = 0,$$

where the second inequality holds due to the independence between different clients given $\mathcal{F}_r$. Then $\{X_t\}_{t \leq IS}$ is a martingale difference sequence with respect to $\{\mathcal{H}_t\}_{t \leq IS}$. Also, $\|X_t\| \leq \sigma$ almost surely for all $t \in [IS]$ and $\mathbb{E}_{t-1}[\|X_t\|] \leq \sigma^2$. Therefore, by Lemma 7, with probability [5] at least $1 - \delta$,

$$\left\| \frac{1}{S} \sum_{i \in \mathcal{S}_r} \sum_{k=0}^{I-1} \boldsymbol{\epsilon}_{r,k}^i \right\|^2 \leq \frac{1}{S^2} \left( 3\sigma \log \frac{1}{\delta} + 3\sqrt{SI\sigma^2 \log \frac{1}{\delta}} \right)^2$$

$$\leq \frac{9\sigma^2 \log^2 \frac{1}{\delta}}{S^2} + \frac{9I\sigma^2 \log \frac{1}{\delta}}{S}$$

$$\leq \frac{9\sigma^2 I \log \frac{1}{\delta}}{S} \left( \frac{\log \frac{1}{\delta}}{SI} + 1 \right)$$

$$\leq \frac{18\sigma^2 I \log^2 \frac{1}{\delta}}{S}.$$

Without loss of generality, we assume $\log(1/\delta) \geq 1$. We can apply the exact same argument to bound the remaining noise terms:

$$\left\| \frac{1}{S} \sum_{i \in \mathcal{S}_r} \sum_{k=0}^{I-1} \tilde{\boldsymbol{\epsilon}}_{r,k}^i \right\|^2 \leq \frac{18\sigma^2 I \log \frac{1}{\delta}}{S},$$

and

$$\left\| \frac{1}{N} \sum_{i=1}^{N} \sum_{k=0}^{I-1} \tilde{\boldsymbol{\epsilon}}_{r,k}^i \right\|^2 \leq \frac{18\sigma^2 I \log \frac{1}{\delta}}{N} \leq \frac{18\sigma^2 I \log \frac{1}{\delta}}{S},$$

---

[5]In fact, we first verify that the bound holds with probability at least $1 - \delta$ given $\mathcal{F}_r$ and $\mathcal{S}_r$. Since the upper bound does not depend on $\mathcal{F}_r$ and $\mathcal{S}_r$, we can conclude the bound holds with the same unconditional probability.

each with probability $1 - 3\delta$. Plugging back into Equation 39,

$$\mathbb{1}_{\mathcal{A}_r}\|\bar{\boldsymbol{x}}_{r+1} - \bar{\boldsymbol{x}}_r\|^2 \leq \mathbb{1}_{\mathcal{A}_r}\left(5\eta^2 I^2\|\nabla f(\bar{\boldsymbol{x}}_r)\|^2 + 192\Gamma_1^2\eta^4 I^4\left(2\sigma + \frac{\gamma}{\eta}\right)^2 + \frac{432\sigma^2\eta^2 I \log\frac{1}{\delta}}{S}\right),$$

with probability $1 - 9\delta$. Also, using Equation 25,

$$\mathbb{1}_{\mathcal{A}_r}(AL_0 + BL_1\|\nabla f(\bar{\boldsymbol{x}}_r)\|) \leq \mathbb{1}_{\mathcal{A}_r}\left(AL_0 + 4BL_1\left(2\sigma + \frac{\gamma}{\eta}\right)\right) \leq \mathbb{1}_{\mathcal{A}_r}\Gamma_1.$$

Finally,

$$A_3 = \frac{1}{2}\sum_{r=0}^{R-1}\mathbb{1}_{\mathcal{A}_r}(AL_0 + BL_1\|\nabla f(\bar{\boldsymbol{x}}_r)\|)\|\bar{\boldsymbol{x}}_{r+1} - \bar{\boldsymbol{x}}_r\|^2$$

$$\leq \sum_{r=0}^{R-1}\mathbb{1}_{\mathcal{A}_r}\left(\frac{5}{2}\Gamma_1\eta^2 I^2\|\nabla f(\bar{\boldsymbol{x}}_r)\|^2 + 96\Gamma_1^3\eta^4 I^4\left(2\sigma + \frac{\gamma}{\eta}\right)^2 + 216(AL_0 + BL_1\|\nabla f(\bar{\boldsymbol{x}}_r)\|)\frac{\sigma^2\eta^2 I \log\frac{1}{\delta}}{S}\right)$$

$$\leq \sum_{r=0}^{R-1}\mathbb{1}_{\mathcal{A}_r}\left[\frac{5}{8}\eta I\|\nabla f(\bar{\boldsymbol{x}}_r)\|^2 + \frac{216 BL_1\sigma^2\eta^2 I \log\frac{1}{\delta}}{S}\|\nabla f(\bar{\boldsymbol{x}}_r)\|\right.$$

$$\left. + 96\Gamma_1^3\eta^4 I^4\left(2\sigma + \frac{\gamma}{\eta}\right)^2 + \frac{216 AL_0\sigma^2\eta^2 I \log\frac{1}{\delta}}{S}\right],$$

with probability $1 - 9\delta$.

**Bounding $A_4$:** Due to normalization of the update $\boldsymbol{g}_{r,k}^i$ under $\bar{\mathcal{A}}_r$, we have

$$A_4 = \gamma^2\sum_{r=0}^{R-1}\mathbb{1}_{\bar{\mathcal{A}}_r}\frac{AL_0 + BL_1\|\nabla f(\bar{\boldsymbol{x}}_r)\|}{2}\left\|\frac{1}{S}\sum_{i\in\mathcal{S}_r}\sum_{k=0}^{I-1}\frac{\boldsymbol{g}_{r,k}^i}{\|\boldsymbol{g}_{r,k}^i\|}\right\|^2$$

$$\leq \gamma^2 I^2\sum_{r=0}^{R-1}\mathbb{1}_{\bar{\mathcal{A}}_r}\frac{AL_0 + BL_1\|\nabla f(\bar{\boldsymbol{x}}_r)\|}{2}.$$

Combining the respective bounds for $A_1, A_2, A_3, A_4$ into Equation 28,

$f(\bar{\boldsymbol{x}}_R) - f(\bar{\boldsymbol{x}}_0)$

$$\leq \sum_{r=0}^{R-1}\mathbb{1}_{\mathcal{A}_r}\left[-\frac{1}{8}\eta I\|\nabla f(\bar{\boldsymbol{x}}_r)\|^2 + \left(2\Gamma_1\eta^2 I^2\left(2\sigma + \frac{\gamma}{\eta}\right) + \frac{216 BL_1\sigma^2\eta^2 I \log\frac{1}{\delta}}{S}\right)\|\nabla f(\bar{\boldsymbol{x}}_r)\|\right.$$

$$\left. + 96\Gamma_1^3\eta^4 I^4\left(2\sigma + \frac{\gamma}{\eta}\right)^2 + \frac{216 AL_0\sigma^2\eta^2 I \log\frac{1}{\delta}}{S}\right]$$

$$+ \sum_{r=0}^{R-1}\mathbb{1}_{\bar{\mathcal{A}}_r}\left[-\gamma I\left(\frac{3}{4} - 2BL_1\rho\gamma I - \frac{1}{2}BL_1\gamma I\right)\|\nabla f(\bar{\boldsymbol{x}}_r)\| - \gamma I\left(\frac{1}{8}\frac{\gamma}{\eta} - 6\sigma - 2AL_0\gamma I - \frac{5}{2}BL_1\kappa\gamma I\right)\right]$$

$$+ \eta\left(12\sigma^2 + 64I\left(\kappa + 5\rho\left(2\sigma + \frac{\gamma}{\eta}\right)\right)\right)^2\left(\frac{1}{S} - \frac{1}{N}\right)\log\frac{1}{\delta},$$

which is the desired result. Note that the bound on $A_1$ holds with probability $1 - 5\delta$ and the bound on $A_2$ holds with probability $1 - 9\delta$, and we initially supposed the event $\mathcal{E}$, which holds with probability $1 - \delta$. So the overall result holds with probability at least $1 - 15\delta$. $\qquad\square$

## A.5  Proof of Theorem 1

**Theorem 1 restated.** Let $\epsilon \leq \frac{AL_0}{16BL_1\rho}$ and $\delta \in (0,1)$. Denote $K = \left\lceil\frac{\log(RN/\delta)}{\log(N/(N-S))}\right\rceil$, $\Gamma_1 := AL_0 + BL_1\kappa + 4BL_1\rho\left(2\sigma + \frac{\gamma}{\eta}\right)$ and $\Gamma_2 := 64\left(\kappa + 5\rho\left(2\sigma + \frac{\gamma}{\eta}\right)\right)^2\left(\frac{1}{S} - \frac{1}{N}\right)$. If

$$\eta \leq \min\left\{\frac{1}{90(K+1)\Gamma_1 I}, \frac{\epsilon}{32\Gamma_1 I\left(74\sigma + \frac{AL_0}{BL_1\rho}\right)}, \frac{S\epsilon^2}{216 AL_0\sigma^2\log\frac{1}{\delta}}, \frac{\Delta}{\log\frac{1}{\delta}}\min\left\{\frac{1}{12\sigma^2}, \frac{1}{\Gamma_2 I}\right\}\right\},$$

and $\gamma = \left(72\sigma + \frac{AL_0}{BL_1\rho}\right)\eta$, then Algorithm 1 satisfies $\frac{1}{R}\sum_{r=0}^{R-1}\|\nabla f(\bar{\boldsymbol{x}}_r)\| \leq 35\epsilon$ with probability at least $1 - 15\delta$, as long as $R \geq \frac{8\Delta}{\epsilon^2\eta I}$.

*Proof.* First, under our choice of $\eta$ and $\gamma$,

$$(K+1)\Gamma_1\eta I \leq \frac{1}{60},$$

and

$$2(K+1)\eta I\left(2\sigma + \frac{\gamma}{\eta}\right) \leq \frac{\left(2\sigma + \frac{\gamma}{\eta}\right)}{30\Gamma_1} = \frac{\left(2\sigma + \frac{\gamma}{\eta}\right)}{30(AL_0 + BL_1\kappa + 4BL_1\rho\left(2\sigma + \frac{\gamma}{\eta}\right)} \leq \frac{1}{120BL_1\rho} \leq \frac{1}{120L_1},$$

where the last line follows from $B, \rho \geq 1$. Therefore Equation 9 holds under our choice of $\eta$ and $\gamma$, so the condition of Lemma 12 is satisfied. Denoting

$$U(\boldsymbol{x}) = -\gamma I\left(\frac{3}{4} - 2BL_1\rho\gamma I - \frac{1}{2}BL_1\gamma I\right)\|\nabla f(\boldsymbol{x})\| - \gamma I\left(\frac{1}{8}\frac{\gamma}{\eta} - 6\sigma - 2AL_0\gamma I - \frac{5}{2}BL_1\kappa\gamma I\right),$$

and

$$V(\boldsymbol{x}) = -\frac{1}{8}\eta I\|\nabla f(\boldsymbol{x})\|^2 + \left(2\Gamma_1\eta^2 I^2\left(2\sigma + \frac{\gamma}{\eta}\right) + \frac{216BL_1\sigma^2\eta^2 I\log\frac{1}{\delta}}{S}\right)\|\nabla f(\boldsymbol{x})\| +$$

$$96\Gamma_1^3\eta^4 I^4\left(2\sigma + \frac{\gamma}{\eta}\right)^2 + \frac{216AL_0\sigma^2\eta^2 I\log\frac{1}{\delta}}{S},$$

Lemma 12 gives us

$$f(\bar{\boldsymbol{x}}_R) - f(\bar{\boldsymbol{x}}_0) \leq \sum_{r=0}^{R-1}\left[\mathbb{1}_{\bar{\mathcal{A}}_r}U(\bar{\boldsymbol{x}}_r) + \mathbb{1}_{\mathcal{A}_r}V(\bar{\boldsymbol{x}}_r)\right] + 12\eta\sigma^2\log\frac{1}{\delta} + \Gamma_2\eta I\log\frac{1}{\delta}. \tag{40}$$

We will bound each of $\mathbb{1}_{\bar{\mathcal{A}}_r}U(\bar{\boldsymbol{x}}_r)$ and $\mathbb{1}_{\mathcal{A}_r}V(\bar{\boldsymbol{x}}_r)$ under our choices of $\gamma$ and $\eta$.

To bound $U(\bar{\boldsymbol{x}}_r)$, notice

$$-\frac{3}{4} + 2BL_1\rho\gamma I + \frac{1}{2}BL_1\gamma I \leq -\frac{3}{4} + \frac{5}{2}BL_1\rho\gamma I$$

$$\leq -\frac{3}{4} + \frac{5}{2}\eta IBL_1\rho\frac{\gamma}{\eta}$$

$$= -\frac{3}{4} + \frac{5}{2}\eta IBL_1\rho\left(72\sigma + \frac{AL_0}{BL_1\rho}\right)$$

$$= -\frac{3}{4} + \frac{5}{2}\eta I\left(72BL_1\rho\sigma + AL_0\right)$$

$$= -\frac{3}{4} + 45\Gamma_1\eta I$$

$$\leq -\frac{3}{4} + \frac{1}{2}$$

$$\leq -\frac{1}{4},$$

and

$$-\frac{1}{8}\frac{\gamma}{\eta} + 6\sigma + 2AL_0\gamma I + \frac{5}{2}BL_1\kappa\gamma I = -\frac{1}{8}\frac{\gamma}{\eta} + 6\sigma + \eta I\frac{\gamma}{\eta}\left(2AL_0 + \frac{5}{2}BL_1\kappa\right)$$

$$\leq -\frac{1}{8}\frac{\gamma}{\eta} + 6\sigma + 2\Gamma_1\eta I\frac{\gamma}{\eta}$$

$$\leq \left(-\frac{1}{8} + \frac{1}{30}\right)\frac{\gamma}{\eta} + 6\sigma$$

$$\leq -\frac{1}{12}\left(72\sigma + \frac{AL_0}{BL_1\rho}\right) + 6\sigma$$

$$\leq 0.$$

So

$$U(\bar{\boldsymbol{x}}_r) = -\frac{1}{4}\gamma I \|\nabla f(\bar{\boldsymbol{x}}_r)\| \le -\frac{1}{4}\epsilon \eta I \|\nabla f(\bar{\boldsymbol{x}}_r)\|,$$

where the last inequality holds due to the fact $\epsilon \le \frac{AL_0}{16BL_1\rho} \le 72\sigma + \frac{AL_0}{BL_1\rho} = \frac{\gamma}{\eta}$. To bound $V(\bar{\boldsymbol{x}}_r)$, notice

$$2\Gamma_1 \eta^2 I^2 \left(2\sigma + \frac{\gamma}{\eta}\right) \le \frac{1}{16}\epsilon \eta I,$$

since $\eta \le \frac{\epsilon}{32\Gamma_1 I\left(74\sigma + \frac{AL_0}{BL_1\rho}\right)} = \frac{\epsilon}{32\Gamma_1 I\left(2\sigma + \frac{\gamma}{\eta}\right)}$, and

$$
\begin{aligned}
\frac{216BL_1\sigma^2\eta^2 I \log\frac{1}{\delta}}{S} &= \frac{BL_1}{AL_0}\eta I \frac{216AL_0\sigma^2\eta \log\frac{1}{\delta}}{S} \\
&\le \frac{BL_1}{AL_0}\epsilon^2 \eta I \\
&\le \frac{1}{16}\epsilon \eta I,
\end{aligned}
$$

where the last line uses $\epsilon \le \frac{AL_0}{16BL_1\rho} \le \frac{AL_0}{16BL_1}$. Lastly,

$$
\begin{aligned}
96\Gamma_1^3\eta^4 I^4 &\left(2\sigma + \frac{\gamma}{\eta}\right)^2 + \frac{216AL_0\sigma^2\eta^2 I \log\frac{1}{\delta}}{S} \\
&\le \eta I \left(96\Gamma_1^3\eta^3 I^3 \left(2\sigma + \frac{\gamma}{\eta}\right)^2 + \frac{216AL_0\sigma^2\eta \log\frac{1}{\delta}}{S}\right) \\
&\le \eta I \left(\frac{8}{5}\epsilon^2 + \epsilon^2\right) \\
&\le 3\epsilon^2 \eta I.
\end{aligned}
$$

So

$$
\begin{aligned}
V(\bar{\boldsymbol{x}}_r) &= -\frac{1}{8}\eta I \|\nabla f(\bar{\boldsymbol{x}}_r)\|^2 + \frac{1}{8}\epsilon \eta I \|\nabla f(\bar{\boldsymbol{x}}_r)\| + 3\epsilon^2 \eta I \\
&\le -\frac{1}{8}\epsilon \eta I \|\nabla f(\bar{\boldsymbol{x}}_r)\| + 4\epsilon^2 \eta I,
\end{aligned}
$$

where the last line came from the inequality $x^2 \ge 2ax + a^2$ with $x = \|\nabla f(\bar{\boldsymbol{x}}_r)\|$ and $a = \epsilon$. Combining the bounds of $U(\bar{\boldsymbol{x}}_r)$ and $V(\bar{\boldsymbol{x}}_r)$,

$$\max\{U(\bar{\boldsymbol{x}}_r), V(\bar{\boldsymbol{x}}_r)\} \le -\frac{1}{8}\epsilon \eta I \|\nabla f(\bar{\boldsymbol{x}}_r)\| + 4\epsilon^2 \eta I.$$

Plugging this into Equation 40 yields

$$f(\bar{\boldsymbol{x}}_R) - f(\bar{\boldsymbol{x}}_0) \le -\frac{1}{8}\epsilon \eta I \sum_{r=0}^{R-1} \|\nabla f(\bar{\boldsymbol{x}}_r)\| + 4\epsilon^2 \eta R I + 12\eta \sigma^2 \log\frac{1}{\delta} + \Gamma_2 \eta I \log\frac{1}{\delta},$$

and by rearranging we have

$$
\begin{aligned}
\frac{1}{R}\sum_{r=0}^{R-1} \|\nabla f(\bar{\boldsymbol{x}}_r)\| &\le \frac{8(f(\bar{\boldsymbol{x}}_0) - f(\bar{\boldsymbol{x}}_R))}{\epsilon \eta R I} + 32\epsilon + \frac{96\sigma^2 \log\frac{1}{\delta}}{\epsilon R I} + \frac{8\Gamma_2 \log\frac{1}{\delta}}{\epsilon R} \\
&\le \frac{8\Delta}{\epsilon \eta R I} + 32\epsilon + \frac{96\sigma^2 \log\frac{1}{\delta}}{\epsilon R I} + \frac{8\Gamma_2 \log\frac{1}{\delta}}{\epsilon R} \\
&\le 33\epsilon + \frac{96\sigma^2 \log\frac{1}{\delta}}{\epsilon R I} + \frac{8\Gamma_2 \log\frac{1}{\delta}}{\epsilon R}, \quad\quad\quad\quad (41)
\end{aligned}
$$

where the last line comes from $R \geq \frac{8\Delta}{\epsilon^2 \eta I}$. Finally, $\eta \leq \frac{\Delta}{\log \frac{1}{\delta}} \min\left\{\frac{1}{12\sigma^2}, \frac{1}{\Gamma_2 I}\right\}$ implies

$$R \geq \frac{8\Delta}{\epsilon^2 \eta I} \geq \frac{96\sigma^2 \log \frac{1}{\delta}}{\epsilon^2 I},$$

and

$$R \geq \frac{8\Delta}{\epsilon^2 \eta I} \geq \frac{8\Gamma_2 \log \frac{1}{\delta}}{\epsilon^2},$$

so

$$\frac{96\sigma^2 \log \frac{1}{\delta}}{\epsilon R I} + \frac{8\Gamma_2 \log \frac{1}{\delta}}{\epsilon R} \leq 2\epsilon.$$

Plugging into Equation 41 gives the result. $\qquad\square$

### A.6 Proof of Corollary 1

**Corollary 1 restated.** If, under the setting of Theorem 1, we additionally choose

$$\eta = \min\left\{\frac{1}{90(K+1)\Gamma_1 I}, \frac{\epsilon}{32\Gamma_1 I\left(74\sigma + \frac{AL_0}{BL_1\rho}\right)}, \frac{S\epsilon^2}{216 AL_0 \sigma^2 \log \frac{1}{\delta}}, \frac{\Delta}{\log \frac{1}{\delta}}\min\left\{\frac{1}{12\sigma^2}, \frac{1}{\Gamma_2 I}\right\}\right\},$$

and

$$I \leq \frac{27 AL_0 \sigma^2 \log \frac{1}{\delta}}{4\epsilon \Gamma_1 S\left(74\sigma + \frac{AL_0}{BL_1\rho}\right)},$$

with

$$\epsilon \leq \min\left\{\frac{16\left(74\sigma + \frac{AL_0}{BL_1\rho}\right)}{45(K+1)}, \sqrt{\frac{18\Delta AL_0}{S}}, \frac{32\Delta\Gamma_1\left(74\sigma + \frac{AL_0}{BL_1\rho}\right)}{\Gamma_2 \log \frac{1}{\delta}}\right\},$$

then with probability $1 - 14\delta$, Algorithm 1 will reach an $\epsilon$-stationary point with iteration complexity

$$RI = O\left(\frac{\Delta L_0 \sigma^2 \log \frac{1}{\delta}}{S\epsilon^4}\right).$$

*Proof.* From the condition on $\epsilon$,

$$\epsilon \leq \frac{16\left(74\sigma + \frac{AL_0}{BL_1\rho}\right)}{45(K+1)}$$

$$\frac{\epsilon}{32\Gamma_1 I\left(74\sigma + \frac{AL_0}{BL_1\rho}\right)} \leq \frac{1}{90(K+1)\Gamma_1 I}.$$

From the condition on $I$,

$$\frac{\epsilon}{32\Gamma_1 I\left(74\sigma + \frac{AL_0}{BL_1\rho}\right)} \geq \frac{\epsilon}{32\Gamma_1\left(74\sigma + \frac{AL_0}{BL_1\rho}\right)} \frac{4\epsilon \Gamma_1 S\left(74\sigma + \frac{AL_0}{BL_1\rho}\right)}{27 AL_0 \sigma^2 \log \frac{1}{\delta}} = \frac{S\epsilon^2}{216 AL_0 \sigma^2 \log \frac{1}{\delta}}$$

Also

$$\frac{S\epsilon^2}{216 AL_0 \sigma^2 \log \frac{1}{\delta}} \leq \frac{S}{216 AL_0 \sigma^2 \log \frac{1}{\delta}} \frac{18\Delta AL_0}{S} = \frac{\Delta}{12\sigma^2 \log \frac{1}{\delta}},$$

and

$$\frac{S\epsilon^2}{216 AL_0 \sigma^2 \log \frac{1}{\delta}} \leq \frac{S}{216 AL_0 \sigma^2 \log \frac{1}{\delta}} \frac{32\Delta\Gamma_1\left(74\sigma + \frac{AL_0}{BL_1\rho}\right)}{\Gamma_2 \log \frac{1}{\delta}} \frac{27 AL_0 \sigma^2 \log \frac{1}{\delta}}{4\Gamma_1 IS\left(74\sigma + \frac{AL_0}{BL_1\rho}\right)}$$

$$\leq \frac{\Delta}{\Gamma_2 I \log \frac{1}{\delta}}.$$

Therefore $\eta = \frac{S\epsilon^2}{216AL_0\sigma^2 \log \frac{1}{\delta}}$. Choosing $R = \frac{8\Delta}{\epsilon^2 \eta I}$,

$$RI = \frac{8\Delta}{\epsilon^2 \eta} = \frac{1296A\Delta L_0\sigma^2 \log \frac{1}{\delta}}{S\epsilon^4} = O\left(\frac{\Delta L_0\sigma^2 \log \frac{1}{\delta}}{S\epsilon^4}\right).$$

$\square$

## B    Deferred Proofs of the Lower Bound

---

**Algorithm 2** Clipped Minibatch SGD

---

1:  Initialize $\boldsymbol{x}_0$
2:  **for** $r = 0, 1, \ldots, R - 1$ **do**
3:      Sample $\mathcal{S}_r \subset [N]$ uniformly at random such that $|\mathcal{S}_r| = S$
4:      $\boldsymbol{g}_r = \frac{1}{SI} \sum_{i \in \mathcal{S}_r} \sum_{k=0}^{I-1} \nabla F_i(\boldsymbol{x}_r, \xi_{r,k}^i)$
5:      Update $\boldsymbol{x}_{r+1} \leftarrow \boldsymbol{x}_r - \min\left(\eta, \frac{\gamma}{\|\boldsymbol{g}_r\|}\right)\boldsymbol{g}_r$
6:  **end for**

---

Our proof of Theorem 2 analyzes three separate functions satisfying Assumption 1, and concludes that, in order for clipped minibatch SGD to avoid divergence on the first two functions, one must choose parameters $\gamma, \eta$ such that convergence on the third function is slow. These three functions are separately analyzed in Lemmas 13, 15, and 16. Each lemma assumes a particular value of the initial point $\boldsymbol{x}_0$: this assumption can be made without loss of generality, since for the initialization $\boldsymbol{x}_0$ we may always translate each function to simulate the assumed initialization.

**Lemma 13.** *Let $\delta \in (0, 1)$ and denote $Q = \left\lfloor \frac{\kappa + (\rho-1)M}{\kappa + (\rho+1)M} N \right\rfloor$. Suppose*

$$\rho > \frac{2 + \log(2-\delta)}{\log(2-\delta)}, \quad N \geq \frac{(\rho+1)(1 + \log(2-\delta))}{(\rho-1)\log(2-\delta) - 2}, \quad 1 \leq S \leq \frac{\log(2-\delta)(Q+1)}{N - Q + \log(2-\delta)}, \quad (42)$$

*and $\frac{\gamma}{\eta} \leq M$. For any $0 < \epsilon < M$, there exists a problem instance $\{f_i\}_{i=1}^N \in \mathcal{F}(L_0, L_1, M, \kappa, \rho, N)$ such that, with probability at least $\delta$, clipped minibatch SGD with parameters $\gamma, \eta, S$ will generate iterates $\{\boldsymbol{x}_r\}_{r=0}^R$ with $\|\nabla f(\boldsymbol{x}_r)\| > \epsilon$ for all $r$.*

The proof of Lemma 13 will require the following lemma.

**Lemma 14** (Lemma 5.3 in [20])**.** *Let $\{X_t\}_{t=1}^\infty$ be a Markov chain over states $\{i\}_{i=0}^\infty$, such that 0 is an absorbing state, and the transition distribution elsewhere is as follows:*

$$X_{t+1}|\{X_t = i\} = \begin{cases} i - 1 & \text{w.p.} \quad p \\ i + 1 & \text{w.p.} \quad 1 - p \end{cases}$$

*Define the absorb probabilities $\alpha_i := \mathbb{P}(\exists t > 0 : X_t = 0 \mid X_0 = i)$, then:*

$$\alpha_i = \left(\frac{p}{1-p}\right)^i, \quad \forall i \geq 1$$

*Proof of Lemma 13.* Define

$$\phi = \frac{Q}{N}, \quad b = \frac{1+\phi}{1-\phi}M,$$

and

$$v = \frac{3b}{2L_0}, \quad g(x) = -\frac{1}{8v^3}x^4 + \frac{3}{4v}x^2.$$

Also, define

$$\ell_1(x) = \begin{cases} M(x+v) - Mg(v) & x \leq -v \\ -Mg(x) & x \in (-v, v) \\ -M(x-v) - Mg(v) & x \geq v \end{cases}$$

and

$$\ell_2(x) = \begin{cases} -b(x+v) + bg(v) & x \le -v \\ bg(x) & x \in (-v, v) \\ b(x-v) + bg(v) & x \ge v \end{cases}$$

Consider the problem instance defined by

$$f_i(x) = \begin{cases} \ell_1(x) & 1 \le i \le Q \\ \ell_2(x) & Q+1 \le i \le N \end{cases}$$

with global objective

$$f(x) = \phi\ell_1(x) + (1-\phi)\ell_2(x) = \begin{cases} -M(x+v) + Mg(v) & x \le -v \\ Mg(x) & x \in (-v, v) \\ M(x-v) + Mg(v) & x \ge v \end{cases}$$

Note that $b > M$. For this problem instance, we define the stochastic objective as $F_i(x; \xi) = f_i(x)$, so that the true gradient of each local objective is returned by each gradient query of the algorithm.

By construction, each local objective $f_i$ and the global objective $f$ is twice continuously differentiable, with gradient bounded by $M$. Also, each objective is $(L_0, L_1)$-smooth (in fact, they are $L_0$-smooth), since for $|x| \le v$,

$$\begin{aligned} |g''(x)| &= \left| -\frac{3}{2v^3}x^2 + \frac{3}{2v} \right| \\ &= \frac{3}{2v} \left| 1 - \frac{x^2}{v^2} \right| \\ &\le \frac{3}{2v} = \frac{L_0}{b}, \end{aligned}$$

and for $|x| \ge v$, $|f_i''(x)| = 0$. Finally, the collection of objectives satisfies the heterogeneity condition. For $i \le Q$,

$$|f_i'(x)| = |f'(x)| \le \kappa + \rho|f'(x)|,$$

and for $i \ge Q+1$,

$$|f_i'(x)| = b\frac{|f'(x)|}{M} \le \frac{1+\phi}{1-\phi}M\frac{|f'(x)|}{M} \le (\kappa + \rho M)\frac{|f'(x)|}{M} \le \kappa + \rho|f'(x)|,$$

where we used

$$\frac{1+\phi}{1-\phi} \le \frac{1 + \frac{\kappa+(\rho-1)M}{\kappa+(\rho+1)M}}{1 - \frac{\kappa+(\rho-1)M}{\kappa+(\rho+1)M}} = \frac{2\kappa + 2\rho M}{\kappa + (\rho+1)M}\frac{\kappa + (\rho+1)M}{2M} = \frac{\kappa + \rho M}{M},$$

together with the fact $\phi \le \frac{\kappa+(\rho-1)M}{\kappa+(\rho+1)M}$. Therefore $\{f_i\}_{i=1}^N \in \mathcal{F}(L_0, L_1, M, \kappa, \rho, N)$.

Now, we analyze the behavior of clipped minibatch SGD on the instance $\{f_i\}_{i=1}^N$ with initialization $x_0 = v + 1$. Let $r_0$ be the index of the first round (if one exists) in which $x_r < v$. For each round $r$, define the event $B_r = \{\mathcal{S}_r \subset [Q]\}$. To compute $\mathbb{P}(B_r)$, first notice

$$\frac{N-Q}{Q-S+1} \le \frac{\log(2-\delta)}{S}.$$

Therefore,

$$\mathbb{P}(B_r) = \prod_{i=0}^{S-1} \frac{Q-i}{N-i} \ge \left( \frac{Q-S+1}{N-S+1} \right)^S = \frac{1}{\left(1 + \frac{N-Q}{Q-S+1}\right)^S} \ge \frac{1}{\left(1 + \frac{\log(2-\delta)}{S}\right)^S} \ge \frac{1}{e^{\log(2-\delta)}} = \frac{1}{2-\delta},$$

where we used the inequality $\left(1 + \frac{a}{x}\right)^x \le e^a$.

Now define the auxiliary sequence $\{y_r\}_{r=0}^R$ as follows:

$$y_0 = x_0$$

$$y_{r+1} = \begin{cases} y_r + \gamma & \text{if } B_r \\ y_r - \gamma & \text{otherwise} \end{cases}$$

We claim that $y_r \leq x_r$ for all $r \leq r_0$. For any $r < r_0$, if $B_r$ occurs then $y_{r+1} - y_r = x_{r+1} - x_r$, since in this case $g_r = -M$ and $x_{r+1} - x_r = \gamma$, due to the fact that $\frac{\gamma}{\eta} \leq M$. If $B_r$ does not occur, then $x_{r+1} - x_r \geq -\gamma$, due to the clipping operation, so that $y_{r+1} - y_r \leq x_{r+1} - x_r$. Therefore for all $r \leq r_0$,

$$y_r = y_0 + \sum_{q=0}^{r-1} y_{q+1} - y_q \leq x_0 + \sum_{q=0}^{r-1} x_{q+1} - x_q = x_r,$$

which proves our claim. Denoting $D = \{|f'(x_r)| \geq M \text{ for all } r \geq 1\}$,

$$\mathbb{P}(D) \geq \mathbb{P}(x_r \geq -1 \text{ for all } r \geq 1)$$
$$\geq \mathbb{P}(y_r \geq -1 \text{ for all } r \geq 1)$$
$$\overset{(i)}{\geq} 1 - \left(\frac{1 - \mathbb{P}(B_r)}{\mathbb{P}(B_r)}\right)^{\lceil \frac{x_0 - v}{\gamma} \rceil}$$
$$\overset{(ii)}{\geq} 1 - \frac{1 - \mathbb{P}(B_r)}{\mathbb{P}(B_r)}$$
$$= \frac{2\mathbb{P}(B_r) - 1}{\mathbb{P}(B_r)}$$
$$\geq \left(\frac{2}{2 - \delta} - 1\right)(2 - \delta)$$
$$= \delta,$$

where $(i)$ comes from applying Lemma 14 to the sequence $y_r$ and $(ii)$ uses $\lceil \frac{x_0 - v}{\gamma} \rceil \geq 1$ and $\frac{1 - \mathbb{P}(B_r)}{\mathbb{P}(B_r)} \leq 1 - \delta$. $\qquad\square$

The following Lemma uses the same function as in the first half of the proof of Theorem 2 in [8]. However, the sequence of iterates computed by clipped minibatch SGD (our setting) evolves differently than that of GD (their setting).

**Lemma 15.** *Suppose $\frac{\gamma}{\eta} > M$, $M > \frac{L_0}{L_1}$, and $\eta \geq \frac{2}{L_1 M}\left(1 + \log \frac{L_1 M}{L_0}\right)$. For any $0 < \epsilon < M$, there exists a problem instance $\{f_i\}_{i=1}^N \in \mathcal{F}(L_0, L_1, M, \kappa, \rho, N)$ such that clipped minibatch SGD with parameters $\gamma, \eta$, and any $S$ will generate iterates $\{x_r\}_{r=0}^R$ with $\|\nabla f(x_r)\| > \epsilon$ for all $r$.*

*Proof.* Consider the following function,

$$f(x) = \begin{cases} \frac{L_0}{L_1^2} \exp\left(-L_1 x - 1\right) & x < -\frac{1}{L_1} \\ \frac{L_0}{2} x^2 + \frac{L_0}{2L_1^2} & x \in \left[-\frac{1}{L_1}, \frac{1}{L_1}\right] \\ \frac{L_0}{L_1^2} \exp\left(L_1 x - 1\right) & x > \frac{1}{L_1} \end{cases}$$

and the problem instance $f_i = f$ for all $i \in [N]$, with the initialization $x_0 = \frac{1}{L_1}\left(1 + \log \frac{M L_1}{L_0}\right)$. We define the stochastic objective for this problem instance as $F_i(x; \xi) = f_i(x)$, so that the true gradient of each local objective is returned by each gradient query of the algorithm.

Note that $f$ is bounded from below and $(L_0, L_1)$-smooth. Also, since all clients have the same objective and gradients are computed deterministically, this problem instance satisfies Assumption 1. Also, our setting of $x_0$ is consistent with the definition of $M$, since

$$|f'(x_0)| = \frac{L_0}{L_1} \exp\left(L_1 x_0 - 1\right)$$
$$= \frac{L_0}{L_1} \frac{M L_1}{L_0} = M.$$

First, define $w := \frac{1}{L_1}\left(1 + \log\left(\frac{\gamma}{\eta}\frac{L_1}{L_0}\right)\right)$. By the condition $\frac{\gamma}{\eta} > M$, we know $w > x_0$, and by construction $f'(w) = \frac{\gamma}{\eta}$. Since $f'$ is increasing on $[0, \infty)$, and decreasing on $(-\infty, 0]$, we know $|f'(w)| \geq \frac{\gamma}{\eta}$ if and only if $|x| \geq w$. This means that the clipping operation will be performed when $|x| \geq w$, and will not be performed for $|x| < w$.

Now we analyze the behavior of clipped minibatch SGD on this problem instance. Suppose that $|x_0| \leq |x_r| \leq w$ for some $r \geq 0$. Then

$$\eta \geq \frac{2L_1 x_0}{L_0 \exp(L_1 x_0 - 1)} \geq \frac{2L_1 |x_r|}{L_0 \exp(L_1 |x_r| - 1)},$$

where the first inequality comes from the definition of $x_0$ and the condition $\eta \geq \frac{2}{L_1 M}\left(1 + \log\frac{L_1 M}{L_0}\right)$, and the second inequality comes from the fact that $\phi(x) := \frac{2L_1 x}{L_0 \exp(L_1 x - 1)}$ is decreasing on $\left[\frac{1}{L_1}, \infty\right)$. Therefore

$$\eta|f'(x_r)| = \eta\frac{L_0}{L_1}\exp(L_1|x_r| - 1) \geq 2|x_r|. \tag{43}$$

Since $\text{sign}(f'(x_r)) \neq \text{sign}(x_r)$, this implies $|x_{r+1}| = |x_r - \eta f'(x_r)| \geq 2|x_r| - |x_r| = |x_r|$. This shows that the sequence of iterates $\{x_r\}$ is non-decreasing in absolute value until (if ever) the absolute value exceeds $w$. If $|x_r| \leq w$ for all $r$, then we are done, since in this case we have $|x_r| \geq |x_0|$ and therefore $|\nabla f(x_r)| \geq |\nabla f(x_0)| = M > \epsilon$ for all $r$.

Otherwise, let $\bar{r}$ be the first index $r$ for which $|x_r| > w$. Without loss of generality, assume $x_{\bar{r}} > 0$ (for $x_{\bar{r}} < 0$, the same argument applies with signs reversed). Since $|x_{\bar{r}-1}| \leq w$,

$$x_{\bar{r}} = x_{\bar{r}-1} - \eta f'(x_{\bar{r}-1}) \leq -w - \eta f'(-w) = \gamma - w.$$

So

$$x_{\bar{r}+1} = x_{\bar{r}} - \gamma \leq (\gamma - w) - \gamma = -w.$$

Therefore $|f'(x_{\bar{r}+1})| \geq |f'(-w)| = |f'(w)| = \frac{\gamma}{\eta}$, and so $x_{\bar{r}+2} = x_{\bar{r}+1} + \gamma = x_{\bar{r}}$. We can then show by induction that $x_{\bar{r}+2n} = x_{\bar{r}}$ and $x_{\bar{r}+2n+1} = x_{\bar{r}+1}$ for all $n \geq 0$. Since $|f'(x_{\bar{r}})|, |f'(x_{\bar{r}+1})| \geq |f'(w)| > |f'(x_0)| > \epsilon$, we have $|f'(x_r)| > \epsilon$ for all $r$. $\qquad\square$

The following lemma is nearly identical to the second half of the proof of Theorem 2 in [8]. We include it here for the sake of completeness.

**Lemma 16.** *Suppose $\frac{\gamma}{\eta} > M$ and $\eta < \frac{2}{L_1 M}\left(1 + \log\frac{L_1 M}{L_0}\right)$. For any $0 < \epsilon < M$, there exists a problem instance $\{f_i\}_{i=1}^N \in \mathcal{F}(L_0, L_1, M, \kappa, \rho, N)$ such that clipped minibatch SGD with parameters $\gamma, \eta$, and any $S$ requires at least*

$$\frac{L_1 M \left(f(x_0) - f^* - \frac{15\epsilon^2}{16L_0}\right)}{2\epsilon^2\left(1 + \log\frac{L_1 M}{L_0}\right)}$$

*rounds in order to find an $\epsilon$-stationary point.*

*Proof.* Consider the following function,

$$f(x) = \begin{cases} -\epsilon x & x < -\frac{3\epsilon}{2L_0} \\ -\frac{L_0^3}{27\epsilon^2}x^4 + \frac{L_0}{2}x^2 + \frac{9\epsilon^2}{16L_0} & x \in \left[-\frac{3\epsilon}{2L_0}, \frac{3\epsilon}{2L_0}\right] \\ \epsilon x & x > \frac{3\epsilon}{2L_0} \end{cases}$$

and the problem instance $f_i = f$ for all $i \in [N]$, with the initialization $x_0 = \frac{3\epsilon}{2L_0} + d$. We define the stochastic objective for this problem instance as $F_i(x, \xi) = f_i(x)$, so that the true gradient of each local objective is returned by each gradient query of the algorithm.

Note that $f$ is bounded from below and $(L_0, L_1)$-smooth. Also, since all clients have the same objective and gradients are computed deterministically, this problem instance satisfies Assumption 1. Also, our setting of $x_0$ is consistent with the definition of $M$, since $|f'(x_0)| = \epsilon < M$.

Now we analyze the behavior of clipped minibatch SGD on this problem instance. Since this problem instance is homogeneous with deterministic gradients, we have $\|g_r\| \leq \epsilon < M < \frac{\gamma}{\eta}$, so clipped minibatch SGD will always perform unnormalized updates. Therefore, for $x_r \geq \frac{3\epsilon}{2L_0}$,

$$x_{r+1} = x_r - \eta g_r = x_r - \eta\epsilon > x_r - \frac{2\epsilon}{L_1 M}\left(1 + \log\frac{L_1 M}{L_0}\right).$$

Therefore

$$x_r > x_0 - \frac{2\epsilon r}{L_1 M}\left(1 + \log\frac{L_1 M}{L_0}\right) = \frac{3\epsilon}{2L_0} + d - \frac{2\epsilon r}{L_1 M}\left(1 + \log\frac{L_1 M}{L_0}\right)$$

as long as $\frac{2\epsilon r}{L_1 M}\left(1 + \log\frac{L_1 M}{L_0}\right) \leq d$, or $r \geq \frac{dL_1 M}{2\epsilon\left(1 + \log\frac{L_1 M}{L_0}\right)}$. Notice that

$$f(x_0) = \frac{3\epsilon^2}{2L_0} + d\epsilon,$$

so

$$d = \frac{1}{\epsilon}\left(f(x_0) - \frac{3\epsilon^2}{2L_0}\right) = \frac{1}{\epsilon}\left(f(x_0) - f^* - \frac{15\epsilon^2}{16L_0}\right).$$

Plugging into the above bound on $r$ tells us that $x_r > x_0$ as long as $r \geq \frac{L_1 M\left(f(x_0) - f^* - \frac{15\epsilon^2}{16L_0}\right)}{2\epsilon^2\left(1 + \log\frac{L_1 M}{L_0}\right)}$. The result follows from the fact that $|f'(x_r)| = \epsilon$ for all $x_r > x_0$. $\qquad\square$

We can now combine Lemmas 13, 15, and 16 to prove Theorem 2.

*Proof of Theorem 2.* Let $\gamma, \eta > 0$ be given. We will prove the requirement on $R$ for three cases of $\gamma, \eta$. In the case $\frac{\gamma}{\eta} \leq M$, then Lemma 13 demonstrates that clipped minibatch SGD cannot guarantee to find an $\epsilon$-stationary point with probability greater than $1 - \delta$ for any $R$. Second, if $\frac{\gamma}{\eta} > M$ and $\eta \geq \frac{2}{L_1 M}\left(1 + \log\frac{L_1 M}{L_0}\right)$, then Lemma 15 demonstrates that clipped minibatch SGD can fail to find an $\epsilon$-stationary point for any $R$. In the remaining case, i.e. $\frac{\gamma}{\eta} > M$ and $\eta < \frac{2}{L_1 M}\left(1 + \log\frac{L_1 M}{L_0}\right)$, Lemma 16 demonstrates that for finding an $\epsilon$-stationary point, the number of iterations clipped minibatch SGD requires is at least

$$R \geq \frac{L_1 M\left(f(x_0) - f^* - \frac{15\epsilon^2}{16L_0}\right)}{2\epsilon^2\left(1 + \log\frac{L_1 M}{L_0}\right)}.$$

$\qquad\square$

## C   Additional Experimental Results

### C.1   Hyperparameter Information

**Learning Rate and Clipping Threshold**   For each dataset, we tune the hyperparameters $\gamma$ and $\eta$ with separate grid searches. Specifically, we first tune the clipping threshold $\frac{\gamma}{\eta}$ with grid search, then we tune the learning rate $\eta$ with grid search. During the grid search for $\eta$, $\gamma$ is chosen so that the clipping threshold $\frac{\gamma}{\eta}$ is equal to the tuned value from the search for $\frac{\gamma}{\eta}$. Due to computational constraints, we do not perform this search for every algorithm in every setting. Instead, we follow the tuning process of [9]: we use the above procedure to tune $\eta$ and $\gamma$ for CELGC [32] separately for each $S \in \{2, 4, 6, 8\}$, while fixing $s = 30\%$ (SNLI) or $s = 10\%$ (Sent140). We then reuse the tuned values of $\gamma$ and $\eta$ for all other algorithms and settings sharing the same dataset and $S$. Our theory suggests that the best learning rate $\eta$ depends on the number of participating clients $S$, and we follow this guidance by tuning the parameters separately for each value of $S$. The search ranges and tuned values for each dataset are shown in Table 3.

Table 3: Hyperparameter tuning information for each dataset.

| | Parameter | Search range | Tuned value |
|---|---|---|---|
| SNLI | $\frac{\gamma}{\eta}$ | $\{0.01, 0.03, 0.1, 0.3, 1.0, 3.0\}$ | $S = 2$: 1.0
$S = 4$: 1.0
$S = 6$: 1.0
$S = 8$: 1.0 |
| | $\eta$ | $\{0.003, 0.01, 0.03, 0.1\}$ | $S = 2$: 0.03
$S = 4$: 0.03
$S = 6$: 0.03
$S = 8$: 0.03 |
| Sent140 | $\frac{\gamma}{\eta}$ | $\{0.01, 0.03, 0.1, 0.3, 1.0, 3.0\}$ | $S = 2$: 0.3
$S = 4$: 0.3
$S = 6$: 0.3
$S = 8$: 0.3 |
| | $\eta$ | $\{0.003, 0.01, 0.03, 0.1\}$ | $S = 2$: 0.03
$S = 4$: 0.03
$S = 6$: 0.03
$S = 8$: 0.03 |

**Network Architecture** For both datasets, the network is composed of a one-layer bidirectional RNN encoder followed by a three-layer fully connected classifier. The encoder has hidden size 2048 and max pooling, and the decoder has hidden size 512 with tanh activations. Input sentences are encoded as sequences of GloVe vectors. For SNLI, where each input is a pair of sentences, each sentence is passed through the encoder separately, and the resulting representations are concatenated and used as input for the encoder.

## C.2 Learning Curves

Figures 4 and 5 contain learning curves (by communication rounds) of training loss and testing accuracy for all settings, for SNLI and Sentiment140 respectively. The experiments show that our proposed algorithm EPISODE++ significantly outperform all other baselines in various settings on these two datasets, including different client participation ratio and different heterogeneity level.

## C.3 Training with Homogeneous Data

Here we include learning curves for an additional experiment that uses homogeneous data across clients, i.e. $s = 0\%$. In this setting, we use $N = 8$ and $S = 4$, and all other settings are the same as described in Section 6. We compare the six algorithms described in Section 6. Results are shown in Figure 6. The results are consistent with the experiments that use heterogeneous data: EPISODE++ outperforms all other algorithms in terms of training loss and testing accuracy.

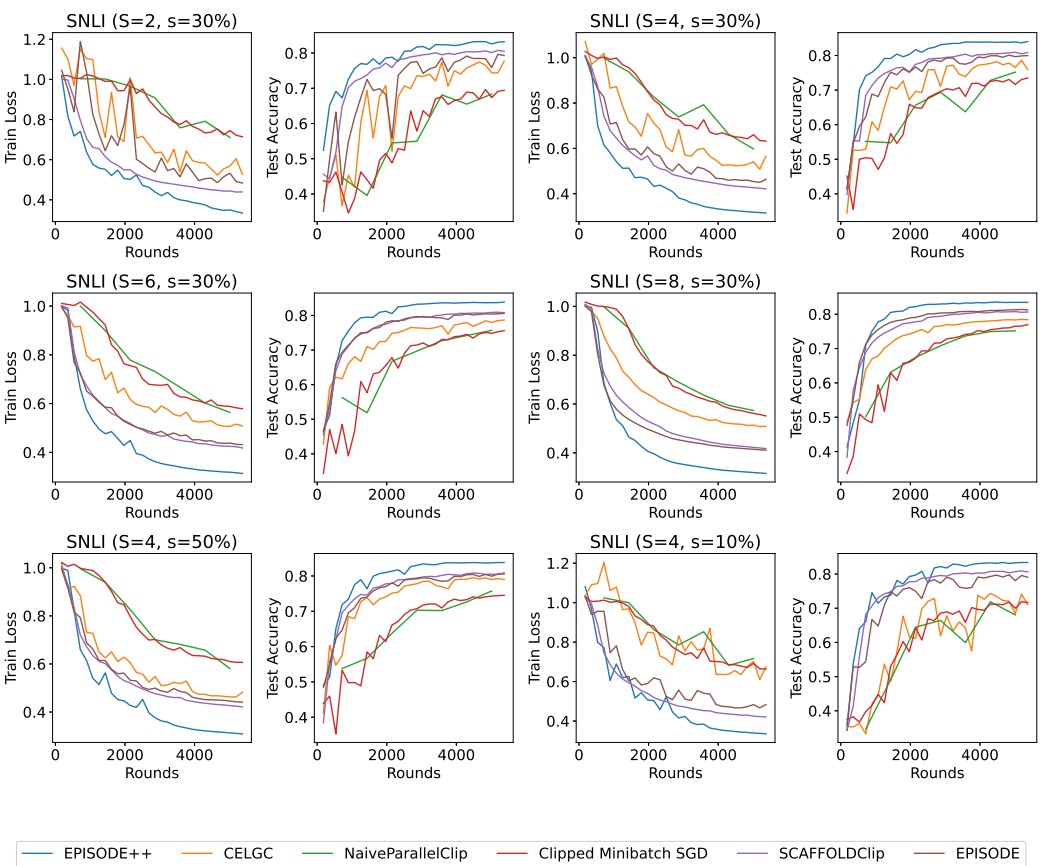

Figure 4: Learning curves for SNLI under all settings. For NaiveParallelClip, we show the first 5375 rounds to compare all algorithms with a fixed number of communication rounds.

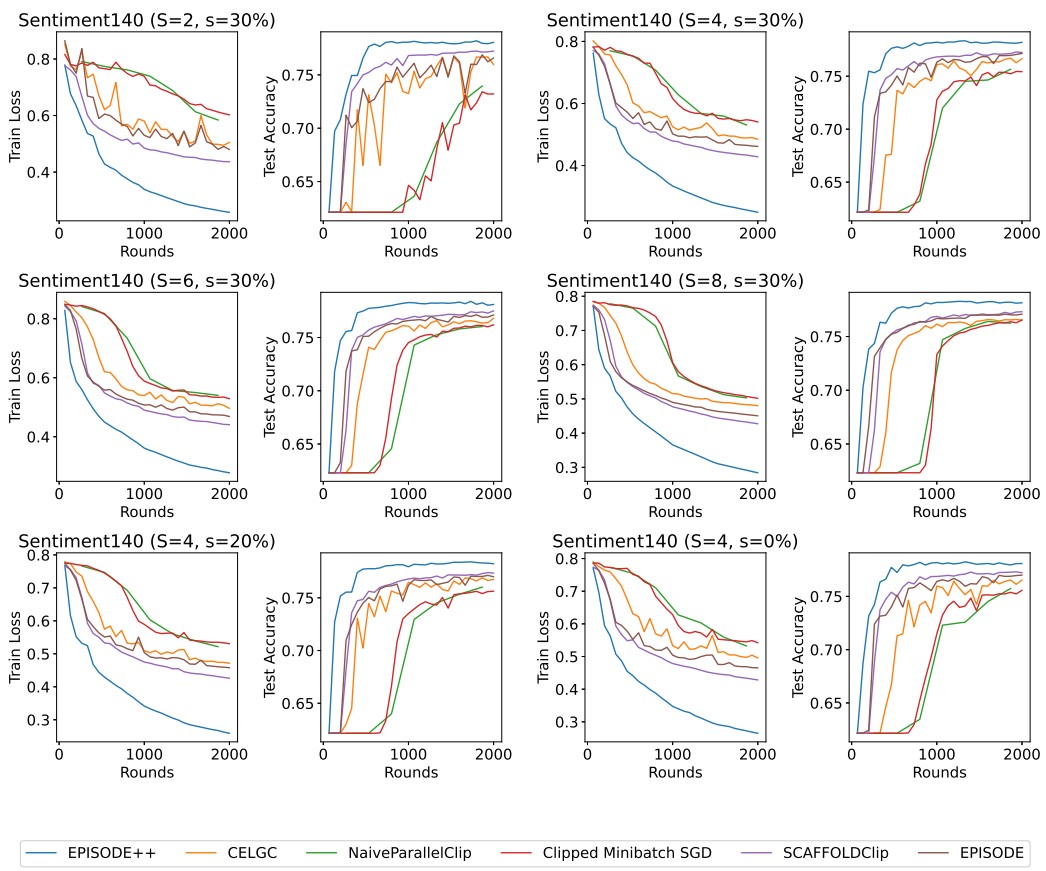

Figure 5: Learning curves for Sentiment140 under all settings. For NaiveParallelClip, we show the first 2000 rounds to compare all algorithms with a fixed number of communication rounds.

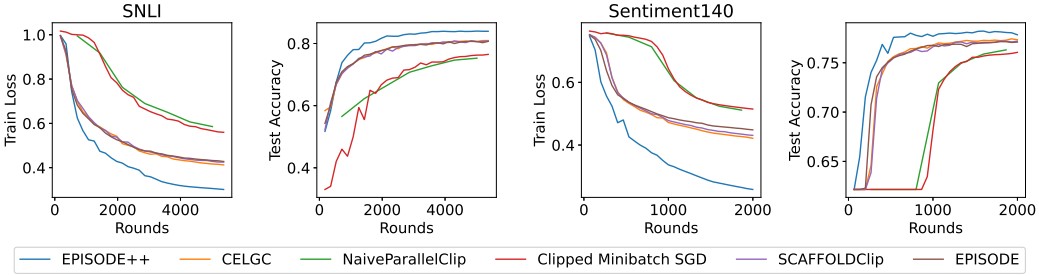

Figure 6: Learning curves for SNLI and Sentiment140 with $N = 8, S = 4$, using homogeneous data, i.e. $s = 100\%$. We compare all algorithms with a fixed number of communication rounds.

