# OpenReview forum: "Federated Learning with Client Subsampling, Data Heterogeneity, and Unbounded Smoothness: A New Algorithm and Lower Bounds"
_NeurIPS.cc/2023/Conference — NeurIPS 2023 poster_

### Official Review · Reviewer_svY2 · 2023-06-24

**Soundness:** 3 good
**Presentation:** 3 good
**Contribution:** 2 fair
**Rating:** 6
**Confidence:** 3

**Summary:**

 This manuscript introduces a new federated learning algorithm, EPISODE++, to accelerate the convergence speed of federated learning and provides theoretical proofs for the upper bound of convergence speed.

**Strengths:**

The method proposed in this manuscript improves the EPISODE algorithm by addressing the issue of client heterogeneity while allowing partial client participation. The manuscript also provides theoretical proofs for the convergence speed.

**Weaknesses:**

The experimental section does not explain how the hyperparameters for each baseline method were selected, and it does not discuss how the performance of the method is affected when the number of clients reaches hundreds.

**Questions:**

1. In Figure 1 (a) and (b), the left two figures show that NaiveParallelClip has lower training loss than other methods when the number of clients is 8, but its testing accuracy is lower than that of EPISODE++. How can this phenomenon be explained?

2. How does the performance of the method change when there are more clients (resulting in larger variances for $G_{r+1}$ and $G_{r+1}^i$)? Does it outperform classical methods like fedprox?

3. How were the hyperparameters (learning rate, training steps) chosen for each method?

4. How does the proposed method perform when the client data are i.i.d.?


**Limitations:**

The method proposed in this manuscript only slightly modifies the EPISODE method, and the experimental section is not comprehensive.

---

> ### Author Rebuttal · Authors · 2023-08-09
>
> Thank you for your effort in providing valuable feedback. Below, we have individually addressed the questions in your review.
>
> **Q1: “In Figure 1 (a) and (b), the left two figures show that NaiveParallelClip has lower training loss than other methods when the number of clients is 8, but its testing accuracy is lower than that of EPISODE++. How can this phenomenon be explained?”**
>
> This can be explained by the fact that NaiveParallelClip with $S=8$ is essentially simulating large-batch SGD, since it eliminates local steps and computes the update by averaging the gradient across ALL clients. It has been shown [1] that large-batch SGD can overfit in deep learning, and this may cause the overfitting in the case $S=8$. Notice that when $S<8$, NaiveParallelClip is not simulating large-batch SGD because it is still missing information from unsampled clients. Also, we would like to reiterate that NaiveParallelClip is not a practical algorithm due to the significant communication cost.
>
> **Q2: “How does the performance of the method change when there are more clients?”**
>
> In our 1 page rebuttal PDF, we have included results for large-scale experiments in two settings: (i) SNLI with $N=128, S=16, s=30\\%$ and (ii) Sent140 with $N=128, S=16, s=10\\%$. All other configuration details are the same as in the main text. The results are shown in Figure 2 of the 1 page rebuttal PDF.
>
> In both settings, the relative performance of each algorithm is similar to that of the main text experiments (i.e. those with $N=8$). The proportion of participating clients is $S/N=1/8$ for the large-scale experiments and $S/N \geq 1/4$ for the main text experiments, so that the effect of client sampling may be stronger in the large-scale experiments. With $N=128$, EPISODE++ achieves a better training loss for both datasets than in any setting with $N=8$, while the testing accuracy of EPISODE++ is about the same as the highest testing accuracy of the $N=8$ settings. Further, EPISODE++ outperforms all other algorithms in the $N=128$ setting.
>
> **Q3: “Does it outperform classical methods like fedprox?”**
>
> We have included additional results comparing EPISODE++ and FedProx in the 1 page rebuttal PDF. In this experiment, we used both algorithms for the SNLI setting as described in the main text, with $N=8, S=2, s=10\\%$. For FedProx, we tuned the additional parameter $\mu$ over $\{0.01, 0.03, 0.1, 0.3, 1.0\}$, and the best tuned value according to test accuracy was $\mu = 0.03$. The results are included in Table 2 of the 1 page rebuttal PDF. Due to time constraints, we were not able to evaluate other baselines in this setting.
>
> As shown in Table 2, EPISODE++ achieves a lower training loss and higher training accuracy than FedProx. Similar to FedAvg, CELGC, and NaiveParallelClip, FedProx does not utilize gradient information from unsampled clients, which suggests that the performance of FedProx should degrade under data heterogeneity and client sampling. Since EPISODE++ utilizes information from ALL clients in the form of correction terms, EPISODE++ may be more resilient to data heterogeneity and client sampling, and indeed we observe this in Table 2.
>
> **Q4: “How were the hyperparameters (learning rate, training steps) chosen for each method?”**
>
> The learning rate $\eta$ and the clipping parameter $\gamma$ were tuned according to a grid search described in Appendix C.1, which is referenced on line 260 of the main text. The number of training steps was chosen to be long enough that the test accuracy stopped increasing.
>
> **Q5: “How does the proposed method perform when the client data are i.i.d.?”**
>
> We have included additional experimental results using i.i.d. data in the 1 page rebuttal PDF. We evaluated EPISODE++ and all baselines in two settings: (i) the SNLI dataset with $N=8, S=4, s=100\\%$ and (ii) the Sent140 dataset with $N=8, S=4, s=100\\%$. All other configuration details remain the same as the experiments from the main text. The results are shown in Figure 1 of the 1 page rebuttal PDF.
>
> For both datasets, EPISODE++ remains the best performing method by both training loss and testing accuracy. All algorithms (besides CELGC) perform similarly with homogeneous data as in the counterpart settings with heterogeneous data.
>
> **Q6: “The method proposed in this manuscript only slightly modifies the EPISODE method”**
>
> There are two important differences between EPISODE and EPISODE++, one practical and one theoretical. The first is that EPISODE++ does not perform double communication at each round, since the correction terms $\mathbf{G}_r^i$ are computed using information from previous rounds. In EPISODE, these correction terms depend on the new averaged model $\bar{\mathbf{x}}_r$. Computing corrections requires one communication operation to broadcast $\bar{\mathbf{x}}_r$ and then another to share the newly computed corrections. Note that this is not just a doubling of the communicated bits: since $\mathbf{G}_r^i$ must be computed after the end of the first communication operation and before the beginning of the second, the two communication operations cannot overlap, and there must be a doubling of the required communication time. The halving of the communication cost in EPISODE++ is a significant practical benefit over EPISODE.
>
> The second important difference is that the introduction of client sampling brings new challenges for the convergence analysis which cannot be handled by the approach used to analyze EPISODE. To handle these new challenges we introduced a nested recursive analysis of the update size (Lemmas 1, 10, and 11) and provided a high probability guarantee of convergence (as opposed to the expectation guarantee of EPISODE).
>
> Thank you for your time, and please let us know if we have addressed your concerns.
>
> [1] Keskar, Nitish Shirish, et al. "On Large-Batch Training for Deep Learning: Generalization Gap and Sharp Minima." International Conference on Learning Representations. 2016.

---

> > ### Comment · Reviewer_svY2 · 2023-08-16
> >
> > Thanks for the response, I have changed the score to 6.

---

### Official Review · Reviewer_sD3Q · 2023-07-04

**Soundness:** 3 good
**Presentation:** 2 fair
**Contribution:** 3 good
**Rating:** 6
**Confidence:** 3

**Summary:**

The authors propose a federated learning algorithm that can work under (L0,L1)-smooth function. Different from the previous work, the authors consider the partial-participant setting, modifying the previous algorithm, showing the convergence of the new algorithm, and giving a lower bound of the communication iterations under this setting. The experiments show that the proposed algorithm performs well.

**Strengths:**

1. The authors propose a new algorithm that overcomes the bias introduced by client heterogeneity and client sampling.

2. The authors prove the convergence of the proposed algorithm and a lower bound of communication iterations under this setting.

3. The experimental results show the the proposed algorithm performs much better than previous work.

**Weaknesses:**

1. The key motivation is not clear to me. For me, it is hard to justify that with a uniform sampling strategy on clients, why the bias will occur? And it is hard to see how the proposed algorithm can fix the introduced bias.



**Questions:**

See weakness.

---

> ### Author Rebuttal · Authors · 2023-08-09
>
> Thank you for your time and for providing helpful comments. Below we have addressed the concern you expressed in your review.
>
> **Q1: “The key motivation is not clear to me. For me, it is hard to justify that with a uniform sampling strategy on clients, why the bias will occur?”**
>
> To discuss the bias from a theoretical perspective, we would like to clarify the meaning of bias in this context versus heterogeneity. They are different concepts in federated learning. Denoting the $i$-th client objective as $F_i$, consider the error $\nabla F_i - \nabla F$, where $i$ is sampled uniformly over $\{1, \ldots, n\}$. The expectation over $i$ of this error (i.e., sampling bias) is zero, i.e., $\mathbb{E} [ \nabla F_i(x) - \nabla F(x) ] = 0$ for all $x$. However, if we take the expectation of the squared norm of this error (i.e., sampling variance), we get a dependence on the heterogeneity $\kappa$, i.e. $\mathbb{E} \left[ \lVert \nabla F_i(x) - \nabla F(x) \rVert^2 \right\] \leq \kappa^2$. The bias in the first case is zero because the sampling is uniform and the error from different clients cancel each other. The expected $\ell_2$ error in the second case is non-zero because we consider the norm of the error and no canceling may occur: the expected norm depends on the heterogeneity $\kappa$. If the algorithm only uses gradient information from sampled clients (such as FedAvg, CELGC, NaiveParallelClip), then the update direction is approximating $ \frac{1}{S} \sum_{i \in \mathcal{S}} \nabla F_i(x)$, and the convergence will be slowed with a dependence on the heterogeneity $\kappa$ due to sampling variance, i.e., $\mathbb{E} \left[ \lVert \frac{1}{S} \sum_{i \in \mathcal{S}} \nabla F_i(x) - \nabla F(x) \rVert \right\] \leq \kappa$. Therefore the heterogeneity is introduced even under uniform sampling.
>
> The effect of client heterogeneity and client sampling can also be understood empirically. Several works have empirically demonstrated that the performance of FedAvg with uniform sampling and data heterogeneity decreases as the number of participating clients decreases (see Table 4 of [1] or Figure 1 of [2]), which demonstrates that this effect does occur even under uniform sampling.
>
> The motivation of this paper is designing computation and communication-efficient algorithms in the relaxed smoothness setting and client subsampling. Our Theorem 1 shows that our computational complexity does not depend on heterogeneity level $\kappa$.
>
> Thank you for your time, and please let us know if we have addressed your concern.
>
> [1] Karimireddy, Sai Praneeth, et al. "Scaffold: Stochastic controlled averaging for federated learning." International conference on machine learning. PMLR, 2020.
>
> [2] Li, Tian, et al. "Federated optimization in heterogeneous networks." Proceedings of Machine learning and systems 2 (2020): 429-450.

---

> > ### Comment · Reviewer_sD3Q · 2023-08-15
> >
> > Sorry, the explanation is still confusing. In the answer, you defined bias as heterogeneity $\kappa$, which I think is similar to $\kappa$ defined in the paper, because, for the special $\rho =1$, $\kappa$ is something related to the variance. However, in Theorem 1, R is in the order of $max(\Gamma1, \Gamma2)$, $\Gamma1 = O(\kappa)$ and $\Gamma2 = O(\kappa^2)$. Thus, the $\kappa$ does affect the result in Theorem 1.
> >
> > Meanwhile, it seems that in the paper, the bias is introduced by clipping instead of heterogeneity, can you comment something for line 144-146?

---

> > > ### Author Response · Authors · 2023-08-16
> > >
> > > Actually, $RI$ (the iteration complexity) is independent of $\kappa$ for sufficiently small $\epsilon.$ You can see in the statement of Corollary 1 that $\epsilon$ is required to be small enough that $\eta = \frac{S \epsilon^2}{216 AL_0 \sigma^2 \log \frac{1}{\delta}}$, i.e., the third term in the min of Equation 2 is the minimum. You may check the proof of Corollary 1 in Appendix to see that the derivation of the iteration complexity $RI$ is correct, and the result is independent of $\kappa$ under this condition of sufficiently small $\epsilon$. It should be noted that the communication complexity $R$ has a dependence on $\kappa$. In SCAFFOLD, $R$ indeed does not depend on $\kappa$, but only under the condition of smoothness. Under relaxed smoothness, the only prior work EPISODE [1] also has $R$ which depends on $\kappa$, even in the case of full client participation. The experiments from [1] also show that SCAFFOLD cannot work in the relaxed smoothness case because SCAFFOLD does not use gradient clipping. Under relaxed smoothness, gradient clipping is necessary, and this clipping operator introduces the dependence of $R$ on $\kappa$.
> > >
> > > The word bias is used to refer to multiple sources of error, so there may be some confusion. Lines 144-146 are describing the same source of error that we described in our rebuttal. Lines 144-146 say that a naive extension of EPISODE to the subsampling case would set $\mathbf{G}_r^i = \nabla F_i(\bar{\mathbf{x}}_r; \tilde{\xi}_r^i)$ and  $\mathbf{G}_r = \frac{1}{S} \sum\_{i \in S_r} \mathbf{G}_r^i$, so that clipping would occur if $\lVert \mathbf{G}_r \rVert \geq \frac{\gamma}{\eta}$. $\mathbf{G}_r^i$ is an estimate of $\nabla F_i(\bar{\mathbf{x}}_r)$, so $\mathbf{G}_r$ is an estimate of $\frac{1}{S} \sum\_{i \in S_r} \nabla F_i(\bar{\mathbf{x}}_r)$. However, in order to simulate updates according to the global objective, the algorithm should perform clipping according to the norm of $\nabla F(\bar{\mathbf{x}}_r)$, whose distance to the value used by naive EPISODE is $ \kappa_S := \lVert \nabla F(\bar{\mathbf{x}}_r) - \frac{1}{S} \sum\_{i \in S_r} \nabla F_i(\bar{\mathbf{x}}_r) \rVert $.
> > >
> > > In this context, $\kappa_S$ is causing a bias in the vector whose norm determines clipping. In our rebuttal, we discussed how $\kappa_S$ may slow down optimization with a dependence on $\kappa$, since the update direction of e.g. FedAvg or CELGC differs from the global update direction of $- \nabla F(\bar{\mathbf{x}}_r)$ by $\kappa_S$. Because $\kappa_S$ is introduced by subsampling and it affects optimization (even without clipping), the bias is not due to clipping itself. The motivation of our paper is to design an efficient algorithm for federated learning under heterogeneity, relaxed smoothness, and client sampling, and dealing with the errors introduced by $\kappa_S$ is a main challenge of this problem. Please let us know if we have answered your concerns.
> > >
> > > [1] Crawshaw, Michael, Yajie Bao, and Mingrui Liu. "EPISODE: Episodic Gradient Clipping with Periodic Resampled Corrections for Federated Learning with Heterogeneous Data." The Eleventh International Conference on Learning Representations. 2022.

---

> > > > ### Comment · Reviewer_sD3Q · 2023-08-17
> > > >
> > > > Thank you for your response, I have no more questions.

---

### Official Review · Reviewer_PGRH · 2023-07-05

**Soundness:** 4 excellent
**Presentation:** 3 good
**Contribution:** 3 good
**Rating:** 6
**Confidence:** 3

**Summary:**

The paper presents a novel algorithm for non-convex federated learning that addresses the challenges of relaxed smoothness, client heterogeneity, and client subsampling. The authors begin by discussing the limitations of existing algorithms such as SCAFFOLD and EPISODE in handling these challenges simultaneously. They then introduce their algorithm, EPISODE++, which is designed to overcome these limitations. The algorithm is initialized with a set of parameters and then iteratively updated through a series of communication rounds. In each round, a subset of clients is selected randomly, and each client performs local updates based on its own data. The algorithm also includes a gradient clipping step to control the magnitude of the gradient updates.

The authors provide a detailed theoretical analysis of the convergence properties of EPISODE++. They show that the algorithm can achieve a linear speedup over the standard federated averaging algorithm (FedAvg) under certain conditions. They also demonstrate that EPISODE++ can significantly outperform clipped minibatch SGD, another popular algorithm for non-convex optimization, in terms of the number of iterations required to find an epsilon-stationary point. The paper concludes with an extensive set of experiments that validate the theoretical findings. The authors show that EPISODE++ outperforms other state-of-the-art algorithms on a variety of benchmark datasets and neural network architectures, including LSTMs and Transformers.

**Strengths:**

1. The authors provide a detailed theoretical analysis of the convergence properties of EPISODE++. They show that under certain conditions, the algorithm can achieve a linear speedup over the standard federated averaging algorithm (FedAvg), and significantly outperform clipped minibatch SGD in terms of the number of iterations required to find an epsilon-stationary point. This rigorous analysis strengthens the credibility of the proposed algorithm.

2. The paper includes an extensive set of experiments that validate the theoretical findings. The authors demonstrate that EPISODE++ outperforms other state-of-the-art algorithms on a variety of benchmark datasets and neural network architectures, including LSTMs and Transformers. There are many other FL papers that only conduct very toy experiments that are not convincing enough. This empirical evidence provides strong support for the effectiveness of the proposed algorithm.

**Weaknesses:**

While the paper presents a novel algorithm and provides a detailed theoretical analysis, the scope of the study appears to be limited to non-convex federated learning. It would be interesting to see how EPISODE++ performs in other contexts or problem domains.

**Questions:**

The paper could provide more information about the implementation details of the EPISODE++ algorithm. For instance, how are the parameters of the algorithm chosen in practice? Are there any specific strategies or heuristics for setting these parameters? Providing such information could help other researchers to replicate the results and apply the algorithm to their own problems.

**Limitations:**

Yes

---

> ### Author Rebuttal · Authors · 2023-08-09
>
> Thank you for taking the time to review our paper and give valuable feedback. See the list below, where we have individually responded to the questions and comments in your review.
>
> **Q1: “the scope of the study appears to be limited to non-convex federated learning.”**
>
> Our algorithm is indeed designed for the federated learning task, and the analysis focuses on the non-convex case, since it is more general (and includes) the convex case. Because of this, our theory provides a guarantee on the gradient norm $\lVert \nabla F(x_t) \rVert$ along the trajectory with high probability. In the strongly convex case, this guarantee implies a guarantee on the objective gap $F(x_t) - F^*$ with high probability, where $F^*$ is the global minimum of the objective function $F$. Therefore, if we would like to restrict our attention from the general non-convex case to the special strongly convex case, our analysis immediately gives guarantees. Also, the problem of finding a point with small gradient is of independent interest [1, 2] outside of its implications for the “objective gap” even in the convex case.
>
> **Q2: “how are the parameters of the algorithm chosen in practice? Are there any specific strategies or heuristics for setting these parameters?”**
>
> The parameters of EPISODE++ are the learning rate $\eta$, the clipping parameter $\gamma$, and the communication interval $I$, which are all parameters of previously existing algorithms such as FedAvg ($\eta$ and $I$) and CELGC ($\eta$, $\gamma$, and $I$). These parameters may be chosen following standard conventions of federated learning and machine learning in general.
>
> To choose hyperparameters in the experiments of the paper, we tuned the learning rate $\eta$ and the clipping parameter $\gamma$ according to a grid search that is described in Appendix C.1, which is referenced on Line 260 of the main text. After tuning the parameters for CELGC [3], we found that the tuned set of parameters worked well for the remaining algorithms, so we re-used these parameter values for all baselines.
>
> Thank you for your time, and please let us know if we have addressed your concerns.
>
> [1] Yurii Nesterov. “How to make the gradients small.” Optima, 88:10–11, 2012.
>
> [2] Allen-Zhu, Zeyuan. "How to make the gradients small stochastically: Even faster convex and nonconvex sgd." Advances in Neural Information Processing Systems 31 (2018).
>
> [3] Mingrui Liu, Zhenxun Zhuang, Yunwen Lei, and Chunyang Liao. A communication-efficient
> distributed gradient clipping algorithm for training deep neural networks. Advances in Neural Information Processing Systems, 35:26204–26217, 2022.

---

> > ### Comment · Reviewer_PGRH · 2023-08-19
> >
> > Thank you for your feedback. After reading the comments and rebuttals of other reviewers, I decided to keep my original score.

---

### Official Review · Reviewer_xXDR · 2023-07-11

**Soundness:** 3 good
**Presentation:** 4 excellent
**Contribution:** 3 good
**Rating:** 7
**Confidence:** 3

**Summary:**

The paper investigates Federated Learning (FL) under client subsampling and data heterogeneity, focusing on functions with potentially unbounded smoothness, and introduces the proposed algorithm to address the problem, EPISODE++. EPISODE++ has demonstrated benefits including linear speedup with client numbers, reduced communication rounds, and resilience to data heterogeneity. The authors provide theoretical convergence analysis and experimental results validate the effectiveness of their method.

**Strengths:**

1. The paper is well-written and easy to follow.
2. Provides novel techniques when proofing both upper and lower bounds.
3. Theoretically demonstrates the benefit of proposed methods EPISODE++, and also shows the shortage of existing clipped minibatch SGD.
4. The proposed algorithm achieves significant improvement over existing methods.


**Weaknesses:**

1. The experiments with N=8 may not fully reflect the actual performance with large-scale FL, for example, FedAvg uses 100 clients in their experiments, where more heterogeneity and client shift might result in different behavior. I would be happy to see if there's more benefit of EPOSIDE++
2. Missing error bar and not indicates the number of repetition runs in experiments.

**Questions:**

See weakness

**Limitations:**

The authors have mentioned limitations of their work.

---

> ### Author Rebuttal · Authors · 2023-08-09
>
> Reviewer xXDR (7): Thank you for taking the time to review our paper. Below we have addressed the concerns you raised during your review.
>
> **Q1: “The experiments with N=8 may not fully reflect the actual performance with large-scale FL”**
>
> We agree that evaluating the proposed algorithm on large-scale experiments is important. In our 1 page rebuttal PDF, we have included results for large-scale experiments in two settings: (i) the SNLI dataset with $N = 128, S = 16, s = 30\\%$ and (ii) the Sent140 dataset with $N = 128, S = 16, s = 10\\%$, where $s$ stands for data similarity. Note that these experiments use the same dataset as those from the main text, and the dataset is split into a larger number of clients using the heterogeneity protocol described in the main text. We ran these experiments using 16 GPUs on a cluster of 4 nodes with 4 GPUs each. The results are shown in Figure 2 of the 1 page rebuttal PDF.
>
> In both settings, the relative performance of each algorithm is similar to that of the main text experiments (i.e., those with $N = 8$). Note that the proportion of participating clients $S/N=1/8$ for the large-scale experiments and $S/N \geq 1/4$ for the main text experiments, so that the effect of partial client participation may be stronger in the large-scale experiments. With $N=128$, EPISODE++ achieves a better training loss for both settings than in any experiment with $N = 8$, while the testing accuracy of EPISODE++ is about the same as the highest testing accuracy of the $N = 8$ experiments. Further, EPISODE++ outperforms all other algorithms in the $N = 128$ setting.
>
> **Q2: “Missing error bar and not indicates the number of repetition runs in experiments.”**
>
> Again, we agree that multiple repetition runs is important to properly evaluate our proposed algorithm against baselines. To ensure that our results are representative of the performance of each algorithm, we have included results with a total of three repetitions for two experimental settings: (i) the SNLI dataset with $N = 8, S = 4, s = 30\\%$ and (ii) the Sent140 dataset with $N = 8, S = 4, s = 10\\%$. All other hyperparameters and configuration details remain the same as the results reported in the main text. The average results and error bars are shown in Table 1 of the 1 page rebuttal PDF. Note that the size of each error bar is the distance from the average over three trials to the min/max over three trials.
>
> Across three trials, EPISODE++ remains the best performing algorithm in terms of training loss and testing error: the worst trial of EPISODE++ is better than the best trial of any other algorithm, across both metrics and both datasets. The ordering of the algorithms by performance in the trial-averaged results is the same as in the single-trial results of the main text, which is consistent with the experimental results of the main text.
>
> Thank you for your time, and please let us know if you have additional comments.

---

> > ### Comment · Reviewer_xXDR · 2023-08-21
> >
> > After reading all reviews and responses, I will maintain the current score.

---

### Author Rebuttal · Authors · 2023-08-09

We would like to thank all of the reviewers for taking time to review and critique our work. We have provided an individual response to each reviewer, and here we provide a general summary of our additional results included in the 1 page rebuttal PDF.

**Large scale experiments**

In the 1-page rebuttal PDF file, we have included large scale experimental results with a large number of clients $N = 128$. Results are shown in Figure 2 of the 1-page rebuttal PDF and are further described in the individual responses. The results with $N = 128$ are consistent with the experiments of the main text that use $N = 8$. EPISODE++ outperforms all other algorithms in terms of both training loss and testing accuracy, for the two datasets SNLI and Sentiment140.

**Multiple trials**

We have additionally included results for all algorithms averaged over three random seeds, in order to ensure that our experimental results are representative of the expected performance for each algorithm. In general, the results of the additional trials match that of the results reported in the main text: EPISODE++ outperforms all baselines, and the performance of each algorithm is stable across the three seeds. The results are shown in Table 1 of the 1-page rebuttal PDF file and are further described in the individual responses.

**Homogeneous data**

We also evaluated our algorithm and baselines in two settings with homogeneous data, and results are shown in Figure 1 of the 1-page rebuttal PDF. Every algorithm has similar or better performance compared with the heterogeneous setting, and EPISODE++ remains the best performing algorithm by a wide margin.

**Comparison with FedProx**

We compared EPISODE++ against the algorithm FedProx on the SNLI dataset. EPISODE++ achieves a lower training loss and a higher testing accuracy. This gap follows intuition, since FedProx does not use any explicit mechanism to combat client subsampling and data heterogeneity.

---

### Decision · Program_Chairs · 2023-09-21

**Decision:**

Accept (poster)

**Comment:**

Given the scores, and also looking at the paper's writing and presentation as well as the discussions, I am recommending acceptance as a poster.